# GEPA: Reflective Prompt Evolution Can Outperform Reinforcement Learning

Lakshya A Agrawal[1], Shangyin Tan[1], Dilara Soylu[2], Noah Ziems[4],
Rishi Khare[1], Krista Opsahl-Ong[5], Arnav Singhvi[2,5], Herumb Shandilya[2],
Michael J Ryan[2], Meng Jiang[4], Christopher Potts[2], Koushik Sen[1],
Alexandros G. Dimakis[1,3], Ion Stoica[1], Dan Klein[1], Matei Zaharia[1,5], Omar Khattab[6]

[1]UC Berkeley    [2]Stanford    [3]BespokeLabs.ai    [4]Notre Dame    [5]Databricks    [6]MIT

## Abstract

Large language models (LLMs) are increasingly adapted to downstream tasks via reinforcement learning (RL) methods like Group Relative Policy Optimization (GRPO), which often require thousands of rollouts to learn new tasks. We argue that the interpretable nature of *language* often provides a much richer learning medium for LLMs, compared to policy gradients derived from sparse, scalar rewards. To test this, we introduce GEPA (**Ge**netic-**Pa**reto), a prompt optimizer that thoroughly incorporates *natural language reflection* to learn high-level rules from trial and error. Given any AI system containing one or more LLM prompts, GEPA samples trajectories (e.g., reasoning, tool calls, and tool outputs) and reflects on them in natural language to diagnose problems, propose and test prompt updates, and combine complementary lessons from the Pareto frontier of its own attempts. As a result of GEPA's design, it can often turn even just a few rollouts into a large quality gain. Across six tasks, GEPA outperforms GRPO by 6 percentage points on average and by up to 19pp, while using up to $35\times$ fewer rollouts. GEPA also outperforms the leading prompt optimizer, MIPROv2, by over 10 percentage points (e.g., +12pp on AIME-2025), and demonstrates promising results as an inference-time search strategy for code optimization. We release our code at https://github.com/gepa-ai/gepa.

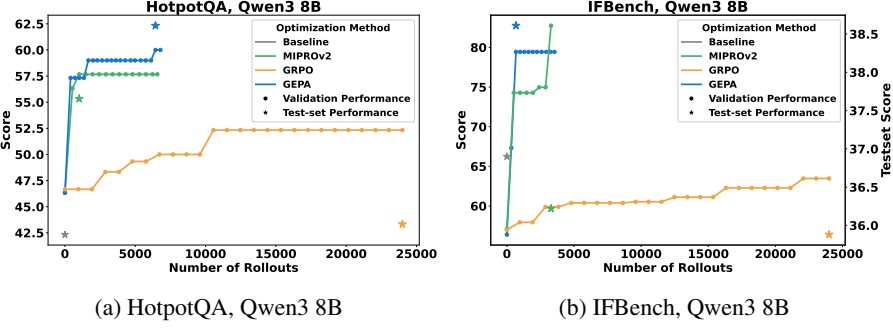

(a) HotpotQA, Qwen3 8B    (b) IFBench, Qwen3 8B

Figure 1: A comparison of learning behavior of the GEPA prompt optimizer against a state-of-the-art prompt optimizer (MIPROv2) and GRPO (24,000 rollouts). As more rollouts are sampled, the prompt optimizers can learn much more quickly than GRPO. GEPA substantially outperforms both GRPO and MIPROv2 in final score. The Test-set star markers demonstrate the performance gap in a held-out set of questions.

## 1 Introduction

Large language models (LLMs) have enabled development of agents and systems that combine fuzzy natural-language specifications with tools like retrieval and code execution. This raises the question of how LLMs should be optimized for downstream performance. One popular approach is Reinforcement Learning with Verifiable Rewards (RLVR), e.g. with Group Relative Policy Opti-

mization (GRPO) (Shao et al., 2024), which treats success metrics as end-of-rollout scalar rewards used to estimate policy gradients (Lambert, 2025). While these RL approaches are effective, they typically require tens of thousands of rollouts in practice to fit new tasks. For example, recent works leveraging GRPO typically use up to hundreds of thousands of rollouts for training (Chen et al., 2025b; Wu et al., 2025b; Zhang et al., 2025; Jin et al., 2025; Si et al., 2025; Wang et al., 2025a; Chen et al., 2025a; Sha et al., 2025; Lin et al., 2025a; Peng et al., 2025; Song et al., 2025). This sample inefficiency can quickly become a serious bottleneck: many downstream LLM applications invoke expensive tool calls, have limited inference budget for sampling from the LLM itself, or simply cannot finetune the weights of the largest or best-performing LLMs.

We observe that rollouts sampled from even highly sophisticated LLM systems can be serialized into traces of natural language: they contain nothing but the instructions of each LLM module, the resulting LLM reasoning chains, tool calls, and potentially the internal workings of the reward function (e.g., compiler error messages, before they are collapsed into scalar rewards). Because such serialized trajectories are readily understood by modern LLMs, we argue that *algorithms that learn deliberately in natural language by reflecting on these trajectories* can make more effective use of the strong language priors that LLMs have, compared with standard RL approaches.

We introduce GEPA (Genetic-Pareto), a *reflective* prompt optimizer for compound AI systems that merges textual reflection with multi-objective evolutionary search. GEPA iteratively mutates prompts using natural language feedback drawn from new rollouts. In each mutation, the candidate prompt is derived from an ancestor, accumulating high-level lessons derived from observations and LLM feedback. To avoid local optima that afflict greedy prompt optimization, GEPA maintains a Pareto front: instead of evolving only the global best prompt, it stochastically explores the top-performing prompts for each problem instance. This diversification enables robust generalization and mitigates getting stuck in local minima.

We evaluate GEPA across multi-hop reasoning (HotpotQA; Yang et al. 2018), Math (AIME, LiveBench-Math; Balunović et al. (2025); White et al. (2025)), instruction following (IFBench; Pyatkin et al. 2025b), privacy-aware delegation (PUPA; Li et al. 2025a), and retrieval-augmented verification (HoVer; Jiang et al. 2020), using both open (Qwen3 8B; Yang et al. 2025; Team 2025) and proprietary (GPT-4.1 Mini; OpenAI 2025) models. We find that GEPA generalizes well and is highly sample-efficient: on Qwen3 8B, it outperforms GRPO (24k rollouts) by up to 19pp while using up to $35\times$ fewer rollouts, with an average gain of +6pp across six tasks. GEPA also surpasses the prior state-of-the-art, MIPROv2 (Opsahl-Ong et al., 2024), on all benchmarks and models, achieving +13pp aggregate gains, over double MIPROv2's +5.6pp.

Qualitatively, GEPA-learned prompts are quite rich. Figure 2 shows excerpts from a prompt crafted for the query-creation module of a multi-hop question answering system used in HotpotQA, and Figure 26 shows that even a single *reflective* update often yields large gains. These results highlight that reflective prompt evolution with language feedback enables improved sample efficiency and robust generalization, offering a practical approach to optimizing complex AI workflows in data- or budget-constrained environments. We also demonstrate GEPA as an inference-time search strategy for code optimization on NPUEval (Kalade & Schelle, 2025) & KernelBench (Ouyang et al., 2025) in Sec E, and for adversarial prompt search in Sec F.

## 2 PROBLEM STATEMENT

We follow related work in defining a **compound AI system** as any modular system composed of one or more language model (LLM) invocations, potentially interleaved with external tool calls, orchestrated through arbitrary control flow. This definition subsumes a broad class of real-world LLM-based AI systems, including *agents*, *multi-agent systems*, and general-purpose scaffolding techniques like ReAct (Yao et al., 2023), Archon (Saad-Falcon et al., 2025), etc. Following Soylu et al. (2024); Khattab et al. (2024); Opsahl-Ong et al. (2024); Tan et al. (2025), we formalize such a system as $\Phi = (M, C, \mathcal{X}, \mathcal{Y})$, where $M = \langle M_1, \ldots, M_{|M|} \rangle$ denotes language modules, $C$ specifies control flow logic, and $\mathcal{X}, \mathcal{Y}$ are global input/output schemas. Each module $M_i = (\pi_i, \theta_i, \mathcal{X}_i, \mathcal{Y}_i)$ is an LLM subcomponent: $\pi_i$ is its (system) prompt including instructions and few-shot demonstrations; $\theta_i$ the underlying model weights; $\mathcal{X}_i, \mathcal{Y}_i$ are input/output schemas. At runtime, $C$ orchestrates the sequencing and invocation of modules—e.g., passing outputs from one module to another, in-

---

**Seed Prompt for Second-Hop of Multi-Hop QA System**

Given the fields `question`, `summary_1`, produce the fields `query`.

---

**GEPA's Optimized Prompt for Second-Hop of Multi-Hop QA System, GPT-4.1 Mini**

**You will be given two input fields: `question` and `summary_1`. Your task:** Generate a new search query (`query`) *optimized for the second hop* of a multi-hop retrieval system.

- The original user question is typically complex and requires information from multiple documents to answer.
- The first hop query is the original question (used to retrieve initial documents).
- Your goal: generate a query to retrieve documents *not* found in first hop but necessary to answer the question completely.

**Input Understanding:** `question` is the original multi-hop question posed by the user. `summary_1` is a concise summary of information from a document retrieved in the first hop, which partially addresses the question.

**Purpose and Context:**

- Your generated `query` aims to find the *missing pieces* of information needed to fully answer the question. . . .
- The query must retrieve relevant documents *NOT* found in first hop . . . for final answer extraction.

**Key Observations and Lessons:**

- First-hop documents often cover one entity or aspect.
- Remaining relevant documents often involve connected concepts mentioned in `summary_1` but not explicitly asked in the question. The `query` should target these *missing*, but linked, documents. Avoid merely paraphrasing the original question or restating known facts from `summary_1`.
- Infer what broader or related entities/concepts might provide the crucial missing information. For example:
  - If `summary_1` describes a population for a small civil parish, but the question wants the total population of the wider region, your query should target that wider region (e.g., "Madeira archipelago population in 2011").
  - If `summary_1` covers a song and the question asks for the album, target album-level documents.

**How to Build the Query:**

- Identify entities or topics mentioned in `summary_1` that are related but different from first-hop documents.
- Reframe the query to explicitly mention these broader or related entities *connected to the original question*.
- Include relevant key context from the question to maintain specificity, but shift focus to the missing piece.
- The goal is to retrieve documents that link or complement what was retrieved initially.

**Practical Strategy:**

- Read the `summary_1` carefully to spot references to bigger contexts or other entities not covered in the first hop.
- Ask: "What entity or aspect does this summary hint at that could answer the original question but was not found yet?"
- Formulate a precise, focused factual query targeting that entity or concept to retrieve the missing documents.

**Output:**

- Produce `query` as a clear, concise question or keyword phrase designed for efficient retrieval of second-hop documents.
- Ensure the query relates logically to the original question while targeting the broader or complementary knowledge identified in `summary_1`. . . . Do not include the original question or simply rephrase it. Do not duplicate information already well-covered by the first hop retrieval . . .

Figure 2: This figure shows an example prompt generated by GEPA for the second-hop document retrieval to be performed in a multi-hop question-answer system, along with the seed prompt it started with. Appendix N compares GEPA's prompts for all tasks with prompts generated by MIPROv2.

voking modules conditionally, or leveraging tool APIs. This way, $C$ can invoke different modules in any order multiples of times.

Given $\Phi$, let $\Pi_\Phi = \langle \pi_1, \ldots, \pi_{|M|} \rangle$ denote the collection of all module prompts and $\Theta_\Phi = \langle \theta_1, \ldots, \theta_{|M|} \rangle$ the set of module weights. The learnable parameters are thus $\langle \Pi, \Theta \rangle_\Phi$. For a task instance $(x, m)$—where $x$ maps to the input schema $\mathcal{X}$ and $m$ contains evaluator metadata (e.g., gold answers, evaluation rubrics, code unit tests)—the system induces an output $y = \Phi(x; \langle \Pi, \Theta \rangle_\Phi)$. A metric $\mu : \mathcal{Y} \times \mathcal{M} \to [0, 1]$ then measures the output quality of $y$ with respect to metadata $m$ (for example by calculating, exact match, F1, pass rate, etc.). The optimization problem is thus defined as follows, where $\mathcal{T}$ is a task distribution.:

$$\langle \Pi^*, \Theta^* \rangle_\Phi = \arg \max_{\langle \Pi, \Theta \rangle_\Phi} \mathbb{E}_{(x,m)\sim\mathcal{T}} \left[ \mu\big(\Phi(x; \langle \Pi, \Theta \rangle_\Phi), m\big) \right]. \tag{1}$$

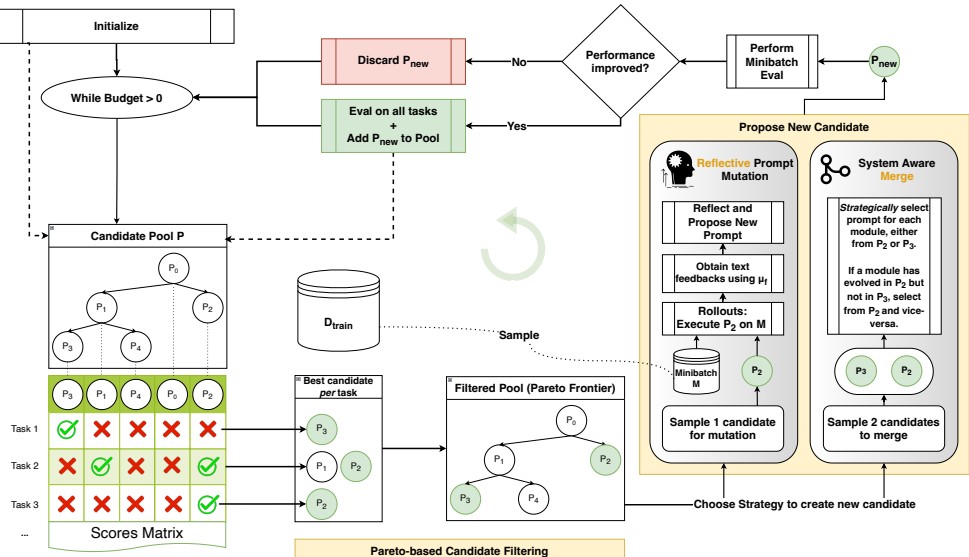

Figure 3: GEPA proposes a new candidate in every iteration by improving existing candidates using one of the two strategies (Reflective Prompt Mutation (Section 3) or System Aware Merge (Appendix D.1)), first evaluating them on a minibatch, and if improved, evaluating on a larger dataset. Instead of selecting the best performing candidate to mutate always, which can lead to a local-optimum, GEPA introduces Pareto-based candidate sampling (Section 3.1), which filters and samples from the list of best candidates *per* task, ensuring sufficient diversity. Overall, these design decisions allow GEPA to be highly sample-efficient while demonstrating strong generalization.

We adopt this general problem formulation, allowing updates to both prompts and weights of language modules, to enable comparisons between optimization algorithms that operate in different parameter spaces (e.g., GEPA vs. GRPO).

**Sample-Efficient Optimization.** In many real-world scenarios, rollouts—concretely, invocations of $\Phi$ plus evaluation by $\mu$—are often computationally, monetarily, or timewise expensive. The optimizer is thus limited to at most $B$ rollouts on a dataset $\mathcal{D}_{\text{train}} = \{(x, m)_i\}_{i=1}^N$ with full access to $\mu$. The goal is to identify parameters $\langle \Pi^*, \Theta^* \rangle_\Phi$ that maximize held-out performance, subject to not exceeding the rollout budget $B$:

$$\langle \Pi^*, \Theta^* \rangle_\Phi = \arg\max_{\langle \Pi, \Theta \rangle_\Phi} \mathbb{E}_{(x,m) \sim \mathcal{T}} \left[ \mu\big(\Phi(x; \langle \Pi, \Theta \rangle_\Phi), m\big) \right], \quad \text{s.t.} \quad \#\text{rollouts} \leq B. \quad (2)$$

The core challenge, then, is: *How do we extract maximal learning signal from every expensive rollout to enable effective adaptation of complex, modular AI systems in low-data or budget-constrained settings?*

## 3 GEPA: REFLECTIVE PROMPT EVOLUTION

We introduce GEPA (Genetic-Pareto), a sample-efficient optimizer for compound AI systems motivated by three core principles: genetic prompt evolution (Section 3), reflection using natural language feedback (Section 3), and Pareto-based candidate selection (Section 3.1). Figure 3 gives an overview of GEPA and the full GEPA algorithm is formalized in Figure 5. GEPA receives the following inputs: A system $\Phi$ instantiated with simple prompts to be optimized, training dataset $D_{train}$ (consisting of task instances $(x, m)$ as described in Section 2), the standard evaluation metric $\mu$ for the task, a feedback function $\mu_f$ (introduced in Section 3) and the total rollout budget $B$. Note that GEPA evolves only the set of prompts, denoted as $\Pi_\Phi$, whereas the underlying LLM weights, denoted by $\Theta_\Phi$ remains fixed.

**Genetic Optimization Loop:** Given an AI system $\Phi$, the goal is to identify parameters $\Pi_\Phi$ that maximize task performance. GEPA begins with a *candidate pool* $\mathcal{P}$ containing only the base system, where each *candidate* is a concrete instantiation of $\langle \Pi, \Theta_{frozen} \rangle_\Phi$. It then enters an optimization

loop, repeatedly proposing new candidates until the evaluation budget is exhausted. Candidates are derived from existing ones via *reflective mutation* or *crossover*, guided by feedback from rollouts, with each inheriting learning signals from its parents and its own rollout so that GEPA accumulates knowledge along the genetic tree. In each iteration, GEPA (i) selects promising candidates, (ii) proposes and evaluates a variant on a minibatch of tasks, and (iii) if it outperforms its parent(s), adds it to $\mathcal{P}$ with ancestry records and evaluate on $D_{pareto}$, the validation set used for selection. After the budget is exhausted, GEPA returns the candidate with the best aggregate performance on $D_{pareto}$.

**Reflective Prompt Mutation:** Natural language traces generated during the execution of a compound AI system offer rich *visibility* into the behavior and responsibilities of each module, as they capture the intermediate inferences and underlying reasoning steps. When these traces are paired with the final outcome of the system (e.g., success or failure), they provide substantial *diagnostic* value, allowing practitioners to trace errors or successes back to specific decisions made at the module level. LLMs can leverage these traces via *reflection* to perform implicit credit assignment, attributing responsibility for the final outcome to the relevant modules. This process of *reflection* can then be used to make targeted updates to individual modules, making large and effective updates to the whole system's behavior.

Given a *candidate* to mutate in the current iteration of the optimization loop (stochastically selected from the Pareto-frontier, see Section 3.1 below), GEPA executes the selected candidate on a stochastically sampled minibatch of input queries from the trainset, tracing the program's execution. From the execution traces, GEPA extracts the module's inputs, outputs, and reasoning, and calls the *feedback function* $\mu_f$, which returns a numeric score and text feedback including details about the evaluation (like compiler error messages, failed rubrics, etc.). GEPA selects the module (among the $|M|$ modules that the language program contains) to be updated based on a policy (round-robin), and a reflection LM is then shown the (current prompt, language program trajectory, score, feedback) with the task to reflectively attribute successes or failures to prompt elements and propose revised instructions. The updated module, with the rest of the language program, is evaluated again on the minibatch, and if the score improves, then the new program is added to the candidate pool. The meta-prompt for reflective prompt updates is shown in Appendix C and the full algorithm is presented in Algorithm 1.

**Evaluation traces as diagnostic signals:** The text that LLMs produce is the *execution trace* of the AI system. The text that the environment produces to compute the reward (e.g. compiler error messages before giving reward 0) is the *evaluation trace*. Beyond reflection on execution traces, we identify a second valuable source of diagnostic information in the evaluation traces. Many evaluation metrics apply rich strategies (e.g., code evaluation may involve compilation, execution, and profiling), producing natural language traces before computing a scalar reward. We propose leveraging these *evaluation traces* for reflective credit assignment and targeted prompt updates. GEPA achieves this by extending rewards $\mu$ into a *feedback function* $\mu_f$ that extracts textual traces during evaluation and returns them with the final score as `feedback_text`. When available, such feedback can even be module-specific (e.g. in multi-hop systems the evaluator may provide feedback after each hop). In practice, there are domains where human-graders are able to rate the AI system's responses, along with providing detailed feedback justifying their scalar ratings. When available, $D_{train}$ can be augmented with such human-written explanations for each instance; during reflection, and GEPA can consume these explanations as auxiliary `feedback_text` to guide targeted prompt updates, even when natural-language feedback from rollouts is limited or unavailable.

## 3.1 PARETO-BASED CANDIDATE SELECTION

GEPA is a highly modular algorithm that supports various strategies for candidate selection, with the choice of strategy governing the exploration–exploitation tradeoff. A naive approach is to always select the best-performing candidate, but this often traps the optimizer in a local optimum: once a dominant strategy is found, it becomes difficult to surpass, and the optimizer exhausts its budget without learning new, potentially better strategies. Figure 3a illustrates this behavior: after finding one new strategy (the first child node), the search repeatedly attempts to refine it, fails to improve, and ultimately depletes the budget.

To address this, GEPA employs a Pareto-based "illumination" strategy (Mouret & Clune, 2015), shown in Algorithm 2. For each training instance, GEPA records the highest score across all can-

didates, forming a Pareto frontier. Candidates that achieve the best score on at least one task are retained, while strictly dominated ones are pruned. From this pruned set, GEPA stochastically samples a candidate, weighting probabilities by how many tasks each candidate leads. This strategy helps GEPA escape local optima without inflating the search, efficiently balancing exploration and exploitation by focusing resources on candidates that embody "winning" strategies within the optimization budget.

## 4 EVALUATION

We adopt a standard train/validation/test split. Optimizers have full access to the train split, including text and labels, for program tuning. Although optimizers may monitor the performance of candidate parameters (like model checkpoints) by tracking scores on the validation set (to implement early stopping, for example), direct access to the content of validation instances is restricted. We evaluate on six benchmarks—AIME-2025 (Balunović et al., 2025), LiveBench-Math (White et al., 2025), HotpotQA (Yang et al., 2018), IFBench (Pyatkin et al., 2025b), HoVer (Jiang et al., 2020), and PUPA (Li et al., 2025a)—each paired with existing compound AI systems and feedback functions. Experiments use Qwen3 8B (Yang et al., 2025) and GPT-4.1 Mini (OpenAI, 2025) with standardized inference settings, and compare against state-of-the-art optimizers MIPROv2 (Opsahl-Ong et al., 2024), Trace (with its OptoPrime optimizer) (Cheng et al., 2024), TextGrad (Yuksek-gonul et al., 2025), and GRPO[1] (Shao et al., 2024). Appendix G provides further details on benchmarks, systems, and feedback functions (Subsection G.1); models and inference settings (Subsection G.2); monetary cost to run the experiments (Subsection G.3); and optimizer configurations (Subsection G.4). Table 1, Table 2 and Figure 9 summarize our main results, from which we derive the following observations:

Table 1: Benchmark results for different optimizers with Qwen3 8B. GEPA and GEPA+Merge achieve better performance than GRPO with far fewer rollouts on all benchmarks except AIME. For example, for IFBench, GEPA found optimal prompts after just 678 rollouts achieving 38.61%, outperforming GRPO's test set score of 35.88% with 24,000 rollouts.

| Qwen3 8B | HotpotQA | IFBench | Hover | PUPA | AIME-2025 | LiveBench-Math | Aggregate | Improvement |
|---|---|---|---|---|---|---|---|---|
| Baseline | 42.33 | 36.90 | 35.33 | 80.82 | 27.33 | 48.70 | 45.23 | — |
| GRPO | 43.33 | 35.88 | 38.67 | 86.66 | **38.00** | 51.26 | 48.91 | +3.68 |
| MIPROv2 | 55.33 | 36.22 | 47.33 | 81.55 | 20.00 | 46.60 | 47.84 | +2.61 |
| GEPA | 62.33 | **38.61** | **52.33** | **91.85** | 32.00 | **51.95** | **54.85** | **+9.62** |
| GEPA+Merge | **64.33** | 28.23 | 51.67 | 86.26 | 32.00 | **51.95** | 52.40 | +7.17 |
| Total optimization budget (# rollouts) | | | | | | | | |
| GEPA (+Merge) | 6871 | 3593 | 7051 | 2426 | 1839 | 1839 | 3936 | — |
| GRPO | 24000 | 24000 | 24000 | 24000 | 24000 | 24000 | 24000 | — |

**Observation 1: Reflective Prompt Evolution is highly sample-efficient and can outperform weight-space reinforcement learning:** Across six benchmarks, GEPA adapts rapidly and generalizes robustly in compound AI systems—beating GRPO (24,000 rollouts) by up to 19 percentage points while using up to $35\times$ fewer rollouts. It reaches optimal test performance with $4$–$35\times$ fewer rollouts and exceeds GRPO on 5 out of 6 tasks by 19.0, 2.73, 13.66, 5.19, and 0.7 percentage points. GEPA matches GRPO's best validation after only 243, 402, 330, 1143, 1179, and 306 rollouts—up to $78\times$ greater sample efficiency. GEPA+Merge widens the gap, outperforming GRPO by 21 percentage points at a comparable rollout budget to GEPA.

The majority of GEPA's rollout budget is spent on validation, where scores are utilized solely for candidate selection and not for producing learning signals. If we restrict the analysis to train set rollouts, GEPA requires only 79 to 737 rollouts to reach optimal performance. To match GRPO's best validation scores, GEPA achieves this with only 102, 32, 6, and 179 train rollouts for four tasks, respectively, underscoring the high sample efficiency of learning based on reflective prompt evolution.

Since tracking candidates' validation performance accounts for majority of GEPA's rollout budget, sample efficiency can be further improved by evaluating on a smaller validation set or by tracking

---

[1]We use LoRA for GRPO due to its low cost and succesful adoption with GRPO (Wang et al., 2025b; Xu et al., 2025b; Li et al., 2025b; Yue et al., 2025; Sun et al., 2025; Hayou et al., 2025; Zhao et al., 2025; Teknium et al., 2024; Zhao et al., 2024; Sidahmed et al., 2024). Additionally, we explore full-parameter finetuning. Figure 10 shows a similar result comparing GEPA to GRPO with full finetuning.

Table 2: Benchmark results for different optimizers evaluated on GPT-4.1 Mini. As a prompt-optimization system, GEPA works off-the-shelf on *closed-source* models as well, outperforming state-of-the-art prompt optimizers including MIPROv2 (in 2 settings: Instruction-only optimization ("MIPROv2-No-Demos") as well as joint instruction and few-shot optimization ("MIPROv2")), Trace (with its OptoPrime optimizer), and TextGrad. Additionally, GEPA-optimized prompts demonstrate strong cross-model generalization: "GEPA-Qwen-Opt"—optimized entirely for (and using) the weaker Qwen3-8B—achieves a +9% gain when evaluated on GPT-4.1-Mini without modification, notably outperforming all baselines (MIPROv2, TextGrad, Trace) that optimized directly for (and using) GPT-4.1-Mini.

| GPT-4.1 Mini | HotpotQA | IFBench | Hover | PUPA | AIME-2025 | LiveBench-Math | Aggregate | Improvement |
|---|---|---|---|---|---|---|---|---|
| Baseline | 38.00 | 47.79 | 46.33 | 78.57 | 49.33 | 58.20 | 53.03 | — |
| Trace (OptoPrime) | 60.33 | 51.19 | 46.00 | 74.18 | 45.33 | 60.74 | 56.30 | +3.27 |
| MIPROv2-No-Demos | 38.00 | 52.04 | 51.33 | 91.85 | 48.67 | 60.97 | 57.14 | +4.11 |
| MIPROv2 | 58.00 | 49.15 | 48.33 | 83.37 | 51.33 | 61.84 | 58.67 | +5.64 |
| TextGrad | 62.33 | 48.64 | 47.67 | 85.68 | 46.67 | 63.84 | 59.14 | +6.11 |
| GEPA | **69.00** | 52.72 | 51.67 | 94.47 | **59.33** | **64.13** | 65.22 | +12.19 |
| **GEPA+Merge** | 65.67 | **55.95** | **56.67** | **96.46** | **59.33** | **64.13** | **66.36** | **+13.33** |
| Optimized with Qwen3-8B, evaluated on GPT-4.1-Mini | | | | | | | | |
| GEPA-Qwen-Opt | 65.67 | 49.83 | 54.67 | 90.05 | 52.67 | 59.31 | 62.03 | +9.00 |

scores on dynamically selected validation subsets instead of the full set—both of which we propose as directions for future work. Figures 1a, 1b, 13c and 14c show the full performance-vs-rollouts curve for all optimizers over benchmarks HotpotQA, IFBench, HoVer and PUPA, respectively.

**Observation 2: Reflective prompt evolution enables instruction-optimization alone to outperform joint instruction and few-shot optimization:** We compare GEPA with MIPROv2, a state-of-the-art instruction and few-shot optimizer, using two leading models across six diverse tasks, and observe that GEPA consistently outperforms MIPROv2 in all settings, achieving margins as high as 11.1pp for GPT-4.1 mini and 12pp for Qwen3 8B. Further, GEPA and GEPA+Merge more than double the aggregate gains over baseline seen with MIPROv2 across all benchmarks and models (+13.33pp and +12.19pp vs +5.64pp for MIPROv2).

While prior works such as Opsahl-Ong et al. (2024) and Wan et al. (2024) have provided compelling evidence for the effectiveness of few-shot example optimization—often outperforming instruction-based approaches—our findings suggest an exciting shift in this trend. We attribute this primarily to recent advances in the instruction-following and self-reflective abilities of LLMs, as well as the design choices in GEPA that capitalize on these improved capabilities. To further contextualize our findings, we redo the study on *generalization gap* (the difference between validation and test set performance for optimized prompts) as proposed by Wan et al. (2024). The results presented in Figure 15 reinforce these observations: reflectively evolved instructions now demonstrate a lower generalization gap, underscoring both advancements in model capabilities and the benefits of GEPA's design. We see this as a reflection of the continuous evolution of LLMs and GEPA's ability to effectively leverage these improvements.

We provide the full-length optimized prompts produced by GEPA for all systems, benchmarks, and models in Appendix N, alongside MIPROv2 prompts. Notably, in contrast to prior findings where instruction optimization yielded improvements primarily through quasi-exemplars (Wan et al., 2024), GEPA's prompts frequently contain detailed *declarative* instructions for completing the task, as illustrated in Figure 2.

**Observation 3: The next-candidate selection strategy strongly influences the optimization trajectory and final performance, with Pareto-based sampling providing a distinct advantage.** GEPA refines prompts iteratively with rollout feedback; to test our Pareto-based selection, we compare against a baseline that always picks the best-performing candidate in the `SelectBestCandidate` strategy (which is similar to the strategy used by TextGrad Yuksekgonul et al. (2025)), and `BeamSearch(N=4)` (used by APO Pryzant et al. (2023)). As shown in Table 3, these baselines often yield suboptimal exploration of the prompt search space, leading to poor performance. GEPA with Pareto-based sampling outperforms the `BeamSearch` strategy by upto 11.33pp, and `SelectBestCandidate` strategy by up to 8.17pp, with an aggregate margin of +7.33pp and +6.4pp across all benchmarks, respectively. Figure 4 highlights the difference in optimization trajectories: always choosing the current best candidate gives immediate improvement but quickly

stalls, wasting rollouts on a single candidate. In contrast, our Pareto-based method expands the search by considering all Pareto-optimal candidates (all "winning" strategies found so far), balancing exploration and exploitation and converging to a higher-performing solution within the same rollout budget.

Table 3: Comparing candidate selection strategies across different tasks with Qwen3 8B while keeping the evolution harness fixed. At each step, `SelectBestCandidate` (used by TextGrad Yuksekgonul et al. (2025)) evolves only from the top-scoring candidate. `BeamSearch` maintains a pool of the top-N candidates (used by APO Pryzant et al. (2023)), but is still prone to local optima. In comparison, GEPA's Pareto-based selection yields a +12.44% improvement, significantly outperforming the +6.05% and +5.11% gains of greedy and beam-search strategies respectively.

| Qwen3 8B | HotpotQA | IFBench | Hover | PUPA | Aggregate | Improvement |
|---|---|---|---|---|---|---|
| Baseline | 42.33 | 36.90 | 35.33 | 80.82 | 48.84 | — |
| SelectBestCandidate | 58.33 | 30.44 | 45.33 | 85.45 | 54.89 | +6.05 |
| BeamSearch | 57.33 | 36.39 | 41.00 | 81.08 | 53.95 | +5.11 |
| GEPA | **62.33** | **38.61** | **52.33** | **91.85** | **61.28** | **+12.44** |

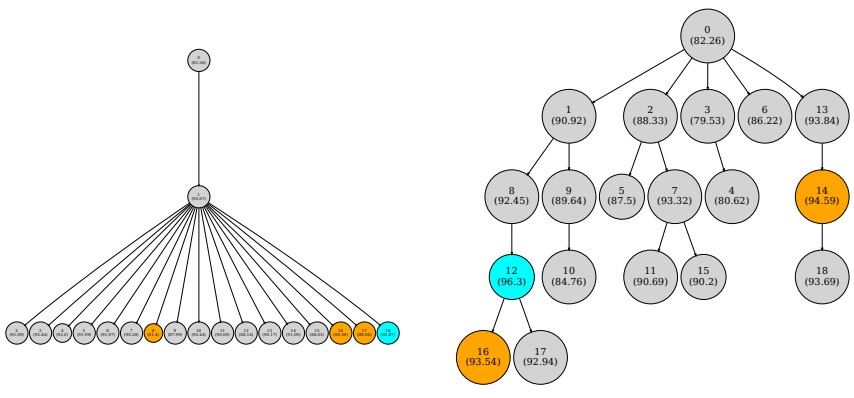

(a) SelectBestCandidate Strategy      (b) Pareto-based candidate sampling.

Figure 4: Comparing the impact of different candidate selection strategies. (Left) As can be seen, selecting the best-performing candidate in every iteration led to a local-optima after one iteration, leading to suboptimal search performance. (Right) On the other hand, using pareto-based candidate selection strategy, GEPA was able to generate a balanced search tree, finding a better performing program within the same budget.

We also find that **instruction-optimized prompts are more efficient and generalize better than few-shot demonstrations (Observation 4)**. Further, **System-aware crossover strategies such as Merge can improve performance, but their effectiveness depends on budget allocation between mutation and crossover and on timing (Observation 5)**. Finally, **GEPA-optimized prompts demonstrate cross-model generalization (Observation 6)**. See Appendix H for further discussion.

## 5 EXTENDED APPLICATIONS OF GEPA

**GEPA For Inference-Time Search** GEPA can be used as an inference-time search technique by setting the validation set equal to the training set ($D_{val} = D_{train}$), under which GEPA iteratively proposes better solutions for each target task. We evaluate this approach on hardware-specific code generation: writing AMD NPU kernels (NPUEval (Kalade & Schelle, 2025)) and generating NVIDIA GPU's CUDA kernels (KernelBench (Ouyang et al., 2025)) using GPT-4o OpenAI (2024). GEPA raises mean kernel vector utilization on NPUEval from 4.25% (sequential refinement baseline) to 30.52%, and increases the fraction of KernelBench tasks with CUDA code surpassing baseline-PyTorch speed from close to 0% to over 20%. Full experimental details are provided in Appendix E.

**GEPA for Adversarial Prompt Search**    GEPA can discover adversarial prompts by inverting the reward—finding instructions that minimize task performance. On AIME-2025 with GPT-5 Mini, starting from a simple instruction, GEPA evolved a trivia-style distractor that reduced pass@1 from 76% to 10%. Adversarial search drew on AIME 2022–2024 problems, with evaluation on AIME-2025 using 5 runs per problem (30 problems; 150 generations). This robustness probe surfaces brittle instruction-following interactions (e.g., trivia with strict formatting) and yields reusable stress tests and regression suites for deployment and show that reflective prompt evolution is also effective for stress-testing them by locating universal, query-agnostic perturbations that sharply degrade accuracy. These findings parallel recent results on query-agnostic adversarial triggers, which report two- to seven-fold increases in error rates across model families; in our AIME-2025 setting, GEPA's learned universal instruction yields a comparable $3.8\times$ increase (errors from 24% to 90%) (Rajeev et al., 2025). Full experimental details are provided in Appendix F.

## 6    RELATED WORK

**Prompt optimization** improves LLMs but often needs manual expertise; for instance, chain-of-thought prompting Wei et al. (2023). To scale this approach, recent methods use LLMs to optimize prompts automatically (Zhou et al., 2022; Yang et al., 2024; Agarwal et al., 2024; Fernando et al., 2024). GEPA leverages LLMs, but differs by incorporating textual environment feedback, Pareto-aware search over candidates, and evolution strategies per submodule within an AI system.

**Evolutionary algorithms** have been used to optimize prompts, e.g., EvoPrompt (Guo et al., 2024), which evolves prompt populations. Rainbow Teaming (Samvelyan et al., 2024) applies quality-diversity evolution to generate diverse adversarial prompts. GEPA additionally uses domain-specific feedback for targeted mutations achieving higher sample efficiency. AlphaEvolve (Novikov et al., 2025) and OpenEvolve (Sharma, 2025) apply evolutionary search directly to code rewriting, excelling when problem solution can be codified. While AlphaEvolve targets a single hard problem, GEPA brings evolution to prompts across domains, combining Pareto-frontier optimization and prompt evolution to transfer tactics from related problems.

**Feedback-driven improvement** often uses reinforcement learning, such as majority voting signals (Zuo et al., 2025), but it can be sample-inefficient when rewards are slow to compute. An alternative is learning in the language space: in-context bandit/self-bootstrapping (Shinn et al., 2023; Madaan et al., 2023) (Monea et al., 2025; Xu et al., 2025a; Feng et al., 2025; Cheng et al., 2024), workflow memory and skills (Wang et al., 2024; 2025c), and test-time strategy synthesis via Dynamic Cheatsheet (Suzgun et al., 2025), reasoning cache (Chen et al., 2025c). GEPA instead uses examples to propose new *instructions*, yielding task-specific rules.

To **optimize compound AI systems and agents** (Lin et al., 2025b), DSPy (Khattab et al., 2022; 2024) searches/bootstraps few-shot examples, TextGrad (Yuksekgonul et al., 2025) backpropagates textual feedback, and MIPROv2 (Opsahl-Ong et al., 2024) jointly aligns instructions and examples via Bayesian optimization; these largely rely on global rewards. Agent-Pro (Zhang et al., 2024) evolves agent policies through dynamic belief generation and reflection on interactive experiences. Optimas (Wu et al., 2025a) introduces globally aligned local rewards per module. GEPA combines global rewards with environment textual feedback per module and maintains a Pareto frontier over individual data instances, matching prompts/agent design to specific examples. The Pareto-guided evolution lets GEPA explore diverse prompt/code/agent design strategies before converging to a robust, generalizable set.

## 7    CONCLUSION

We introduced GEPA, a prompt optimizer for arbitrary LLM agents and workflows that leverages explicit reflection and Pareto-based selection, showing superior sample efficiency compared to reinforcement learning (GRPO), while outperforming leading prompt optimizers (MIPROv2). By explicitly incorporating natural language feedback and maintaining a diverse pool of Pareto-optimal candidates, GEPA rapidly adapts AI systems to new tasks. Our results across benchmarks and models suggest that language-based reflection can offer a scalable strategy for optimizing complex real-world AI workflows, especially in resource-constrained settings. GEPA also shows promise as an inference-time search strategy, showing the ability to write code in challenging domains.

ACKNOWLEDGMENTS

This research is supported in part by gifts from Accenture, Amazon, AMD, Anyscale, Broadcom, Google, IBM, Intel, Intesa Sanpaolo, Lambda, Lightspeed, Mibura, NVIDIA, Samsung SDS, SAP, by the U.S. Department of Energy, Office of Science, Office of Advanced Scientific Computing Research through the X-STACK: Programming Environments for Scientific Computing program (DESC0021982,) and the Defense Advanced Research Projects Agency (DARPA) under Agreement No. HR00112590134. Lakshya A Agrawal is supported by a Laude Slingshot grant provided by the Laude Institute and an Amazon AI PhD Fellowship. We are grateful to the AMD University Program for provisioning AMD NPU Mini PCs, and to Dr. Graham Schelle and Dr. Sarunas Kalade for their support during development. Finally, we thank Dr. Nicholas Tomlin and Prof. Joseph E. Gonzalez for their valuable comments on this manuscript.

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

## A    APPENDIX OUTLINE

- Usage of Large Language Models
- GEPA's Reflection and Prompt Update Meta Prompt
- GEPA Algorithm and Methodology Details
- GEPA For Inference-Time Search (Contd.)
- GEPA for Adversarial Prompt Search (Contd.)
- Evaluation Setup (Contd.)
- Results and Analysis (Contd.)
- Performance vs. Budget (Rollouts) Curves
- Generalization Gap
- Cost vs. Performance Analysis for optimized systems
- GEPA Search Trees
- Visualizing the Iterative Refinement achieved by GEPA
- Examples of best prompts for every benchmark
- GEPA generated prompts for kernel generation
- Number of reflection LM calls made by GEPA during optimization

## B    USAGE OF LARGE LANGUAGE MODELS

The authors used large language models (LLMs) only for polishing prose of text where the complete draft was fully written by the authors initially and polished later with the help of LLM-based assistants including ChatGPT, Gemini, and Perplexity. The authors' used code assistants including Cursor and Copilot to implement the authors' original design and ideas. The scientific contributions, technical methods, ideas and core results are entirely the original work of the authors.

## C    GEPA'S REFLECTION AND PROMPT UPDATE META PROMPT

**GEPA's Meta Prompt**

```
I provided an assistant with the following instructions to perform a task for me:

```
<current instruction>
```

The following are examples of different task inputs provided to the assistant
along with the assistant's response for each of them, and some feedback on how the
assistant's response could be better:

```
<Inputs, Outputs and Feedback for minibatch of examples>
```

Your task is to write a new instruction for the assistant.

Read the inputs carefully and identify the input format and infer detailed task
description about the task I wish to solve with the assistant.

Read all the assistant responses and the corresponding feedback. Identify all
niche and domain specific factual information about the task and include it in the
instruction, as a lot of it may not be available to the assistant in the future.
The assistant may have utilized a generalizable strategy to solve the task, if so,
include that in the instruction as well.
```

**Algorithm 1** GEPA: Reflective Evolutionary Prompt Optimizer

**Require:** Inputs: System $\Phi$, dataset $\mathcal{D}_{\text{train}}$, eval metric $\mu$, feedback function $\mu_f$, budget $B$
**Require:** Hyperparams: minibatch size $b$, Pareto set size $n_{pareto}$
1: Split $\mathcal{D}_{\text{train}}$ into $\mathcal{D}_{\text{feedback}}$, $\mathcal{D}_{\text{pareto}}$, s.t. $|D_{pareto}| = n_{pareto}$
2: Initialize candidates $\mathcal{P} \leftarrow [\Phi]$, parents $\mathcal{A} \leftarrow [\text{None}]$
3: **for** each $(x_i, m_i)$ in $\mathcal{D}_{\text{pareto}}$ **do**
4:     $S_\Phi[i] \leftarrow \mu(\Phi(x_i), m_i)$
5: **end for**
6: **while** budget $B$ not exhausted **do**
7:     $k \leftarrow \text{SELECTCANDIDATE}(\mathcal{P}, S)$
8:     $j \leftarrow \text{SELECTMODULE}(\Phi_k)$
9:     $\mathcal{M} \leftarrow$ minibatch of size $b$ from $\mathcal{D}_{\text{feedback}}$
10:     Gather feedback, scores, traces for $\Phi_k[j]$ on $\mathcal{M}$ using $\mu_f$
11:     $\pi'_j \leftarrow \text{UPDATEPROMPT}(\pi_j, \text{feedbacks}, \text{traces}[j])$
12:     $\Phi' \leftarrow$ Copy of $\Phi_k$ w/ module $j$ updated by $\pi'_j$
13:     $\sigma, \sigma' \leftarrow$ avg score on $\mathcal{M}$ (before, after)
14:     **if** $\sigma'$ improved **then**
15:         Add $\Phi'$ to $\mathcal{P}$; Add $k$ to $\mathcal{A}$
16:         **for** each $(x_i, m_i)$ in $\mathcal{D}_{\text{pareto}}$ **do**
17:             $S_{\Phi'}[i] \leftarrow \mu(\Phi'(x_i), m_i)$
18:         **end for**
19:     **end if**
20: **end while**
21: **return** $\Phi^*$ maximizing average score on $\mathcal{D}_{\text{pareto}}$

**Algorithm 2** Pareto-based candidate selection

1: **function** SELECTCANDIDATE($\mathcal{P}, S$)
2:     // Build instance-wise Pareto sets
3:     **for** each $i$ **do**
4:         $s^*[i] \leftarrow \max_k S_{\mathcal{P}[k]}[i]$
5:         $\mathcal{P}^*[i] \leftarrow \{\mathcal{P}[k] : S_{\mathcal{P}[k]}[i] = s^*[i]\}$
6:     **end for**
7:     $\mathcal{C} \leftarrow$ unique candidates in $\bigcup_i \mathcal{P}^*[i]$
8:     $D \leftarrow \emptyset$
9:     **while** there exists $\Phi \in \mathcal{C} \setminus D$ dominated by another in $\mathcal{C} \setminus D$ **do**
10:         $D \leftarrow D \cup \{\Phi\}$
11:     **end while**
12:     Remove $D$ from each $\mathcal{P}^*[i]$ to get $\hat{\mathcal{P}}^*[i]$
13:     Let $f[\Phi] =$ number of $i$ for which $\Phi \in \hat{\mathcal{P}}^*[i]$
14:     Sample $\Phi_k$ from $\hat{\mathcal{C}}$ with probability $\propto f[\Phi_k]$
15:     **return** index $k$ of $\Phi_k$ in $\mathcal{P}$
16: **end function**

Figure 5: (**Left**) GEPA's core algorithm for reflective prompt evolution. GEPA works iteratively, in each iteration, selecting some of the current candidates to evolve (line 7), executing the identified candidate on a minibatch of rollouts, while utilizing a special *feedback function* $\mu_f$ to gather module specific feedback when available (lines 9-10, described in detail in Section 3), using an LLM to reflectively update the prompt (line 11), and evaluating whether the system instantiated with the new prompt improved the performance on the minibatch (line 14). If improved, GEPA then proceeds to evaluate the new system candidate on the full $D_{pareto}$ set, adding it to the list of candidates tracked and marking the new system's parent. (**Right**) The SelectCandidate subprocedure used by GEPA's core algorithm is tasked with identifying the best candidate to evolve in the next optimization iteration. GEPA's chief candidate selection strategy is to find non-dominated candidates in the Pareto frontier (of all task instances), and stochastically select one of them based on their appearance frequency in the Pareto front.

```
Provide the new instructions within ``` blocks.
```

Figure C shows the meta-prompt used by GEPA, which guides the LLM to reflectively refine its current instruction based on input–output examples and corresponding feedback from the environment.

# D   GEPA ALGORITHM AND METHODOLOGY DETAILS

Figure 5 presents the core GEPA Algorithm, along with the algorithm for Pareto-based candidate selection.

**Algorithm 3** Check if module combination is desirable

```
 1: function DESIRABLE(a, i, j, P)
 2:     for module m = 1 to |M| do
 3:         π_a ← ancestor's prompt for module m
 4:         π_i ← descendent i's prompt for module m
 5:         π_j ← descendent j's prompt for module m
 6:         if (π_a = π_i and π_j ≠ π_i) or(π_a = π_j and π_i ≠ π_j) then
 7:             return True
 8:         end if
 9:     end for
10:     return False
11: end function
```

**Algorithm 4** MERGE: Genetic Crossover for Modular Candidates

```
 1: function MERGE(P, A, S, r)
 2:     i, j ← r.sample(2, |P|)   // distinct i ≠ j
 3:     A_i ← GETANCESTORS(i, A), A_j ← GETANCESTORS(j, A)
 4:     if i ∈ A_j or j ∈ A_i then
 5:         continue    // skip direct ancestry
 6:     end if
 7:     for a ∈ A_i ∩ A_j do
 8:         if this merge (i, j, a) has been tried before then
 9:             continue
10:         end if
11:         if S[a] > min(S[i], S[j]) then
12:             continue
13:         end if
14:         if not DESIRABLE(a, i, j, P) then
15:             continue
16:         end if
17:         Φ' ← copy of P[a]
18:         for module m = 1 to |M| do
19:             π_a ← P[a].M_m.π
20:             π_i ← P[i].M_m.π
21:             π_j ← P[j].M_m.π
22:             if π_a = π_i and π_j ≠ π_i then
23:                 Φ'.M_m.π ← π_j
24:             else if π_a = π_j and π_i ≠ π_j then
25:                 Φ'.M_m.π ← π_i
26:             else if π_i ≠ π_j ≠ π_a then
27:                 Choose d* = arg max{S[i], S[j]} (break ties randomly)
28:                 Φ'.M_m.π ← π_{d*}
29:             else
30:                 Φ'.M_m.π ← π_i // default
31:             end if
32:         end for
33:         return (Φ', i, j, a)
34:     end for
35:     return None
36: end function
```

Figure 6: Details of System Aware Merge. $r$ represents a seeded stochastic sampler.

### D.1 MERGE: SYSTEM-AWARE CROSSOVER STRATEGY FOR COMPOUND AI OPTIMIZATION

Algorithm 4 provides the instantiation of the System aware Merge strategy used in GEPA+Merge. Intuitively, merge will be helpful when there are candidates in the pool that learn complementary strategies. Algorithm 3 defines the selection criteria: candidates are merged only if they share a common ancestor but have optimized disjoint sets of prompts (complementary strategies), are pareto-optimal, and both candidates improve upon the aggregate performance of the ancestor. GEPA routinely checks if the pool has 2 such candidates, invoking merge when identified. These strict lineage conditions mean merge occurs sparsely.

## E    GEPA FOR INFERENCE-TIME SEARCH (CONTD.)

While the primary focus of this paper is sample-efficient adaptation of AI systems to new tasks, preliminary findings suggest that GEPA may also serve as a promising inference-time search technique. This can be achieved by passing the set of tasks to be solved (for example, a list of Pytorch mod-

ules to be converted to CUDA) as the training set to GEPA, ensuring that both $D_{train}$ and $D_{pareto}$ contain the full set of tasks. This way, GEPA can "overfit" the set of tasks, iteratively proposing better solutions to every problem. We also note that this allows GEPA to apply lessons and insights extracted from rollouts for one task to other tasks. To explore this use case, we conduct preliminary experiments using GEPA as an inference-time search technique for code-generation tasks on two hardware platforms: writing kernels for AMD's recently introduced XDNA2 Architecture (Advanced Micro Devices, 2025) using an early version of the NPUEval benchmark (Kalade & Schelle, 2025), and generating CUDA code for NVIDIA-V100 GPUs using KernelBench (Ouyang et al., 2025).

A distinguishing aspect of these experiments is the use of the feedback function $\mu_f$ to dynamically inject domain-specific knowledge into the optimization process. Specifically, kernel development expertise—often codified in technical manuals and documentation—can be selectively surfaced by retrieving relevant manual sections based on rollout failures (e.g., compiler error messages). By using error information to make targeted retrieval queries, GEPA promotes integration of architectural best practices into prompt evolution, as exemplified by the detailed prompt for NPUEval shown in Figure 27. We also note that generation stochasticity (temperature based sampling) is eliminated by operating under a cache; this ensures that observed improvements tie closely to inference scaling through prompt updates and GEPA's diverse prompt exploration, rather than stochasticity in the model's sampling process.

**NPU Kernels:** We create a sequential refinement agent that iteratively generates kernels (up to 10 times) based on feedback like compiler errors and profiling results (Sequential10), and evaluate the Best-of-N generation. With GPT-4o alone, Sequential10 reaches only 4.25% mean vector utilization. Adding RAG, sourced from technical manuals, improves this to 16.33%, and integrating MIPROv2 further raises it to 19.03%. Notably, applying GEPA to Sequential10 (without RAG) dramatically boosts kernel performance, with several generated kernels achieving up to 70% vector utilization and a mean of 30.52%. Furthermore, a single prompt generated by GEPA enables Sequential10 (again without RAG) to attain a score of 26.85%.

**CUDA Kernels:** For 35 tasks from the KernelBench "representative subset" (Ouyang et al., 2025), spanning three difficulty levels, we ran GEPA with GPT-4o. As depicted in Figure 8, GEPA boosts GPT-4o's close-to-0% $fast_1$ score to above 20% with increasing search budget. This task used an agent that could generate upto 5 sequential refinements based on environment feedback (Sequential5).

These experiments with GPT-4o also demonstrate GEPA's ability to leverage the abilities of frontier LLMs. However, these are early results and warrant further systematic study. We believe that leveraging GEPA for inference-time search, particularly when coupled with domain specific textual feedback, could generalize to other code generation and domain adaptation tasks—a direction we leave for future work.

## F  GEPA FOR ADVERSARIAL PROMPT SEARCH (CONTD.)

We instantiate GEPA for adversarial prompt search by inverting the reward signal: the optimizer proposes prompt edits to include additional information like trivia that minimize task performance (pass@1), while requiring that prompts do not contradict the task and still contain all information needed to solve it. For AIME, GEPA's adversarial search used AIME 2022–2024 problems as the pool for prompt evolution. The learned prompt was evaluated on AIME-2025 (30 problems), using GPT-5 Mini with 5 runs per problem (150 generations total). We started from a clean instruction prompt and evolved a single universal adversarial instruction that is prepended to each query.

The learned adversarial instruction reduced pass@1 on AIME-2025 from 76% (clean prompt) to 10% (adversarial prompt), despite preserving the nominal task description and the required answer-format directive.

Base (clean) prompt:

> You are a helpful assistant. You are given a question and you need to answer
> it. The answer should be given at the end of your response in exactly the format
> '### <final answer>'.

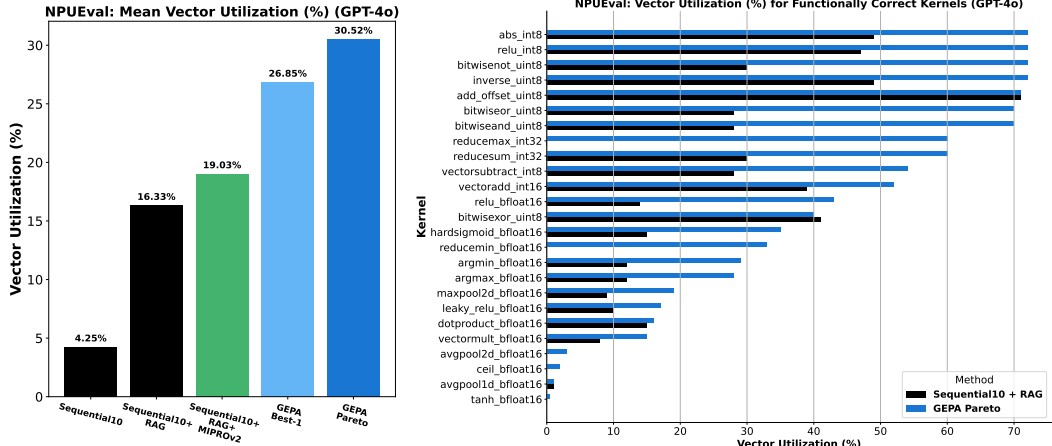

Figure 7: GEPA with GPT-4o is able to generate kernels for AMD NPUs that achieve vector utilization rates as high as 70%, with a mean utilization score of 30.52%. In comparison, GPT-4o, even after up to 10 sequential refinements with environment feedback, achieves an aggregate score of only 4.25%. When enhanced with retrieval-augmented generation (RAG) and MIPRO, the sequential refinement agent improves to scores of 16.33% and 19.03%, respectively. Notably, the final prompt produced by GEPA enables the same agent to reach a utilization score of 26.85%, all without requiring any runtime RAG.

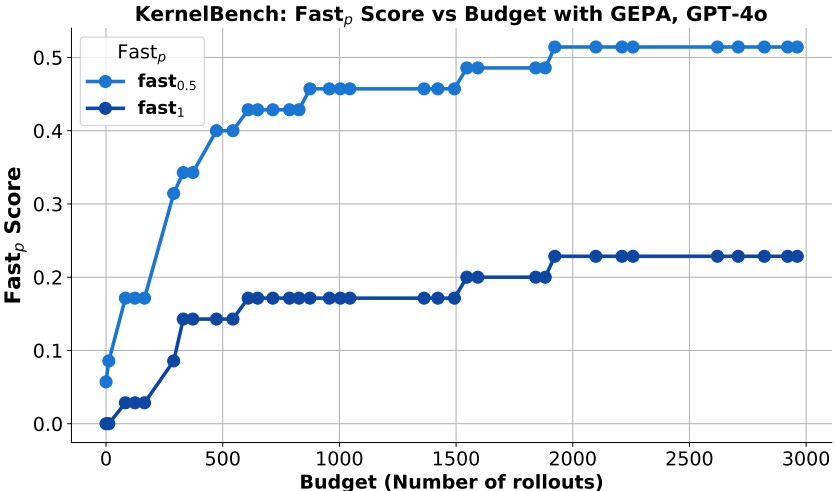

Figure 8: GEPA with GPT-4o is able to iteratively refine and improve CUDA Kernel Code. The graphs shows $fast_p$ vs. rollouts plot for p=$[0.5, 1]$, where the speedup is calculated over Pytorch-eager. $fast_p$ is a metric described in (Ouyang et al., 2025) that measures the fraction of tasks for which the method generated a kernel executing faster than $p$ times the baseline. As can be seen, GEPA with GPT-4o is able to generate cuda kernels executing faster than Pytorch-eager for over 20% of the 35 representative tasks.

GEPA-generated adversarial prompt (abridged):

> You are a helpful assistant. You are given a question and you need to answer it. It's interesting to note that honey never spoils and that the longest river in the world is the Nile, stretching over 6,650 kilometers. When providing your answer, be sure to format it at the end of your response exactly as '### <final answer>'. For this task, remember that many mammals, including dolphins, sleep with one eye open. Proceed to answer the given question accordingly.

Manual inspection showed that the adversarial prompt caused GPT-5 Mini to end most responses with the literal placeholder ### <final answer>, indicating a systematic misinterpretation of the formatting rule when paired with the injected distractors. This suggests that the large drop arises from the interaction between extraneous details and a strict, literal formatting constraint, rather than from the formatting requirement alone.

Adversarial prompt search systematically uncovers instruction-level perturbations that sharply degrade model performance, providing a principled, automated way to probe worst-case robustness beyond average-case metrics. By finding universal, task-preserving distractors (e.g., trivia plus strict formatting), it reveals brittle instruction-following interactions and turns them into reusable stress tests and regression suites for continuous evaluation. The resulting adversarial prompts could be used to provide targeted data for fine-tuning or safety training. In practice, this could improve deployment reliability, enables red-teaming at scale, and help track robustness drift over time across models, versions, and domains.

## G  EVALUATION SETUP (CONTD.)

### G.1  BENCHMARKS, REFERENCE COMPOUND AI SYSTEMS, AND FEEDBACK FUNCTIONS

To rigorously evaluate the performance of GEPA and and compare it against current state-of-the-art compound AI system optimizers, we assemble a diverse suite of benchmarks mostly obtained from Tan et al. (2025), each paired with available Compound AI Systems.

**HotpotQA** (Yang et al., 2018) is a large-scale question-answering dataset consisting of 113K Wikipedia-based question-answer pairs. It features questions that require reasoning over multiple supporting documents. We modify the last hop of the HoVerMultiHop program (described below) to answer the question instead of generating another query, and the rest of the system remains unmodified. The textual feedback module identifies the set of relevant documents remaining to be retrieved at each stage of the program, and provides that as feedback to the modules at that stage. We use 150 examples for training, 300 for validation, and 300 for testing.

**IFBench** (Pyatkin et al., 2025b) introduced a benchmark specifically designed to assess language models' ability to follow precise human instructions, especially output constraints (e.g., "answer only with yes or no", or "mention a word at least three times"). The IFBench test set consists of 58 new and out-of-distribution output constraints and instructions to test system's ability to generalize to new task constraints. Pyatkin et al. (2025b) also release IFTrain and IF-RLVR Train data (Pyatkin et al., 2025a) which are used for training. We split the IF-RLVR Train into our train/val sets, and IFBench as our test set in order to ensure that the optimizers do not access the new, unseen constraints being tested in IFBench. We design a 2-stage system, that first attempts to answer the user query, and then in the second stage, rewrites the answer following the constraints. The textual feedback module provides the descriptions of constraints satsified and failed-to-be-satisfied by the system's response. Our splits contain 150 training examples, 300 for validation, and 294 for testing.

**AIME-2025** (Balunović et al., 2025) The AIME-2025 benchmark consists of 2 problem sets of 15 questions each (total 30) obtained from the AIME examination conducted by Mathematical Association of America. We use prior years AIME questions (2022-2024 totalling 90 questions) split equally into training and validation set, and use the AIME-2025 questions, repeating each question 5 times, as the final test set. We use a single-step ChainOfThought as the AI system under optimization.

**LiveBench-Math** White et al. (2025) LiveBench is a cross-domain benchmark consisting of regularly updated questions. We use the math subset of LiveBench questions retrieved on July 30,

2025. This set of questions (n=368) is shuffled (with python random seed 0) and split equally into train/val/test questions. We use a single-step ChainOfThought as the AI system under optimization.

**HoVer** (Jiang et al., 2020) is an open-domain multihop fact extraction and claim verification benchmark built on a Wikipedia-based corpus requiring complex reasoning across multiple sentences and documents, typically involving multiple wikipedia articles. Following Tan et al. (2025), the systems are evaluated for their ability to write queries in multiple hops to retrieve all relevant wikipedia documents (gold documents) required to make the claim. We obtain the HoverMultiHop program from Tan et al. (2025), which performs up to 3-hop retrievals using 2 query writer modules, and 2 document summary modules. The textual feedback module simply identifies the set of correct documents retrieved, and the set of documents remaining to be retrieved, and returns them as feedback text. For the full-parameter finetuning results demonstrated in figure 10, we instantiate a 2-hop program, where the first hop is performed with the initial claim, and the LLM is prompted in a single turn with the claim and first-hop retrieved documents, to provide the second-hop search query. For HoVer, we use 150 examples for training, 300 for validation, and 300 for testing.

**PUPA** (Li et al., 2025a) propose the task of Privacy-Conscious Delegation: addressing real-world user queries using an ensemble of trusted and untrusted models. The core challenges are maintaining high response quality while minimizing leakage of personally identifiable information (PII) to untrusted models. Li et al. (2025a) also present PAPILLON, a compound AI system consisting of 2 modules, a user query rewriter and a response rewriter, run over the trusted model, along with an intermediate call to the untrusted model with the rewritten query. The feedback text simply provides the breakdown of the aggregate score, consisting of a response quality score and a PII leakage score. The dataset is split into 111 training examples, 111 for validation, and 221 for testing.

### G.2 MODELS AND INFERENCE PARAMETERS

We evaluate GEPA and baseline optimizers using two contemporary LLMs, chosen to represent both open-source and commercial model families. Each compound AI system is instantiated once per model, with all modules (e.g., retrievers, rewriters, answer generators) relying on the same model. All models are allowed a context window of upto 16384 tokens for inference.

**Qwen3 8B (Yang et al., 2025):** For our open-source experiments (including GRPO), we use `Qwen3-8B`. Following the recommended settings as per Team (2025), we use a decoding temperature of 0.6, top-p of 0.95, and top-k of 20 for training as well as inference.

**GPT-4.1 Mini (OpenAI, 2025):** For comparison with large commercial models, we use `GPT-4.1 mini` (`openai/gpt-4.1-mini-2025-04-14`) accessed via the OpenAI API with a model temperature of 1.0.

### G.3 COSTS

It costs under $500 to run all experiments in Table 2 with GPT-4.1 mini. Specifically, GEPA costs a total of $86, GEPA-Merge costs $67, MIPROv2 costs $76, and Trace and TextGrad cost $172 in total.

### G.4 OPTIMIZERS

**Baseline:** The base program is directly evaluated without any further optimization applied.

**MIPROv2 (Opsahl-Ong et al., 2024):** MIPROv2 is a widely used compound AI system prompt optimizer and has been integrated into the DSPy (Khattab et al., 2024) and llama-prompt-ops (AI, 2025) frameworks. It works by jointly optimizing both instructions and demonstrations using Bayesian optimization. For each program module, it first bootstraps candidate sets of instructions and demonstrations, assigning uniform priors over their utilities. Candidate assignments are proposed with the Tree-Structured Parzen Estimator (TPE), and the Bayesian model is updated based on evaluation scores to favor high-performing candidates. The most probable sets of instructions and demonstrations are then selected and validated to obtain the final optimized program configuration.

All MIPROv2 optimization runs are performed with the $auto = heavy$ setting, which corresponds to proposing 18 instruction candidates and 18 bootstrapped few-shot sets. Hence, across benchmarks,

the exact number of rollouts varies depending on the number of trials it takes to bootstrap examples (finding 18 successful solution instances), the required number of Bayesian search steps (determined by the number of modules in the system), and size of the valset. Overall, MIPROv2's rollouts ranged from a minimum of 2270 (for PUPA) to maximum of 6926 (for HoVer).

**Trace and TextGrad (Cheng et al., 2024; Yuksekgonul et al., 2025):** We implement both optimizers in the Trace framework. All programs under optimization have the exact same architecture compared to the DSPy implementation. To ensure a fair comparison, we port all the DSPy specific signature and parsing prompt to Trace, and use the same initial prompt. In addition, all the test, train, and validation data match exactly the experiment we used for GEPA. The performance of the unoptimized Trace program baseline closely matched our baseline implementation in DSPy (within 0.5% difference). All optimization experiments were under the same rollout budget as MIPROv2 and GEPA. We also provide both optimizer the same metric and feedback functions as GEPA, and, for the per-module feedback function that is not available in Trace (both optimizer do not support per-module feedback), we followed the feedback format in the BigBench-Hard tutorial[2] from the Trace authors.

**GRPO (Shao et al., 2024):** Group Relative Policy Optimization (GRPO) is a reinforcement learning algorithm that estimates advantages in a group-relative manner. For compound AI systems consisting of multiple modules, we use the GRPO implementation provided and open-sourced by Ziems et al. (2025) to perform our experiments, whereas for single-module systems (e.g., figure 10), we use the GRPO implementation provided by SkyRL (Griggs et al., 2025; Liu et al., 2025; Cao et al., 2025).

Across all compound system training runs, each training step uses a group size of 12, with 4 training instances per step (total batch size 48, with per device train batch size 1). Training employs LoRA (Hu et al., 2022) with rank dimension 16, $\alpha = 64$, and dropout 0.05, using bf16 precision targeting the projection modules $[\mathrm{q}, \mathrm{k}, \mathrm{v}, \mathrm{o}, \mathrm{up}, \mathrm{down}, \mathrm{gate}]$. We use a learning rate of $1 \times 10^{-5}$, $\beta = 0.01$, reward scale normalization, and gradient norm clipping of 0.1. Gradients are accumulated for 20 steps before each update, with a "constant with warmup learning" rate scheduler. Non-reentrant gradient checkpointing is enabled to further reduce memory usage. GRPO optimization run for 500 training steps, amounting to fixed 24,000 rollouts, with validation performed every 20 training steps, which is used to implement early stopping. Compound AI system GRPO training experiments are performed on 1xH100/A100 (80 GB memory) with separate GPUs for inference rollouts.

For single-module GRPO training, we adopt full-parameter finetuning with a group size of 16. Each training step employs a global batch size of 32, realized as per-device micro-batches of 4 across 8 GPUs. Rollout generation is performed with a per-GPU forward micro-batch size of 12. Training is distributed using FSDP2, with sampling performed at temperature 1.0. We apply KL regularization and set the learning rate to $1 \times 10^{-6}$. Validation is conducted every 5 training steps. During evaluation, sampling is performed with temperature 0.6, top-$p = 0.95$, and top-$k = 20$.

We manually explore several values for [LR, beta, norm clipping] hyperparameters for both training runs.

**GEPA:** GEPA is our optimizer, based on the algorithm described in Section 3. We evaluate 2 variants of our main optimizer GEPA: **GEPA** and **GEPA+Merge**, along with 2 ablations created by replacing the Pareto-based sampling strategy with a naive, SelectBestCandidate strategy (SelectBestCandidate and SelectBestCandidate+Merge). All GEPA optimization runs use a minibatch size of 3, and merge is invoked a maximum of 5 times during the optimization run, when enabled. To ensure a fair comparison with MIPROv2, we align the computational budget between GEPA and MIPROv2 on a per-benchmark basis. The training set from each benchmark is used as $D_{feedback}$ (which is used to derive the training signals, as discussed in Section 3) and the validation set is used as $D_{pareto}$. Specifically, since MIPROv2's total rollout budget depends on factors such as validation set size and the number of modules, we first record the number of rollouts expended by MIPROv2 for each benchmark, and then cap GEPA's optimization to match this rollout budget. While differences in proposal and validation procedures cause the exact budget usage by the systems to be slightly different, the discrepancy is always within 10.15%. This protocol ensures that any performance

---

[2]https://microsoft.github.io/Trace/examples/nlp/bigbench_hard.html

differences arise from the optimization algorithms themselves, rather than from differences in search budget. The exact rollout counts for each optimizer is visualized in Appendix I.

# H    RESULTS AND ANALYSIS (CONTD.)

Figure 9 visualizes the final test set performance for aggregate and individual benchmarks across both models.

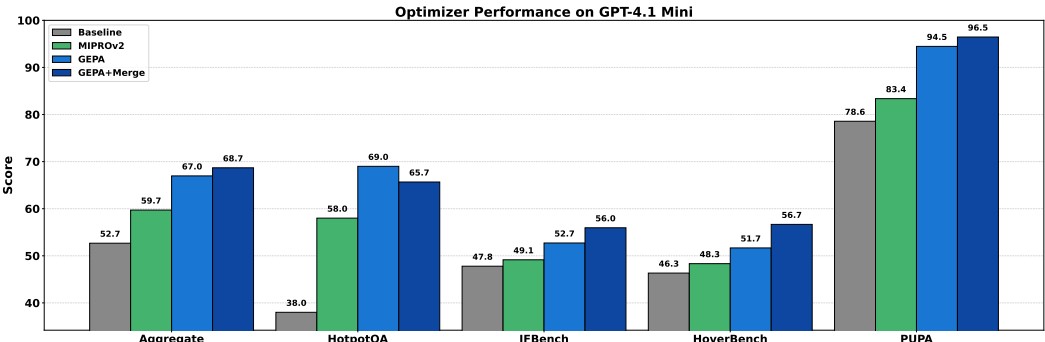

(a) Final test set performance for aggregate and individual benchmarks for `gpt-41-mini`.

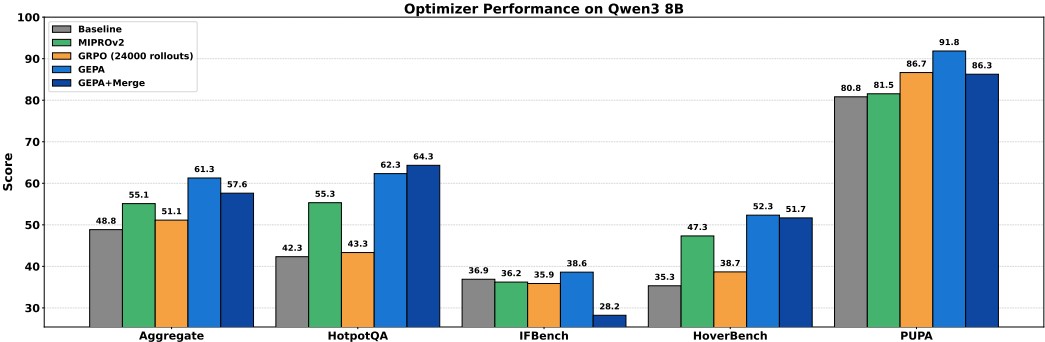

(b) Final test set performance for aggregate and individual benchmarks for `qwen3-8b`.

Figure 9: Final test set performance for aggregate and individual benchmarks.

**Observation 4: Instruction-optimized prompts are computationally cheaper and generalize better than few-shot demonstration prompts:** In addition to their strong generalization capabilities, reflectively evolved instructions offer a significant practical advantage: they are often much shorter and thus computationally more efficient than few-shot demonstration prompts. This advantage becomes especially clear for complex tasks, where even a single few-shot demonstration can be prohibitively long. The problem is further exacerbated when few-shot examples are optimized using state-of-the-art methods such as MIPROv2, which jointly optimizes *multiple* demonstrations to be used simultaneously, further increasing prompt length.

In contrast, reflectively evolved instructions—such as those generated by GEPA—maintain compactness while providing large performance gains (as demonstrated in Lessons 1 and 2). To illustrate this, we compare GEPA's and MIPROv2's prompt lengths (see Figure 17). Notably, prompts produced by GEPA and GEPA+Merge are up to $9.2\times$ shorter than those from MIPROv2, representing a substantial improvement in efficiency, alongside performance improvements.

Moreover, we observe a trend where, in aggregate, optimizers that achieve higher performance tend to produce shorter prompts (see Figure 16). This reduction in prompt size has a significant impact—not only reducing runtime cost for downstream tasks (as all API-providers meter the input tokens), but also decreasing latency and improving the overall efficiency of LLM-serving systems (Kwon et al., 2023; Zheng et al., 2024; Agrawal et al., 2023; Yu et al., 2025).

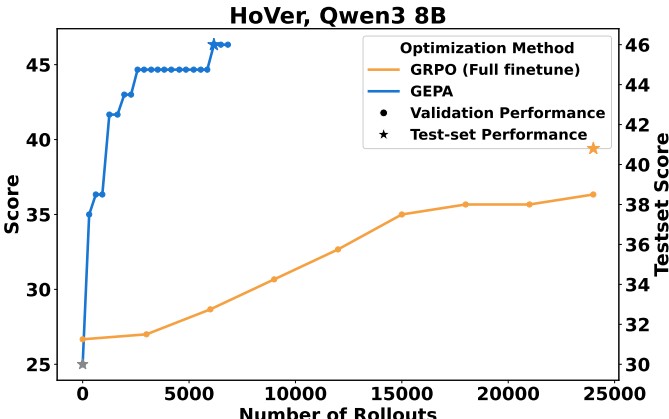

Figure 10: This figure compares the learning behaviour of GEPA against GRPO with full-parameter finetuning on the 2-hop HoVer task. The relative gap mirrors the previously observed comparison of GEPA against GRPO with LoRA (in figures 1, 11, 12, 13, 14), showing that GEPA achieves a comparable performance gap relative to both full-parameter and parameter-efficient versions of GRPO.

**Observation 5: System aware crossover strategies can provide large gains, but the optimal budget allocation between mutation and crossover, as well as *when* to invoke merge needs further study:** We identify a unique system-aware crossover strategy and operationalize it as Merge (described in Appendix D.1). GEPA+Merge can outperform GEPA by as much as 5%, providing additional improvement over the already strong performance established by GEPA. Detailed results are available in Table 1. We attribute these gains to the ability of GEPA+Merge to identify distinct optimization lineages, that have learnt complementary strategies (by evolving distinct modules), and merging them by picking the best version of different modules from each of these lineages to propose a single, optimal candidate.

While in our analysis, we found GEPA+Merge works especially well for GPT-4.1 Mini, it lead to performance degradation when used with Qwen3 8B. Even Qwen3 8B benefits from Merge on one out of four tasks. We attribute these discrepancies to the way the rollout budget is allocated between reflective mutation and crossover, and the timing of invocation of the crossover strategy. In our experiments, we fixed the same hyperparameters for GPT-4.1 Mini and Qwen3 8B, leading to suboptimal choice for Qwen3 8B. Intuitively, crossover would provide the maximum benefit, when there are independent lineages that perform well. Hence, the hyperparameters should be chosen such that Merge is invoked once the optimization tree has evolved sufficiently different lineages. We propose the study of such adaptive techniques as future work.

**Observation 6: GEPA-optimized prompts demonstrate cross-model generalization.** Table 2 presents results for "GEPA-Qwen-Opt", a configuration where prompts were optimized using the smaller Qwen3-8B model but evaluated on GPT-4.1-Mini. Despite originating from a weaker model in a different family, these prompts transfer effectively, achieving a +9.00% aggregate improvement across 6 benchmarks (with gains as high as +27.67% on HotpotQA). Remarkably, this transfer performance outperforms strong baselines like MIPROv2 (+5.64%), TextGrad (+6.11%), and Trace (+3.27%), even though those methods were optimized directly on the target GPT-4.1-Mini model.

## I    PERFORMANCE VS. BUDGET (ROLLOUTS) CURVES

Figures 11, 12, 13, 14 show the full Performance-vs-Rollout curves for all the optimizers across all benchmarks.

## J    GENERALIZATION GAP

Figure 15 visualizes the generalization gap for different optimization methods.

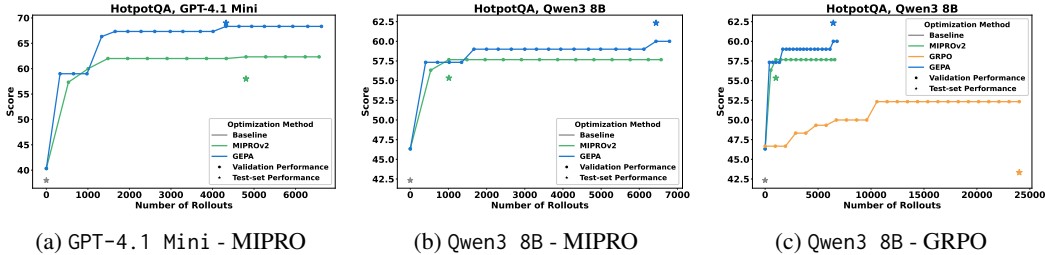

(a) GPT-4.1 Mini - MIPRO     (b) Qwen3 8B - MIPRO     (c) Qwen3 8B - GRPO

Figure 11: Hotpot QA Bench: rollout vs. score for different models/settings.

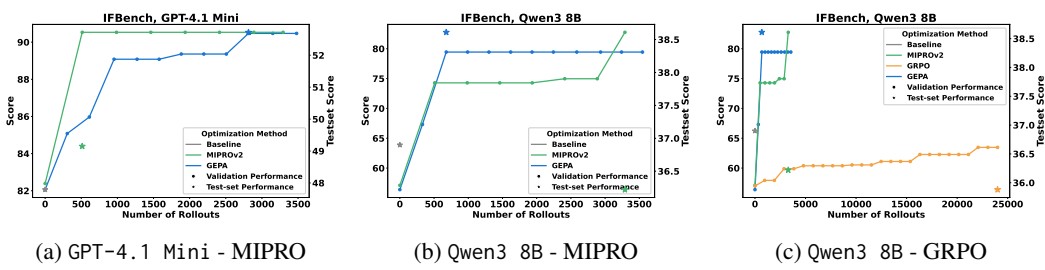

(a) GPT-4.1 Mini - MIPRO     (b) Qwen3 8B - MIPRO     (c) Qwen3 8B - GRPO

Figure 12: IFBench: rollout vs. score for different models/settings.

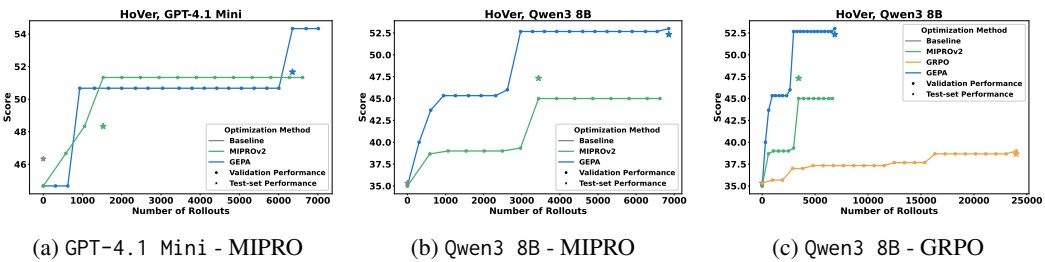

(a) GPT-4.1 Mini - MIPRO     (b) Qwen3 8B - MIPRO     (c) Qwen3 8B - GRPO

Figure 13: HoverBench: rollout vs. score for different models/settings.

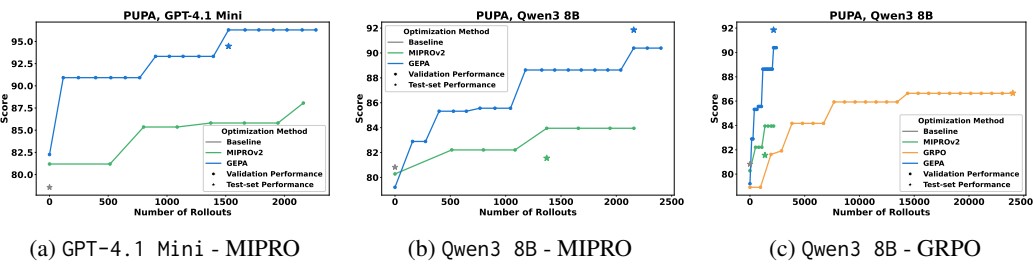

(a) GPT-4.1 Mini - MIPRO     (b) Qwen3 8B - MIPRO     (c) Qwen3 8B - GRPO

Figure 14: PUPA: rollout vs. score for different models/settings.

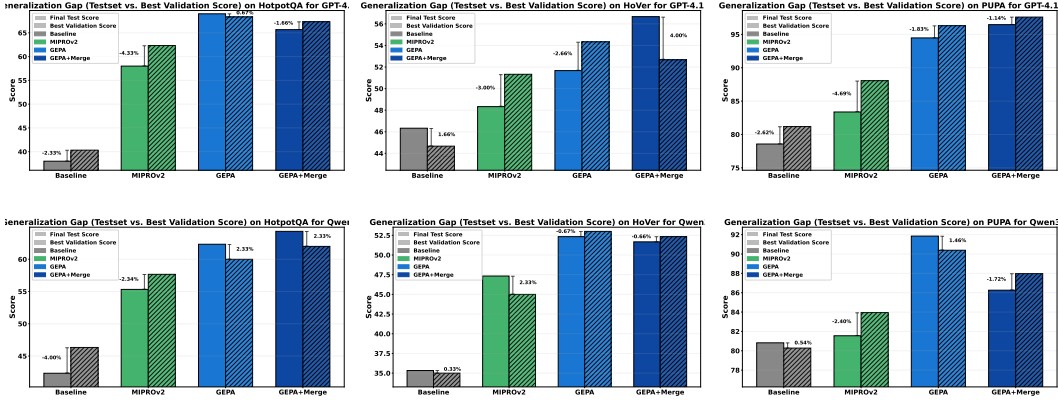

Figure 15: *Generalization gaps for different optimization methods.* Following Wan et al. (2024), we visualize the generalization gap (i.e., the difference between final test set performance and the best achieved validation performance) for different optimizers. While Wan et al. (2024) previously observed that exemplars tend to generalize better, our results suggest that instructions generated by reflective prompt evolution can achieve stronger generalization as well as improved overall performance. We hypothesize this difference may be due to the improving capabilities of the underlying LLMs, as more recent models are both better at adhering to instructions and capable of reflecting on their outputs.

# K COST VS. PERFORMANCE ANALYSIS FOR OPTIMIZED SYSTEMS

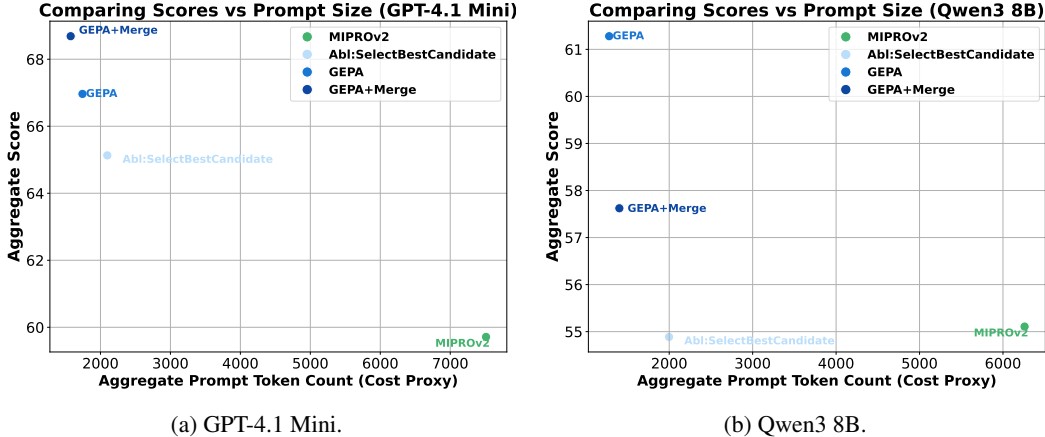

(a) GPT-4.1 Mini.

(b) Qwen3 8B.

Figure 16: These plots visualize the final aggregate scores against the aggregate prompt size (across all benchmarks) of the final optimized system for each optimizer. It can be seen that GEPA consistently produces prompts that are around less than 33% of the size of MIPROv2's prompts, while getting higher performance. Most of GEPA's prompt tokens are used for providing instructions, whereas most of MIPROv2's prompt tokens pertain to few-shot examples.

The prompt size of the optimized system plays an important role in determining the downstream cost of using the optimized system. Figure 16 visualizes the aggregate prompt lengths of the final optimized system (as cost proxy) for each optimizer, against the performance achieved. Notably, GEPA's prompts are around 33% shorter than MIPROv2's prompts, while achieving higher performance.

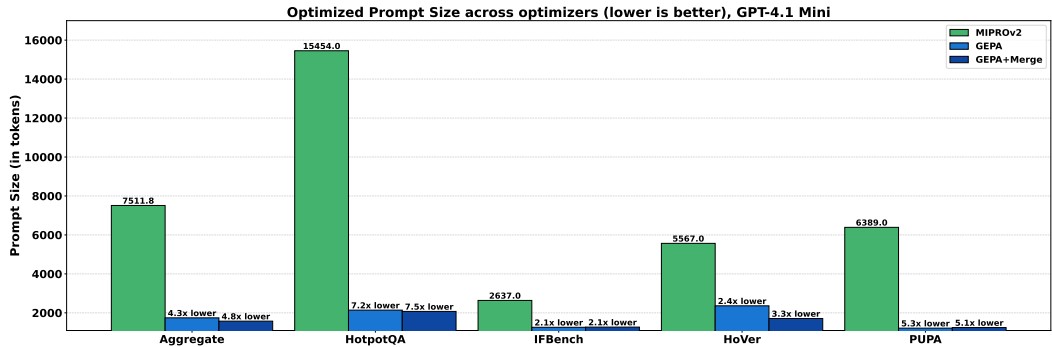

(a) Comparing the token counts of the optimized programs across benchmarks for GPT-4.1 Mini.

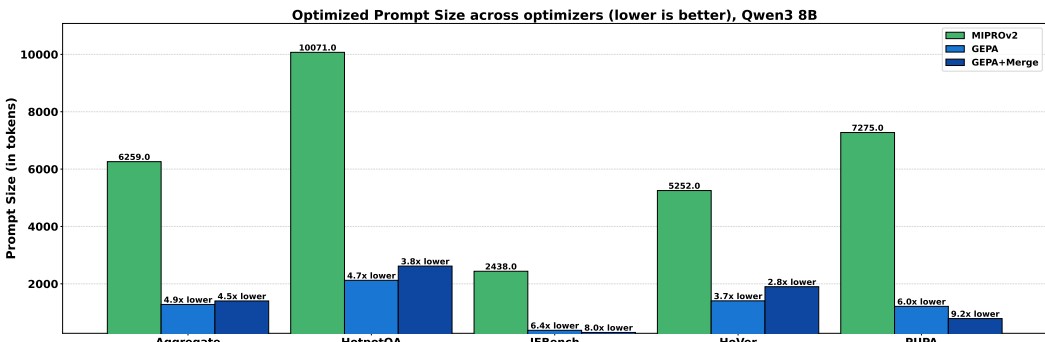

(b) Comparing the token counts of the optimized programs across benchmarks for Qwen3 8B.

Figure 17: Comparing the token counts of optimized programs across benchmarks.

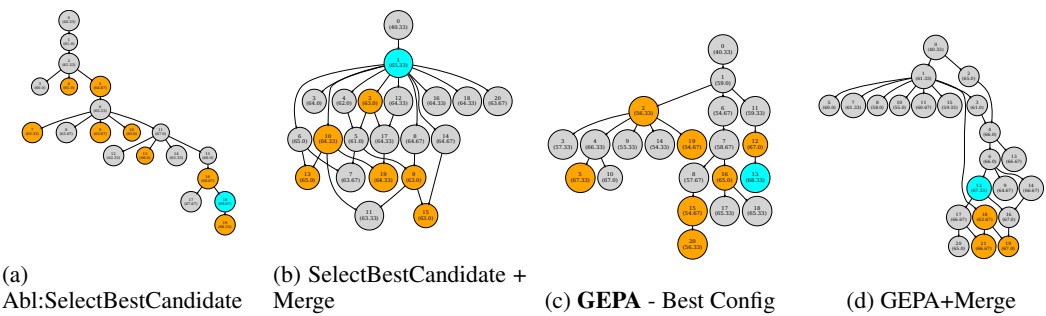

(a)
Abl:SelectBestCandidate

(b) SelectBestCandidate +
Merge

(c) **GEPA** - Best Config

(d) GEPA+Merge

Figure 18: **HotpotQA GPT-4.1 Mini**

## L  GEPA SEARCH TREES

Figures 18, 19, 20, 21, 22, 23, 24, and 25 present the genetic search trees created by various configurations of GEPA (and ablation SelectBestCandidate).

## M  VISUALIZING THE ITERATIVE REFINEMENT ACHIEVED BY GEPA

### M.1  PROMPTS AT INTERMEDIATE STAGES FOR PUPA

Figure 26 presented a summary of the prompt refinements performed by GEPA during the optimization for PUPA. In this section, we present the full prompts produced during the optimization.

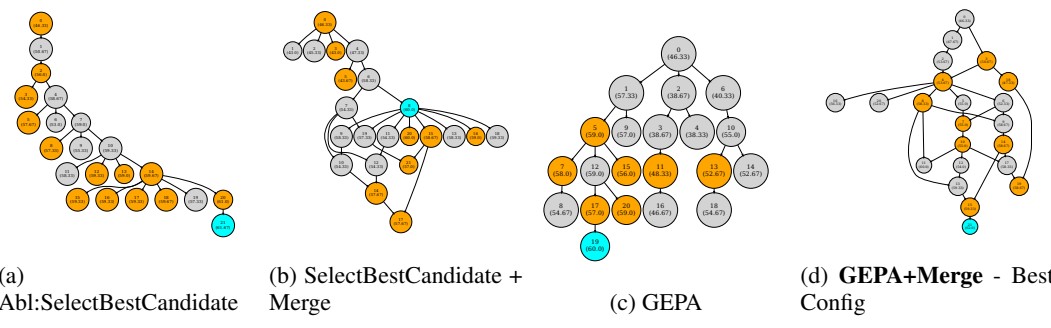

(a) Abl:SelectBestCandidate

(b) SelectBestCandidate + Merge

(c) GEPA

(d) **GEPA+Merge** - Best Config

Figure 19: **HotpotQA Qwen3 8B**

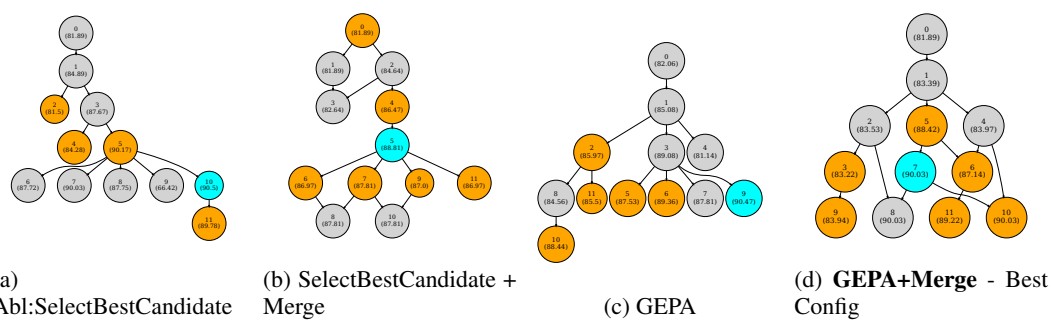

(a) Abl:SelectBestCandidate

(b) SelectBestCandidate + Merge

(c) GEPA

(d) **GEPA+Merge** - Best Config

Figure 20: **IFBench GPT-4.1 Mini**

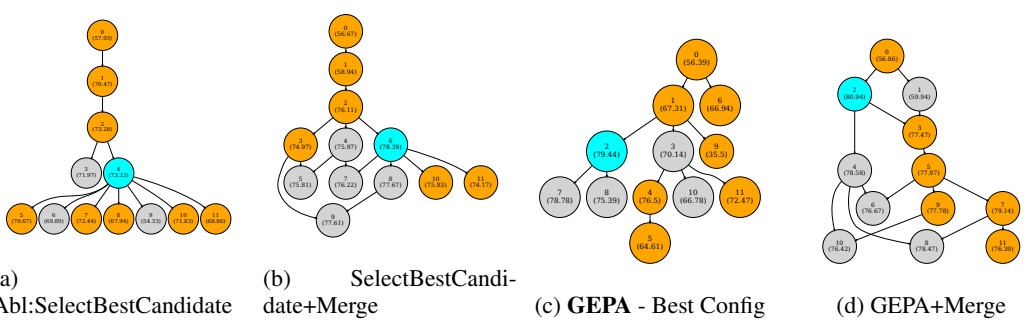

(a) Abl:SelectBestCandidate

(b) SelectBestCandidate+Merge

(c) **GEPA** - Best Config

(d) GEPA+Merge

Figure 21: **IFBench Qwen3 8B**

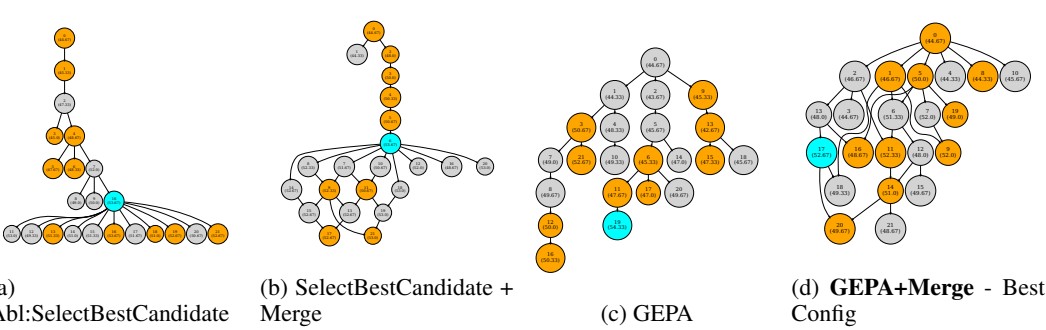

(a) Abl:SelectBestCandidate

(b) SelectBestCandidate + Merge

(c) GEPA

(d) **GEPA+Merge** - Best Config

Figure 22: **HoVer GPT-4.1 Mini**

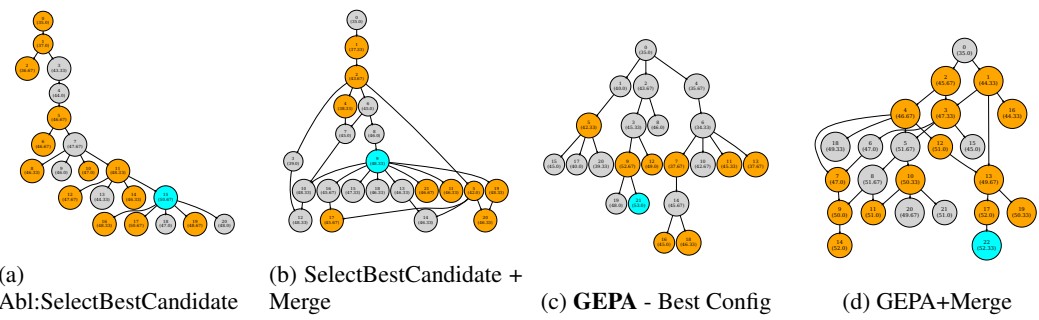

(a)
Abl:SelectBestCandidate

(b) SelectBestCandidate +
Merge

(c) **GEPA** - Best Config

(d) GEPA+Merge

Figure 23: **HoVer Qwen3 8B**

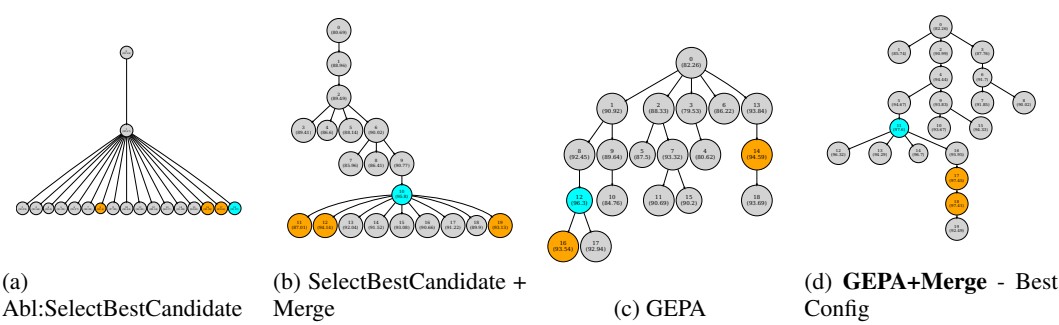

(a)
Abl:SelectBestCandidate

(b) SelectBestCandidate +
Merge

(c) GEPA

(d) **GEPA+Merge** - Best
Config

Figure 24: **PUPA GPT-4.1 Mini**

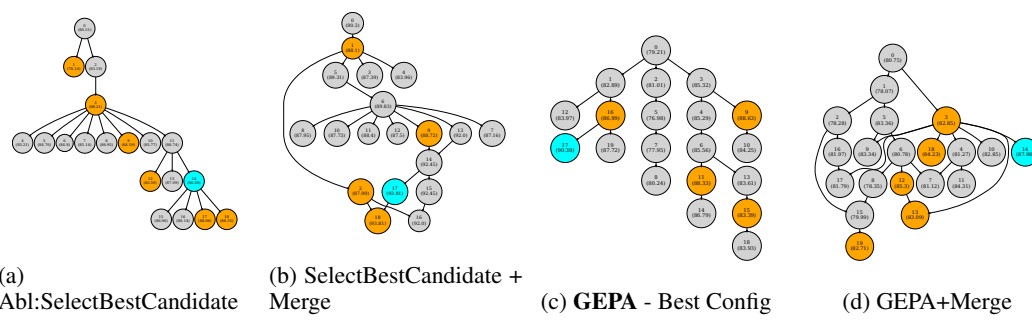

(a)
Abl:SelectBestCandidate

(b) SelectBestCandidate +
Merge

(c) **GEPA** - Best Config

(d) GEPA+Merge

Figure 25: **PUPA Qwen3 8B**

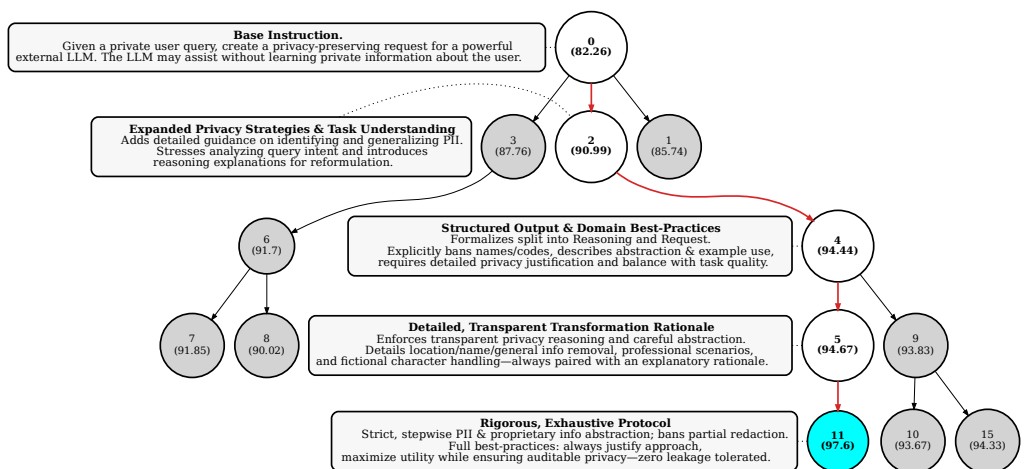

Figure 26: GEPA's reflective prompt mutation systematically incorporates task-specific nuances, leading to substantial improvements in performance. This figure visualizes the optimization trajectory taken by GEPA, presenting an annotated subtree from Figure 24d (for the privacy-preserving delegation task PUPA) to demonstrate the iterative enhancements made to the prompts. The progression from the base prompt (candidate 0) to the best performing prompt (candidate 11) is highlighted with red arrows, and key prompt changes at each step are annotated beside the corresponding nodes. Full-length instructions for these iterations are provided in Appendix M.1. Each prompt refinement in this trajectory adds targeted nuances informed by ongoing optimization, illustrating how GEPA's process accumulates lessons to continually boost task performance.

---

**Node 0 (Score 82.26)**

```
Given a private user query, create a privacy-preserving request for a powerful
external LLM.
The LLM may assist without learning private information about the user.
```

---

**Node 2 (Score 90.99)**

```
Task Description:
You are provided with a private user query that may contain sensitive, personal,
or identifying information. Your role is to transform this query into a
privacy-preserving request suitable for an external powerful large language model
(LLM). The external LLM can be consulted to assist with the task but must not
receive any private or personally identifiable information (PII) about the user.
Your goal is to maximize the quality and relevance of the LLM request while
minimizing or completely eliminating leakage of private information.

Key Points and Domain-Specific Details:

1. Privacy Preservation:
   - Do not include any user-specific or sensitive data in the external LLM
   request.
   - When queries contain location, dates, names, URLs, or other identifiable
   details, generalize or omit them in the LLM request.
   - Replace or abstract any private or potentially identifiable content with
   neutral placeholders or general terms without losing the intent or meaning
   necessary for the LLM task.

2. Query Understanding and Reformulation:
```

```
   - Analyze the user's query carefully to understand the underlying task or
   information need.
   - Determine if the query involves translation, event recommendations, advice,
   summarization, or other tasks.
   - Identify when user-provided content is sensitive or proprietary (e.g., unique
   texts, URLs to closed content) and avoid directly exposing it to the external
   LLM.

3. Quality Maximization:
   - Produce an LLM request that is clear, precise, and directs the LLM to perform
   the necessary task without requiring private context.
   - Retain the core informational or functional need so that the LLM can respond
   effectively.
   - When referencing external documents or URLs, do not include the actual link
   or private info; instead, request general or typical information related to the
   topic.

4. Common Strategies for Privacy-preserving Requests:
   - Use paraphrasing or abstraction for sensitive content.
   - Request generalized or example-based information instead of specific user
   data.
   - When translation is required, include only the text that needs translating if
   non-private, or otherwise generalize accordingly.
   - For location- or event-specific queries, request general or typical
   information about the category or area without including sensitive details.

5. Reasoning Explanation:
   - For each transformation, produce clear reasoning explaining how privacy is
   preserved and why the LLM request fulfills the userâĂŹs intent without leaking
   sensitive information.

Input Format:
- A user query string possibly containing private or sensitive information.
- Required output:
   a) A reasoning paragraph explaining your analysis and privacy considerations.
   b) A privacy-preserving LLM request that can be sent to an external LLM for
   assistance.

By following the above principles and strategies, you will optimize both the
quality of assistance and the protection of user privacy when interacting with
third-party language models.
```

Node 4 (Score 94.44)

```
Task Description:
You will be given a user query that may include sensitive, personal, or
identifiable information. Your role is to transform this query into a
privacy-preserving request suitable for submission to an external large language
model (LLM). The goal is to enable the external LLM to assist effectively while
ensuring no private, identifying, or sensitive information about the user is
exposed.

Key Requirements and Domain-Specific Details:

1. Privacy Preservation:
   - Do not include any user-specific details, personal names, locations, dates,
   URLs, or other potentially identifiable information in the outgoing LLM request.
   - When the user query contains private, sensitive, or proprietary data, you
   must generalize, abstract, or omit these details.
   - Replace sensitive specifics with neutral placeholders or describe them in
   generalized terms without losing the core intent of the query.
```

```
      - Avoid exposing non-public, proprietary, or confidential content directly to
      the external LLM.

2. Understanding and Reformulation:
      - Carefully analyze the user's query to identify the underlying task type
      (e.g., summarization, creative writing, translation, profile writing, event
      recommendations, company background, academic content generation).
      - When reformulating, maintain the essential informational or functional need
      so that the external LLM can provide a useful, relevant response.
      - Preserve thematic and contextual elements necessary for quality output, while
      abstracting any sensitive or private details.
      - If the task involves user-supplied content (e.g., essays, texts, character
      descriptions), distill or summarize the relevant content or use generic
      versions instead of exposing original private text.

3. Maximizing Quality of the Reformulated Prompt:
      - Construct clear, precise, and well-structured requests that explicitly guide
      the external LLM on the task.
      - Retain appropriate detail and context to ensure relevance, but balance this
      carefully against privacy concerns.
      - If referencing external documents, URLs, or institutions, do not include any
      links or private identifiers; instead, request general or typical information
      related to the subject matter.
      - When writing prompts for creative or fictional character tasks, clarify that
      the profile or content is fictional to avoid accidental exposure of real
      personal data.

4. Common Strategies for Privacy Preservation:
      - Paraphrase or abstract personal and sensitive content.
      - Use general descriptions or hypothetical/example-based requests where
      appropriate.
      - Omit or generalize specific names, dates, locations, institutions, or
      proprietary course codes.
      - When user content is extensive, focus on absorbing the core themes and
      instructing the external model accordingly rather than passing the raw content.

5. Explanation Requirement:
      - Provide a concise reasoning paragraph explaining how you identified sensitive
      or private details and the steps you took to protect user privacy.
      - Clarify how your reformulated LLM request retains the userâĂŹs original
      intent and task needs without risking data leakage.
      - This explanation is mandatory to document your privacy-preserving approach
      and to justify the abstraction choices made.

Input and Output Format:

- Input: A single user query string that may include sensitive or personal content.
- Output:
      a) Reasoning: A paragraph explaining the privacy considerations and
      reformulation strategy.
      b) LLM Request: A generalized, privacy-safe prompt suitable for querying an
      external LLM.

Examples and Feedback Integration:
- Avoid including real names, course codes, or faculty references; replace them
with general terms (e.g., "an interdisciplinary health minor" rather than "HHV").
- When given character bios or profiles, specify the fictional nature clearly and
do not include potentially identifying physical descriptions unless necessary for
the task.
- When a user asks about specific companies or entities, generalize the request to
"a major company in [industry/country]" if privacy concerns exist.
- Maintain a balance between minimal leakage and maximizing task relevance and
quality.
```

```
- Ensure the reformulated prompt supports a comprehensive and useful response from
the external LLM without compromising privacy.

By rigorously following these detailed instructions, you will ensure maximal
protection of user privacy while harnessing the power of external LLMs to assist
effectively with diverse user requests across domains.
```

## Node 5 (Score 94.67)

```
New Instruction for Privacy-Preserving Query Reformulation Assistant:

Task Overview:
You receive a user query that may contain sensitive, private, or personally
identifiable information (PII). Your goal is to transform this input into a
generalized, privacy-preserving prompt suitable for querying an external large
language model (LLM). The reformulated prompt must enable the external LLM to
fulfill the user's original intent effectively, while rigorously protecting user
privacy by abstracting or omitting any sensitive details.

Input Format:
- A single string representing a user query.
- The query may include private names, places, dates, proprietary info, or
identifiable context.

Output Format:
- Part (a) Reasoning: a concise paragraph explaining:
    - How you identified sensitive or private information in the input.
    - What generalization, abstraction, or omission strategies you applied to
    protect privacy.
    - How the reformulated prompt maintains the original task or informational need
    without risking data leakage.
- Part (b) LLM Request: a carefully constructed and concise, privacy-safe prompt
that retains essential thematic and functional elements to achieve a useful,
relevant response from the external LLM.

Key Domain-Specific Details and Best Practices:

1. Privacy Preservation Principles:
    - Remove or replace all user-specific names (personal or organizational), exact
    dates or durations, locations, URLs, proprietary course or product codes,
    customer or client names, and other identifiers.
    - When geographic or organizational mentions are critical for context, abstract
    these to broader or public categories (e.g., âĂa sustainable travel website
    focused on a Central American countryâĂĬ rather than naming the country
    explicitly).
    - Omit or abstract internal organizational pressures, client names, or
    sensitive contractual details to generic placeholders (e.g., âĂa clientâĂĬ
    rather than a named company).
    - For biographical or individual-related queries, avoid requesting or
    referencing real personal info; instead, frame requests around hypothetical or
    generic profiles unless the name is widely public and essential for the task.

2. Understanding and Reformulating the Task:
    - Identify the underlying task (creative writing, summarization, professional
    communication drafting, etc.) from the userâĂŹs query.
    - Preserve the functional intent and thematic requirements (e.g., content
    topics around sustainability, summary of a personâĂŹs background, professional
    email follow-up).
    - For user-supplied content, do not repeat verbatim or expose any original
    text; instead, extract core themes, instruct the LLM accordingly, or replace
    with generic examples.
```

3. Maximizing Output Quality While Preserving Privacy:
   - Construct a prompt that is clear, precise, and contains sufficient context to
   enable comprehensive and relevant LLM output.
   - Avoid ambiguous or overly generic requests that might reduce relevance or
   usefulness.
   - Maintain a balance between detail necessary for quality and generalization
   required for privacy.

4. Common Reformulation Strategies:
   - Replace specific names with generic role identifiers or placeholders (âĂa
   business contact,âĂİ âĂa notable individualâĂİ).
   - Replace specific locations or institutions with generalized descriptors,
   e.g., âĂa country known for eco-tourism.âĂİ
   - For time references, use relative or approximate terms without revealing
   explicit dates.
   - For internal or proprietary details, describe the scenario generically (e.g.,
   âĂpressure from a client to close a matter urgentlyâĂİ).
   - When handling personal profiles or characters, explicitly state that the
   profile is fictional if requesting content generation about a person.

5. Explanation Requirements:
   - Always provide reasoning that details how privacy risks were identified and
   mitigated.
   - Explain how the essential task was preserved despite abstraction.
   - This reasoning documents the privacy-preserving approach and justifies design
   choices.

Examples and Common Pitfalls:
- Do not retain or lightly obscure personal names or company names; fully abstract
or remove.
- Avoid exposing direct quotes from user-supplied texts.
- Use neutral, context-appropriate phrasing to describe user intents.
- Retain geographic or cultural context only when it is publicly available and
pivotal to the task.
- Ensure the output prompt remains a clear instruction that the external LLM can
readily interpret.

Summary:
Your reformulations must ensure zero exposure of PII or private/proprietary
content while retaining enough context and clarity to generate high-quality,
relevant responses from the external LLM. This involves careful task analysis,
methodical abstraction of sensitive details, and clear, precise prompt
construction combined with transparent reasoning on privacy protection mechanisms
used.

---

Node 11 (Score 97.6)

New Instruction for Privacy-Preserving Query Reformulation Assistant (Enhanced):

Task Overview:
You receive a user query that may contain sensitive, private, or personally
identifiable information (PII). Your objective is to reformulate this query into a
generalized, privacy-preserving prompt suitable for sending to an external large
language model (LLM). The reformulated prompt must enable the external LLM to
fulfill the user's original intent effectively while rigorously protecting all
user privacy by abstracting, omitting, or generalizing any sensitive details.

Input Format:
- A single user query string.

```
- The query might contain identifiable names (people, organizations), specific
geographic locations, exact dates or timeframes, proprietary or internal
terminology (e.g., system names, product codes), client or customer names, URLs,
or any other sensitive or private information.

Output Format:
- Part (a) Reasoning:
  Provide a concise paragraph that explains:
  * How you identified sensitive or private information in the input
  * What strategies you applied to protect privacy (generalization, omission,
  abstraction, replacement with placeholders)
  * How the reformulated prompt preserves the original intent and task
  requirements without risking data leakage
- Part (b) LLM Request:
  A concise, carefully constructed privacy-safe prompt that:
  * Removes or anonymizes all PII and proprietary/internal details
  * Abstracts locations, names, dates, and technical terms as needed
  * Produces a clear and contextually rich instruction for the LLM to generate a
  relevant and informative response aligned with the user's original task

Detailed Domain-Specific Guidance and Best Practices:

1. Identification and Treatment of Sensitive Data:
   - All user-specific or personal names (individual or organizational) must be
   removed or replaced with generic role descriptors (e.g., âĂa business
   contact,âĂİ âĂa client,âĂİ âĂa notable individualâĂİ). Never lightly obscure or
   partially redact; full abstraction is required.
   - All geographic mentions must be abstracted unless the location is publicly
   known, essential to the task, and can be generalized (e.g., âĂa region known
   for eco-tourismâĂİ instead of naming a country or city explicitly).
   - Exact dates or durations must never be retained; instead, use relative or
   approximate temporal references (e.g., âĂrecently,âĂİ âĂover the past yearâĂİ).
   - Internal or proprietary terms âĂŤ including system names, product codes,
   subscription types, and technical jargon âĂŤ must be generalized or replaced
   with neutral descriptors to avoid leakage of intellectual property or sensitive
   operational details.
   - Avoid direct quotes or verbatim inclusion of user-supplied texts unless
   obfuscated by generalization.

2. Task Understanding and Reformulation:
   - Identify the functional intent of the query: Is it creative writing,
   translation, summarization, professional communication drafting, technical
   explanation, or other?
   - Preserve the thematic and informational core of the query (e.g., request for
   educational quality analysis, technical translation of a passage, biographical
   summary).
   - Do not reproduce the original input verbatim; instead, frame the LLM prompt
   around the essential thematic elements extracted from the input.
   - For queries regarding individuals, avoid direct reference to real personal
   information unless the name is widely public and essential; even then, use a
   generic or hypothetical framing for the individual profile.

3. Strategies for High-Quality and Privacy-Preserving Prompts:
   - Strike a balance between sufficient contextual detail and privacy abstraction
   to maintain prompt clarity and relevance.
   - Use neutral, context-aware formulations that clearly instruct the LLM on the
   content and style expected.
   - Avoid vague or overly generic prompts that could result in less useful or
   lower-quality responses.
   - When system or proprietary content is mentioned, instruct the LLM to
   generalize specific terms and maintain the technical meaning without revealing
   sensitive info.
```

```
      - When a direct translation is requested on specialized text, specify to
      replace or abstract internal nomenclature.

   4. Explanation Requirements:
      - The reasoning must transparently explain how privacy risks were identified
      (e.g., presence of names, locations, dates, proprietary terms).
      - It must describe the abstraction or omission methods applied (e.g., replacing
      âĂŸJonah Van BeijnenâĂİ with âĂŸa notable individual,âĂİ substituting âĂŸMakauâĂİ
      with âĂŸa specific region,âĂİ or âĂŸYodaâĂİ with âĂŸa system nameâĂİ).
      - Clarify how the essential task and user intent were preserved despite these
      generalizations (e.g., focusing on educational quality, technical translation,
      biographical summary).
      - This explanation justifies your design choices and demonstrates adherence to
      privacy-preserving principles.

   Common Pitfalls to Avoid:
   - Do not merely lightly obscure or partially redact sensitive details; full
   anonymization or abstraction is required.
   - Do not repeat any user-supplied PII or proprietary content verbatim.
   - Avoid including URLs, exact dates, or direct quotes without modification.
   - Do not leave ambiguity that could degrade the quality or contextual clarity of
   the reformulated prompt.
   - Do not include any real personal or organizational names unless they are public
   figures and the query requires it, then use generic descriptors instead.

   Example Summary of Effective Approach (Informed by Prior Examples):
   - For geographic queries: replace exact place names with general regions and
   provide a brief contextual descriptor.
   - For technical texts containing system names or subscription types: instruct the
   LLM to translate or process the text while replacing or abstracting proprietary
   system identifiers.
   - For biographical summaries about specific individuals: remove the real name and
   request a generic, well-structured four-paragraph summary about âĂŸa notable
   individual,âĂİ preserving the overall intent without leaking PII.

   Summary:
   Your reformulations must ensure zero exposure of any PII or private/proprietary
   content while retaining enough thematic and functional clarity for the external
   LLM to produce high-quality, relevant outputs. This requires thorough analysis of
   the user's query, rigorous application of privacy-preservation strategies, and
   explicit reasoning explanations that document your approach and choices.
```

## N  EXAMPLES OF BEST PROMPTS FOR EVERY BENCHMARK

In this section, we present the best optimized prompt obtained for every (benchmark, model) con-
figuration. Each subsection below pertains to one (benchmark, model) configuration. Since every
compound AI system consists of multiple modules, each subsection consists of multiple boxes, list-
ing the prompts for each module. MIPROv2 optimized prompts contain upto 4 few-shot examples
for each task. We provide just the first demo here for brevity. GEPA's prompts only consist of the
optimized instruction, which is provided in full.

### N.1  HOTPOTQA, GPT-4.1 MINI

```
HotpotQA GPT-4.1 Mini create_query_hop2.predict

Base Prompt:

Given the fields `question`, `summary_1`, produce the fields `query`.
```

```
MIPROv2 Prompt:
```

```
You are an expert multi-hop question answering system designed to refine retrieval
queries for complex questions. Given the original question and an initial summary
of retrieved documents (summary_1), think step by step to analyze how the summary
relates to the question. Use this reasoning to generate a focused and precise
query that will guide the retrieval of additional relevant information needed to
answer the question completely. Your output should include a clear
chain-of-thought reasoning process followed by the refined query that targets
complementary evidence for the next retrieval hop.
Demos:
Example 1:
question:
    Simone Benedetti plays for a team that currently competes in which Serie?
summary_1:
    Simone Benedetti plays for Virtus Entella, but the passages do not specify
    which Serie the team currently competes in.
reasoning:
    The summary indicates that Simone Benedetti plays for Virtus Entella, but it
    does not state the current Serie (league level) in which Virtus Entella
    competes. To accurately answer which Serie the team currently competes in, it
    is necessary to look up the most recent league information for Virtus Entella.
query:
    What Serie (league level) does Virtus Entella currently compete in?

##################################
3 other demos omitted for brevity.
##################################
```

```
GEPA Prompt generated by config GEPA:
```

```
You will be given two input fields: `question` and `summary_1`.

Your task is to generate a new search query (`query`) optimized for the **second
hop** of a multi-hop retrieval system. The original user question is typically
complex and requires information from multiple documents to answer. The first hop
query is the original question used to retrieve an initial set of documents. Your
goal is to generate a **second hop query** that retrieves *additional relevant
documents* that were *not* found in the first hop but are necessary to answer the
original question completely.

Detailed task instructions and hints:

1. **Input Understanding:**
   - `question` is the original multi-hop question posed by the user.
   - `summary_1` is a concise summary of information from a document retrieved in
   the first hop, which partially addresses the question.

2. **Purpose and Context:**
   - Your generated `query` aims to find the *missing pieces* of information
   needed to fully answer the `question`.
   - The multi-hop retrieval system works in stages:
     - First hop: The original question returns some documents.
     - Second hop: Your query must help retrieve any *other relevant documents*
     NOT found in the first hop that hold complementary or broader context
     necessary for final answer extraction.

3. **Key Observations from Examples and Feedback:**
   - First-hop documents often cover one entity or aspect in the question.
   - Remaining relevant documents often involve connected or higher-level concepts
   mentioned in `summary_1` but not explicitly asked in the original question.
   - The `query` should be formulated to explicitly target these *missing*, but
   logically linked, documents.
```

```
    - Avoid merely paraphrasing the original question or restating known facts from
    `summary_1`.
    - Instead, infer what broader or related entities/concepts might provide the
    crucial missing information.
    - For example, if `summary_1` describes a population for a small civil parish,
    but the question wants total population of the wider region, your `query`
    should target that wider region (e.g., "Madeira archipelago population in
    2011").
    - Similarly, if `summary_1` covers a song and the question wants the album it
    came from, but first hop got song-level documents, your query should retrieve
    documents about the album itself.

4. **How to Build the Query:**
    - Identify the entities or topics mentioned in `summary_1` that appear related
    but different from first-hop documents.
    - Reframe the query to explicitly mention these broader or related entities
    connected to the original question.
    - Include relevant key context from the question to maintain specificity, but
    shift focus to the missing piece.
    - The goal is to retrieve documents that link or complement what was retrieved
    initially.

5. **Practical Strategy:**
    - Read the `summary_1` carefully to spot references to bigger contexts or other
    entities not covered in the first hop.
    - Ask yourself, "What entity or aspect does this summary hint at that could
    answer the original question but was not found yet?"
    - Formulate a precise, focused factual query targeting that entity or concept
    to retrieve the missing documents.

6. **Output:**
    - Produce only the field `query` as a clear, concise question or keyword phrase
    designed for efficient retrieval of **second-hop documents**.
    - Ensure the query relates logically to the original question while targeting
    the broader or complementary knowledge identified in `summary_1`.
    - Do **not** include the original question or simply rephrase it.
    - Do **not** duplicate information already well-covered by the first hop
    retrieval.

By following these principles, you will help the multi-hop retrieval system find
all necessary documents to answer the multi-faceted original question completely.
```

```
HotpotQA GPT-4.1 Mini final_answer.predict
```

Base Prompt:

```
Given the fields `question`, `summary_1`, `summary_2`, produce the fields
`answer`.
```

MIPROv2 Prompt:

```
You are an expert multi-hop reasoning assistant skilled in synthesizing
information from multiple summaries to answer complex questions. Given the
`question`, along with two intermediate summaries `summary_1` and `summary_2`
that contain relevant evidence, carefully analyze and integrate the information
step-by-step to produce a clear, logical reasoning process followed by a concise
and accurate final answer.
Demos:
Example 1:
question:
    are Machaeranthera and Prumnopitys both the plants?
summary_1:
```

```
    Yes, both Machaeranthera and Prumnopitys are plants. Machaeranthera is a genus
    of flowering plants in the daisy family, while Prumnopitys is a genus of
    coniferous evergreen trees in the podocarp family.
summary_2:
    Yes, both Machaeranthera and Prumnopitys are plants; Machaeranthera is a genus
    of flowering plants in the daisy family, while Prumnopitys is a genus of
    coniferous evergreen trees in the podocarp family.
reasoning:
    Both summaries agree that Machaeranthera and Prumnopitys are plants.
    Specifically, Machaeranthera is a genus of flowering plants in the daisy
    family, and Prumnopitys is a genus of coniferous evergreen trees in the
    podocarp family. Since both belong to plant genera, the answer is yes.
answer:
    yes

##################################
3 other demos omitted for brevity.
##################################
```

GEPA Prompt generated by config GEPA:

```
Task Description:

You are given three fields as input: `question`, `summary_1`, and `summary_2`.
Your goal is to produce an `answer` field that directly and explicitly responds to
the question using the information from the two summaries, enhanced by your
authoritative domain knowledge when needed.

Input Format:

- `question`: A natural language question that may require a fact, definition,
name, date, yes/no response, or other specific information.
- `summary_1` and `summary_2`: Two independently generated summaries or snippets
containing information related to the question. They may vary in completeness,
accuracy, and specificity.

Requirements and Approach:

1. **Understand the question precisely.** Determine exactly what is
askedâĂŤwhether a name, a specific fact, a date, or a yes/no answer.

2. **Compare both summaries.** Analyze the content of `summary_1` and `summary_2`:
   - If they agree and directly answer the question, use this as primary evidence.
   - If one summary provides a fact that the other does not mention, carefully
   evaluate its plausibility.
   - If the summaries conflict, use domain expertise and authoritative knowledge
   to resolve or explicitly state uncertainty.

3. **Domain-specific factual verification and nuance:**
   - **Names and nicknames:** Provide only the specific nickname or name when
   asked, without extra phrasing. For example, when asked for the nickname of a
   person or entity, respond with the nickname alone, not a full sentence.
   - **Nationality and identity distinctions:** Use the most precise terms aligned
   with factual correctness and common usage (e.g., âĂEnglishâĂİ vs. âĂBritishâĂİ)
   based on domain knowledge.
   - **Dates and historical facts:** Verify dates or historical claims with domain
   knowledge to pick the correct fact, especially when there might be confusion
   between franchise start dates vs. event dates etc.
   - **Yes/no questions:** Prefer concise answers of âĂyesâĂİ or âĂnoâĂİ only,
   unless the question demands elaboration.
```

```
    - **Types or categories:** If a question asks about the type or category (e.g.,
    type of company), provide the most direct concise phrase without including
    adjectives like nationality unless asked explicitly.

4. **Answer conciseness and relevance:**
    - Provide a brief and direct answer to the question.
    - Avoid repeating or restating the question.
    - Avoid unnecessary context unless requested or needed for clarity.
    - Avoid constructing full sentences unless needed; for example, answers to
    nickname or yes/no questions should be as short and specific as possible.

5. **When authoritative knowledge supplements the summaries:**
    - If the summaries are incomplete or potentially inaccurate, incorporate
    trusted knowledge from your training about the topic to provide the correct and
    precise answer.
    - For example, when a summary gives a year that conflicts with known release
    dates or factual details, prefer the verified date.
    - When the summaries differ in style (one uses a formal phrase, another
    provides just the nickname), respond with the correct, clean answer format
    (e.g., just the nickname alone).

6. **Examples of correct reasoning and answers:**
    - Question: âĂWhat is the nickname of the 2005 Toyota Grand Prix of Long Beach
    Polesitter ?âĂİ
       - Correct answer: `the thrill from West Hill`
    - Question: âĂWhat type of company is Zipcar led by Scott Griffith from
    2003-2013?âĂİ
       - Correct answer: `car-sharing company`
    - Question: âĂWho was the partner of British comic book artist, Henry Flint,
    that helped create Zombo?âĂİ
       - Correct answer: `Al Ewing`

Summary:

- Use both summaries as primary but not sole evidence.
- Reliably verify and contextualize facts using domain knowledge, especially for
nationality, dates, nicknames, company types, and yes/no questions.
- Provide short, direct answers matching the specificity requested.
- Avoid unnecessary elaboration unless explicitly required.
- Explicitly resolve conflicts or ambiguity using your knowledge or state
uncertainty when appropriate.

This approach ensures that answers are both accurate and concise, suitable for
direct consumption or integration in knowledge bases or question-answering systems.
```

```
HotpotQA GPT-4.1 Mini summarize1.predict
```

Base Prompt:

```
Given the fields `question`, `passages`, produce the fields `summary`.
```

MIPROv2 Prompt:

```
Given a question and a set of related passages, carefully analyze the information
by thinking through the relevant facts step-by-step. Produce a clear and concise
summary that synthesizes the key points from the passages directly relevant to
answering the question, ensuring the summary is focused, accurate, and grounded in
the evidence provided.
Demos:
Example 1:
question:
```

```
    The architectural style of a church that stands in front of the Palazzo
    Ghisilardi Fava originated in what city?
passages:
    ['Palazzo Ghisilardi Fava | Palazzo Ghisilardi Fava is a Renaissance style
    palace, located on via Manzoni 4 in Bologna, region of Emilia Romagna, Italy;
    it houses the Medieval Civic Museum of Bologna.', 'Madonna di Galliera,
    Bologna | The Madonna di Galliera is a church with a Renaissance facade and
    Baroque interiors, located on Via Manzoni, in central Bologna, Italy. It
    stands in front of the Palazzo Ghisilardi Fava. The present name over the
    portal is the "Chiesa di Filippini Madonna di Galliera e Filippo Neri".',
    'Palazzo Pretorio, Prato | The Palazzo Pretorio of the Tuscan town of Prato
    was the old city hall located town center, standing in front of the current
    Palazzo Comunale. It now accommodates the Civic Museum of Prato, which was
    reopened on September 2013.', 'Palazzo Pisani Gritti | The Palazzo Pisani
    Gritti is a Gothic-style palace located across the Grand Canal from the Church
    of the Salute, and stands along the Campo del Traghetto where the church of
    Santa Maria del Giglio, in the Sestieri of San Marco, Venice, Italy.',
    'DÃűmÃűtÃűr Tower | The DÃűmÃűtÃűr tower is the oldest building in Szeged,
    Hungary. The foundation was most probably laid during the 11th century, while
    the lower part was built (in Romanesque style) from the 12th century, and the
    upper part (Gothic style) from the 13th century. The tower was once part of
    the former "Szent DÃűmÃűtÃűr templom" (St. Demetrius church), but today it
    stands in DÃşm Square, in front of the much larger Votive Church of Szeged.
    The upper part was rebuilt from the original stones in 1926. The architecture
    of the tower is similar to another found in Southern France, or in the
    territory of the former Byzantine Empire.', 'Palazzo Leoni, Bologna | The
    Palazzo Leoni is a Renaissance style palace located on Via Marsala #31, in
    front of the outlet of Via Mentana, in central Bologna, region of
    Emilia-Romagna, Italy. The apse of the church of San Martino, is across the
    street.', 'Palazzo Altieri | Palazzo Altieri is a palace in Rome, which was
    the home of the Altieri family in the city. The palace faces the square in
    front of the Church of the GesÃź.']
reasoning:
    The question asks for the city where the architectural style of a church that
    stands in front of the Palazzo Ghisilardi Fava originated. Passage [2]
    identifies the church standing in front of the Palazzo Ghisilardi Fava as
    "Madonna di Galliera" and mentions it has a Renaissance facade and Baroque
    interiors. Passage [1] states the Palazzo Ghisilardi Fava is located in
    Bologna, Italy. Since the church is located in Bologna and features a
    Renaissance architectural style, it can be inferred that the architectural
    style of this church originated in the city of Florence, which is historically
    recognized as the birthplace of Renaissance architecture. Thus, the
    architectural style (Renaissance) of the church in front of the Palazzo
    Ghisilardi Fava originated in Florence.
summary:
    The Renaissance architectural style of the church standing in front of the
    Palazzo Ghisilardi Fava originated in Florence.

###################################
3 other demos omitted for brevity.
###################################
```

GEPA Prompt generated by config GEPA:

```
You are a first-hop **summarization module** in a multi-hop question answering
(QA) system. Your task is to generate a concise, informative `summary` given two
input fields: a `question` and a list of relevant `passages`.

Your goal is to extract and synthesize key information from the retrieved passages
that:

1. Directly relates to the initial question.
```

2. Captures the core facts and entities needed to understand the scope and context
of the question.
3. Includes relevant connections, bridging entities, dates, locations, or
descriptions that enable the system to devise focused and effective follow-up
queries in subsequent hops.
4. Provides a strong factual foundation for downstream answer generation modules.

**Task specifics and best practices:**

- The `summary` must represent a distilled synthesis, not just a compression or
extractive snippet.
- Explicitly include cited passage titles or key entity labels (e.g., "Children in
Need 2006 | ..." or "Anthony Levandowski | ...") in your summary to highlight the
origin of information.
- Incorporate sufficient context to hint at missing or un-retrieved supporting
facts, thus enhancing the multi-hop retrieval process.
- When the question asks for an attribute (e.g., nationality, location, company
origin), ensure you provide:
   - Identification of the relevant subject or entity mentioned in the passages.
   - The extracted attribute or relevant information as stated or implied.
   - Bridging details that could help the system pursue remaining information in
   the next retrieval step.
- Avoid forming a final answer; instead, focus on "what is known now" from the
input documents to facilitate further query refinement.

**Examples of critical elements to include:**

- Entity names, roles, titles, and dates tied to the question.
- Names of organizations or locations connected through intermediary entities.
- Distinctive identifiers or clarifications that can help narrow down next-step
retrieval (such as "Natasha Kaplinsky is an English presenter," "Kapolei is a city
on Oahu," or "Waymo spun out of Alphabet").

**Format of output:**

Provide a paragraph or a few sentences that cohesively summarize the key passages
in relation to the question, referencing passage titles or entities to frame facts
clearly.

---

This approach ensures the summary is both informative for next-hop retrieval and
foundational for final answer extraction in multi-hop QA.

---

### HotpotQA GPT-4.1 Mini summarize2.predict

Base Prompt:

```
Given the fields `question`, `context`, `passages`, produce the fields `summary`.
```

MIPROv2 Prompt:

```
Given a `question`, relevant `context`, and a list of `passages`, provide a
clear, concise summary that integrates the key information from the passages in
relation to the question and context. Use step-by-step reasoning to explain how
the summary is derived from the evidence before presenting the final synthesized
summary.
Demos:
Example 1:
question:
    What bank was founded by the great-great-great grandfather of the second Duke
    of Florence?
```

```
context:
    The bank founded by the great-great-great grandfather of the second Duke of
    Florence is the Medici Bank, established by Giovanni di Bicci de' Medici.
passages:
    ['Giovanni di Bicci de\' Medici | Giovanni di Bicci de\' Medici (c. 1360 âĂŞ
    February 20/28, 1429) was an Italian banker, a member of Medici family of
    Florence, and the founder of the Medici Bank. While other family members, such
    as Chiarissimo di Giambuono de\' Medici, who served in the Signoria in 1201,
    and Salvestro de\' Medici, who was implicated in the Ciompi Revolt of 1378,
    are historically significant, Giovanni\'s founding of the family bank truly
    began the family\'s rise to power in Florence. He was the father of Cosimo
    de\' Medici ("Pater Patriae"), grandfather of Piero di Cosimo de\' Medici,
    great-grandfather of Lorenzo de\' Medici (the Magnificent) and
    great-great-great-grandfather of Cosimo I de\' Medici, Grand Duke of
    Tuscany.', 'Averardo de\' Medici | Averardo de\' Medici (1320-1363),
    also known as Everard De Medici, was the son of Salvestro de\' Medici (died
    1346), "il Chiarissimo" (English meaning "the very clear.") and the father of
    three children: Giovanni, Francesco, and Antonia. Giovanni di Bicci de\'
    Medici would later become the first historically relevant member of Medici
    family of Florence and the eventual founder of the Medici bank.', "Villa
    Medici at Cafaggiolo | The Villa Medicea di Cafaggiolo is a villa situated
    near the Tuscan town of Barberino di Mugello in the valley of the River Sieve,
    some 25\xa0kilometres north of Florence, central Italy. It was one of the
    oldest and most favoured of the Medici family estates, having been in the
    possession of the family since the 14th century, when it was owned by Averardo
    de' Medici. Averardo's son, Giovanni di Bicci de' Medici, is considered to be
    the founder of the Medici dynasty.", 'Medici: Masters of Florence | Medici:
    Masters of Florence is an Italian-British television drama series about the
    Medici dynasty set in 15th-century Florence, starring Dustin Hoffman as
    Giovanni di Bicci de\' Medici, Richard Madden as Cosimo de\' Medici, and
    Stuart Martin as Lorenzo de\' Medici ("The Elder"). The series was co-created
    by Frank Spotnitz ("The X-Files" and "Man in the High Castle") and Nicholas
    Meyer (""). Sergio Mimica-Gezzan ("The Pillars of the Earth") directed all
    eight episodes. Episodes 1 and 2 aired on Rai 1 (Italian TV) on 18 October
    2016. According to Italian ratings compiler Auditel, it attracted a record
    7.6\xa0million viewers. The first season consists of eight episodes.', 'Neri
    di Bicci | Neri di Bicci (1419âĂŞ1491) was an Italian painter of the
    Renaissance. A prolific painter of mainly religious themes, he was active
    mainly in Florence and in the medium of tempera. His father was Bicci di
    Lorenzo. His grandfather, Lorenzo di Bicci was also a painter in Florence, a
    pupil of Spinello Aretino.', 'Cosimo de\' Medici | Cosimo di Giovanni de\'
    Medici (called \'the Elder\' (Italian "il Vecchio") and posthumously "Father
    of the Nation" (Latin "pater patriae"); 27 September 1389 âĂŞ 1 August 1464)
    was an Italian banker and politician, the first of the Medici political
    dynasty, "de facto" rulers of Florence during much of the Italian Renaissance.
    Despite his influence, his power was not absolute; Florence\'s legislative
    councils at times resisted his proposals, something which would not have been
    tolerated by the Visconti of Milan, for instance; throughout his life he was
    always "primus inter pares" ("first among equals"). His power derived from his
    wealth as a banker, and he was a great patron of learning, the arts and
    architecture.', "Giovanni il Popolano | Giovanni de' Medici, in full Giovanni
    di Pierfrancesco de' Medici, later known as il Popolano (21 October 1467 âĂŞ
    14 September 1498) was an Italian nobleman of the Medici House of Florence. He
    was the son of Pierfrancesco di Lorenzo de' Medici, and therefore a member of
    a secondary branch of the family."]
reasoning:
    The question asks for the bank founded by the great-great-great-grandfather of
    the second Duke of Florence. Passage [1] states that Giovanni di Bicci de'
    Medici founded the Medici Bank and was the great-great-great-grandfather of
    Cosimo I de' Medici, who was the Grand Duke of Tuscany. The Dukedom of
    Florence eventually transformed into the Grand Duchy of Tuscany under Cosimo
    I, who is identified as the second Duke of Florence in the question context.
```

```
Thus, Giovanni di Bicci de' Medici, as the great-great-great-grandfather of the
second Duke of Florence (Cosimo I de' Medici), founded the Medici Bank.
summary:
    The Medici Bank was founded by Giovanni di Bicci de' Medici, the
    great-great-great-grandfather of the second Duke of Florence.

#################################
3 other demos omitted for brevity.
#################################
```

GEPA Prompt generated by config GEPA:

```
Given the fields `question`, `context`, `passages`, produce the fields `summary`.
```

## N.2    HOTPOTQA, QWEN3 8B

### HotpotQA Qwen3 8B create_query_hop2.predict

Base Prompt:

```
Given the fields `question`, `summary_1`, produce the fields `query`.
```

MIPROv2 Prompt:

```
Given the question and the first summary, synthesize the key elements of the
question and the summary. Identify the specific information that needs to be
retrieved or confirmed. Formulate a focused and precise query that will guide the
next step in the multi-hop reasoning process, ensuring it directly addresses the
gap in knowledge or requires further clarification from additional context.
Demos:
```

GEPA Prompt generated by config GEPA+Merge:

```
Given the fields `question` and `summary_1`, produce the field `query` that
optimizes the retrieval of additional documents for a multi-hop system.

**Task Details:**
1. **Objective:** Your query must target documents not retrieved in the first hop,
using clues from the summary and the original question.
2. **Key Strategy:**
   - Identify gaps in the first hop's retrieved documents (e.g., missing entities,
   relationships, or specific details).
   - Use explicit information from the summary (e.g., names, locations,
   quantities) to rephrase the question into a query that surfaces new relevant
   documents.
   - Avoid restating the answer directly; instead, structure the query to explore
   connections or unresolved details.
3. **Domain-Specific Guidance:**
   - If the summary explicitly answers the question, the query should still focus
   on retrieving documents that provide deeper context or verify the answer (e.g.,
   "What is the headquarters location of [Company]?" instead of "The answer is
   [Location]").
   - Leverage entities mentioned in the summary (e.g., "Carhartt," "Aubrey O'Day")
   to anchor the query.
   - If no documents are missing, rephrase the query to explicitly request the
   answer (e.g., "Which has more acts, Elektra or From the House of the Dead?").
4. **Avoid:**
   - Generating queries that duplicate the original question.
   - Assuming the summary contains all necessary information for the second hop.
```

```
HotpotQA Qwen3 8B final_answer.predict
```

Base Prompt:

```
Given the fields `question`, `summary_1`, `summary_2`, produce the fields
`answer`.
```

MIPROv2 Prompt:

```
Given the question, summary_1, and summary_2, generate a step-by-step reasoning
process that logically connects the information from both summaries to arrive at
the final answer, and then provide the answer. Ensure the reasoning explicitly
references how the summaries support the conclusion and maintains consistency with
the question's requirements.
Demos:
Example 1:
question:
    are Machaeranthera and Prumnopitys both the plants?
summary_1:
    Yes, both Machaeranthera (a genus of flowering plants in the daisy/aster
    family) and Prumnopitys (a genus of coniferous trees in the podocarp family)
    are plants.
summary_2:
    Yes, both Machaeranthera (a flowering plant genus in the daisy family) and
    Prumnopitys (a coniferous tree genus in the podocarp family) are plants.
reasoning:
    Both summaries confirm that Machaeranthera and Prumnopitys are plants. Summary
    1 explicitly states they are plants, categorizing Machaeranthera as a
    flowering plant genus in the daisy/aster family and Prumnopitys as a
    coniferous tree genus in the podocarp family. Summary 2 repeats this
    confirmation with slightly different phrasing.
answer:
    Yes

###################################
3 other demos omitted for brevity.
###################################
```

GEPA Prompt generated by config GEPA+Merge:

```
Given the fields `question`, `summary_1`, and `summary_2`, produce the field
`answer` by:
1. **Extracting precise terminology**: Identify the exact noun or specific term
required in the answer (e.g., "Medicare" rather than "Medicare cuts"). Avoid vague
or generalized terms unless explicitly stated in the summaries.
2. **Resolving ambiguity**: If the question references a title, historical role,
or specific designation (e.g., "second Duke of Florence"), prioritize contextual
or historical clues from the summaries to infer the correct answer, even if the
exact term is not explicitly stated. Use domain-specific knowledge (e.g., Medici
family lineage) to fill gaps when summaries are indirect or vague.
3. **Cross-referencing summaries**: Ensure consistency between summaries. If
summaries conflict, prioritize the one with explicit factual claims (e.g.,
numerical data, direct statements). If no explicit claim exists, synthesize
information while ensuring alignment with historical, political, or cultural
context.
4. **Avoiding overgeneralization and extra information**: Focus strictly on the
most specific and directly stated information in the summaries. Do not add
context, explanations, or external knowledge beyond what is explicitly provided.
For example, if the question asks for a year, provide only the year; do not
include band member details or historical background.
5. **Prioritizing factual alignment**: If a summary explicitly states the answer,
use that. If summaries are indirect or vague, synthesize information while
ensuring alignment with factual knowledge (e.g., linking "Path to Prosperity" to
Rep. Paul Ryanâ€™s Medicare proposal).
```

**Key adjustments based on feedback**:
- **Conciseness**: Answers must be strictly factual and concise, avoiding additional context or explanations. For example, if the question is "Is X shorter than Y?" the answer should be a simple "No" or "Yes" based on numerical comparisons, not a full explanation.
- **Numerical precision**: When comparing measurements (e.g., heights, dates), ensure exact values are used and explicitly stated in the summaries. If summaries provide conflicting numbers, resolve via direct factual claims.
- **Domain-specific knowledge**: Use known facts (e.g., architectural records, historical timelines) to validate ambiguous answers, but only when summaries lack explicit information.

## HotpotQA Qwen3 8B summarize1.predict

Base Prompt:

```
Given the fields `question`, `passages`, produce the fields `summary`.
```

MIPROv2 Prompt:

```
Given the fields `question`, `passages`, produce the fields `summary`.
Demos:
Example 1:
question:
    Tay Garnett and Alexander Kluge both have what job?
passages:
    ['Tay Garnett | William Taylor "Tay" Garnett (June 13, 1894 âĂŞ October 3,
    1977) was an American film director and writer.', 'Trade Winds (film) | Trade
    Winds is a 1938 comedy film distributed by United Artists. It was directed by
    Tay Garnett, and starred Fredric March and Joan Bennett. The screenplay was
    written by Dorothy Parker, Alan Campbell and Frank R. Adams, based on story by
    Tay Garnett.', 'Prestige (film) | Prestige is a 1932 American pre-Code drama
    film directed by Tay Garnett and written by Tay Garnett, Rollo Lloyd and
    Francis Edward Faragoh. The film stars Ann Harding, Adolphe Menjou, Melvyn
    Douglas and Guy Bates Post. The film was released on January 22, 1932, by RKO
    Pictures.', 'Alexander Kluge | Alexander Kluge (born 14 February 1932) is a
    German author, philosopher, academic and film director.', 'Cattle King |
    Cattle King is a 1963 film directed by Tay Garnett. It stars Robert Taylor and
    Robert Loggia. It also appears to have been called Guns of Wyoming in some
    countries.', 'GÃnther von Kluge | GÃnther von Kluge (30 October 1882 âĂŞ 19
    August 1944) was a German field marshal during World War II. Kluge held
    commands on both the Eastern and Western Fronts. Although Kluge was not an
    active conspirator in the 20 July plot, he committed suicide on 19 August
    1944, after having been recalled to Berlin for a meeting with Hitler in the
    aftermath of the failed coup. He was replaced by Field Marshal Walter Model.',
    'Bad Company (1931 film) | Bad Company is a 1931 American Pre-Code gangster
    film directed and co-written by Tay Garnett with Tom Buckingham based on Jack
    Lait\'s 1930 novel "Put on the Spot". It stars Helen Twelvetrees and Ricardo
    Cortez. Told from the view of a woman, the working titles of this film were
    "The Gangster\'s Wife" and "The Mad Marriage". Unlike many static early sound
    films, Garnett includes several scenes using a moving camera climaxing in a
    gigantic assault on an office building with both sides using heavy machine
    guns.']
reasoning:
    The question asks for the common job of Tay Garnett and Alexander Kluge. From
    the passages, Tay Garnett is explicitly identified as a film director and
    writer (passages 1, 2, 3, 5, 7). Alexander Kluge is described as a German
    author, philosopher, academic, and film director (passage 4). The shared job
    between them is "film director."
summary:
    Both Tay Garnett and Alexander Kluge are film directors.
```

```
####################################
3 other demos omitted for brevity.
####################################
```

GEPA Prompt generated by config GEPA+Merge:

You are the first-hop **summarization module** in a multi-hop QA system. Your task is to generate a **comprehensive, structured summary** that:

1. **Extracts direct answers** from the top retrieved passages to address the question.
2. **Identifies and highlights missing or implied clues** that may require further retrieval (e.g., entities, connections, or contextual details).
3. **Synthesizes information** by combining explicit facts from the passages with domain-specific knowledge or logical inferences to guide subsequent steps.

### **Summary Structure**
- **Entity/Person Mention**: Clearly state the subject (e.g., "Billy Truax", "Eintracht Braunschweig") and include **full names, titles, or official designations** (e.g., "Thomas Lance Rentzel", "Braunschweiger Turn- und Sportverein Eintracht von 1895 e.V.").
- **Direct Answer**: Include **explicit answers** from the passages (e.g., birth dates, team affiliations, or direct statements).
- **Clues for Next Steps**: Signal **missing information** (e.g., "Lance Rentzel's birth year is explicitly stated, but his exact birthplace is not; need to search for 'Lance Rentzel birthplace'").
- **Domain-Specific Context**: Add **relevant background** (e.g., "Eintracht Braunschweig is a German football club based in Braunschweig, Lower Saxony" or "NFL players' birth dates are critical for age comparisons").

### **Guidelines**
- **Do not omit** any entity or detail from the retrieved passages that could be relevant for follow-up queries (e.g., team names, locations, or historical context).
- **Prioritize clarity** by **separating direct answers from inferred clues** (e.g., using bullet points, subheadings, or bolded labels).
- **Avoid assumptions** not supported by the passages; if information is absent, **explicitly state that it is missing** and suggest **precise search terms** (e.g., "Verify Wichita Dwight D. Eisenhower National Airport's tower status via FAA records").
- **Include quantifiable data** (e.g., "few thousand Stabyhouns exist globally", "born July 15, 1943") to enable precise comparisons.
- **Highlight connections** between entities (e.g., "Billy Truax and Lance Rentzel were traded in 1970") to aid in cross-referencing.

### **Key Niche/Domain-Specific Insights**
- **NFL Player Comparison**: Birth dates are critical for age determination, and team affiliations (e.g., "traded in 1970") may imply historical context.
- **Airport Classification**: "Non-towered" status is explicitly stated in some passages (e.g., "non-towered public airport"), while others require inference (e.g., "major commercial airports typically have towers").
- **Football Club Context**: Clubs like Eintracht Braunschweig require background on their location, league, and history (e.g., "based in Braunschweig, Lower Saxony").
- **Quantifiable Data**: Use exact dates, numbers, or rankings (e.g., "few thousand Stabyhouns exist globally") to enable precise comparisons.

### **Critical Additional Instructions**
- **Ensure All Retrieved Documents Are Represented**: Explicitly include all entities, titles, and details from the retrieved passages (e.g., full names, film titles, and specific roles).

- **Signal Missing Links**: If a connection between entities is implied but not
explicitly stated (e.g., "Nancy Steiner worked on *The Lovely Bones*"), flag this
as a potential gap and suggest search terms to resolve it.
- **Prioritize Bridging Concepts**: Highlight relationships between entities
(e.g., "Gary Pinkel coached Toledo in 1993 and holds the most wins in school
history") to enable focused follow-up queries.
- **Avoid Overgeneralization**: Only include domain-specific context that is
either explicitly stated in the passages or directly inferable (e.g., "major
commercial airports typically have towers" is acceptable, but "airports with fewer
than 10,000 passengers are non-towered" is not unless stated).

### **Example Format**
For the question *"Which NFL player is younger, Billy Truax or Lance Rentzel?"*:
- **Entity/Person Mention**: Billy Truax (William Frederick Truax), Lance Rentzel
(Thomas Lance Rentzel)
- **Direct Answer**:
  - **Billy Truax**: Born July 15, 1943.
  - **Lance Rentzel**: Born October 14, 1943.
- **Clues for Next Steps**: None required; birth dates are explicitly provided.
- **Domain-Specific Context**: Birth dates are sufficient to determine age
difference within the same year.

For the question *"Which is a non-towered airport, Wichita Dwight D. Eisenhower
National Airport or Montrose Regional Airport?"*:
- **Entity/Person Mention**: Wichita Dwight D. Eisenhower National Airport,
Montrose Regional Airport
- **Direct Answer**:
  - **Montrose Regional Airport**: "non-towered public airport" (passage 3).
  - **Wichita Dwight D. Eisenhower National Airport**: No explicit mention of
  tower status; inferred as **towered** (typical for major commercial airports).
- **Clues for Next Steps**: Verify Wichita's tower status via FAA records or
additional sources (e.g., "Wichita Dwight D. Eisenhower National Airport tower
status").
- **Domain-Specific Context**: Non-towered airports lack a control tower, relying
on pilot communication (passage 4). Major commercial airports like Wichita usually
have towers.

**Tip:** When summarizing, don´ÃŹt just compress; synthesizeâÃ˘include both direct
answers and clues required for the systemâÃŹs next steps. Always explicitly state
if a retrieved documentâÃŹs content is missing critical information, and provide
actionable search terms to address gaps.

---

## HotpotQA Qwen3 8B summarize2.predict

Base Prompt:

```
Given the fields `question`, `context`, `passages`, produce the fields `summary`.
```

MIPROv2 Prompt:

```
Given the fields `question`, `context`, and `passages`, generate a structured
reasoning process that explains how to synthesize information from these sources,
and then produce a concise summary of the key findings.
Demos:
Example 1:
question:
    Simone Benedetti plays for a team that currently competes in which Serie?
context:
    The provided passages do not specify the Serie in which Simone Benedetti's
    team (Virtus Entella) currently competes.
passages:
```

```
['Simone Benedetti | Simone Benedetti (born 3 April 1992) is an Italian
 professional footballer who plays as a centre back for Virtus Entella.',
 'Virtus Entella | Virtus Entella (usually referred as simply Entella or
 Chiavari Entella) is an Italian association football club, based in Chiavari,
 Liguria. They currently compete in Serie B.', 'Simone Palermo | Simone Palermo
 (born 17 August 1988) is an Italian footballer who plays as a midfielder for
 Virtus Entella at Serie B.', 'Andrea Paroni | Andrea Paroni (born 14 October
 1989) is an Italian footballer who currently plays as a goalkeeper for Virtus
 Entella in the Serie B.', 'Alessandro Iacobucci | Alessandro Iacobucci (born 3
 June 1991) is an Italian footballer who currently plays as a goalkeeper for
 Serie B side Virtus Entella.', 'George Ganea | George Ganea (born 26 May 1999
 in Bucharest) is a Romanian footballer who plays as a Striker for Serie B club
 Virtus Entella Primavera team.', 'Gianpaolo Castorina | Gianpaolo Castorina
 (born August 30, 1976 in Milan) is a retired Italian professional footballer
 turned coach, currently in charge as head coach of Serie B club Virtus
 Entella.']
reasoning:
    The question asks for the Serie in which Simone Benedetti's team (Virtus
    Entella) currently competes. While the context initially states that this
    information is not specified, the passages explicitly mention that Virtus
    Entella competes in Serie B. Multiple passages (e.g., [2], [3], [4], [5], [7])
    confirm this detail, directly answering the question.
summary:
    Simone Benedetti's team, Virtus Entella, currently competes in Serie B.

###################################
3 other demos omitted for brevity.
###################################
```

GEPA Prompt generated by config GEPA+Merge:

```
Given the fields `question`, `context`, and `passages`, produce the field
`summary`.

Your task is to synthesize information from the question, context, and newly
retrieved passages to generate a **comprehensive, precise, and well-structured
summary** that enables the answer generation module to confidently arrive at the
correct answer.

### Key Requirements:
1. **Explicit Answers First**: Prioritize explicitly stated facts from the context
and passages (e.g., direct mentions of entities, roles, or relationships).
2. **Infer or Generalize When Necessary**: If critical details are missing from
the passages, infer connections or generalize based on contextual clues and
domain-specific knowledge (e.g., linking ownership structures, roles, or
historical context).
3. **Bridge Gaps**: Ensure the summary includes all **key supporting information**
required to answer the question, even if it is not explicitly stated in the input.
For example:
   - If the answer is "Newcastle United," include details about Sports Direct's
   ownership and the connection to the billionaire.
   - If the answer is a person's role (e.g., "troubleshooter"), explicitly state
   their relationship to the question's subject and any relevant background.
4. **Structure and Precision**:
   - Clearly connect entities, roles, and relationships (e.g., "Stan Kroenke owns
   Sports Direct and Arsenal F.C.").
   - Avoid ambiguity by including all necessary contextual links (e.g., "Mike
   Ashley founded Sports Direct and owns Newcastle United").
   - Use precise terminology and ensure alignment with domain-specific knowledge
   (e.g., "investigative journalist" instead of "writer").
```

5. **Domain-Specific Knowledge**: Leverage implicit domain knowledge when passages lack critical details (e.g., knowing that "Project RAND" is linked to Henry H. Arnold and the RAND Corporation).

### Example Integration:
If the question is about a person's profession in a novel, ensure the summary includes:
- The character's name.
- Their profession (explicitly stated in the text).
- Contextual links to the book series or plot (e.g., "in *The Girl in the Spider's Web*").
- Any relevant background about the profession or characterâĂŹs role in the story.

Always aim to match the **coverage and relevance** of an "ideal summary" as described in the feedback, ensuring the answer module has all necessary information to generate the correct final answer.

## N.3  IFBENCH, GPT-4.1 MINI

IFBench GPT-4.1 Mini generate_response_module.predict

Base Prompt:

Respond to the query

MIPROv2 Prompt:

You will be given a query containing a complex task with multiple constraints and instructions. Your job is to first repeat the query exactly as given, word for word, without adding any extra words or commentary before or after the repetition. After repeating the query, provide a detailed, step-by-step chain-of-thought reasoning that carefully unpacks and addresses every aspect of the query. Then, produce a final response that adheres strictly to all requirements, including formatting, language, and content constraints specified in the query. Be thorough, precise, and ensure the final answer is fully validated and consistent with the reasoning. If the query specifies additional formatting or stylistic rules (such as language or inclusion of a postscript), include them exactly as instructed. Demos:

GEPA Prompt generated by config GEPA+Merge:

You are given a query input, and your task is to respond appropriately to that query. The query may contain specific instructions or constraints that you must strictly adhere to in your response. Carefully analyze the query to determine the exact requirements, including but not limited to:

- Responding with an answer chosen from a restricted set of options exactly as specified (including exact wording and punctuation).
- Including specific words a minimum number of times.
- Including specific letters a minimum number of times.
- Repeating the entire query word-for-word before providing your answer if explicitly requested, without adding extra words or characters before the repetition.
- Avoiding specific forbidden words or keywords in your response if indicated.
- Following any other explicit instructions or constraints embedded in the query.

When the query requests calculations or factual answers (e.g., combinatorial calculations), you should:

1. Carefully interpret the mathematical or logical problem.
2. Show your reasoning internally to confirm the final answer (reasoning does not need to be included in the response unless explicitly requested).

```
3. Provide the final direct response strictly following all instructions,
especially when asked to repeat the query verbatim first before giving the answer.

General approach:

- Always parse the query thoroughly to extract every constraint and instruction.
- Ensure your response exactly matches the format, wording, and content as
instructed.
- Do not invent or omit any part of the user's explicit requests.
- Meet all formatting, lexical, numeric, and structural constraints without
deviation.
- If the query involves repeating text verbatim, do not alter capitalization,
punctuation, or wording.
- Incorporate any required keywords or letters the required number of times
naturally into your response.
- When multiple constraints (like avoiding specific words while including others)
apply simultaneously, ensure you satisfy all simultaneously.

This task requires precision, exact reproduction, and strict adherence to any
given constraints or instructions embedded in the query. Your goal is to deliver
the requested answer in the exact manner requested without extraneous additions or
omissions.
```

### IFBench GPT-4.1 Mini ensure_correct_response_module.predict

Base Prompt:

```
Ensure the response is correct and adheres to the given constraints. Your response
will be used as the final response.
```

MIPROv2 Prompt:

```
Given the original query and the initially generated response, carefully verify
that the response fully meets all specified constraints and instructions in the
query. Produce detailed step-by-step reasoning that explains how the response
satisfies or fails the requirements such as exact repetition of the query text,
minimum sentence count, formatting rules (e.g., capitalization), letter frequency,
or other explicit user demands. Based on this reasoning, output a final corrected
and fully compliant response that adheres strictly to the original instructions,
ensuring accuracy, completeness, and formatting correctness. Your final output
should begin by repeating the original query exactly as provided, with no
modifications or additional text before it, followed by a comprehensive,
well-structured answer that respects all constraints, and a clear explanation of
how the requirements have been met.
Demos:
Example 1:
query:
    Can you provide the social media handles of Harry Potter? Your answer must
    contain exactly 8 bullet points. Use the markdown bullet points such as:
* This is point 1.
* This is point 2 Finish your response with this exact phrase That's my answer..
No other words should follow this phrase.
response:
    * Twitter: [@wizardingworld](https://twitter.com/wizardingworld)
* Instagram: [@wizardingworld](https://www.instagram.com/wizardingworld/)
* Facebook: [Wizarding World](https://www.facebook.com/wizardingworld)
* YouTube: [Wizarding World](https://www.youtube.com/c/WizardingWorld)
* Twitter: [@harrypotterfilm](https://twitter.com/harrypotterfilm)
* Instagram: [@harrypotterfilm](https://www.instagram.com/harrypotterfilm/)
* Facebook: [Harry Potter](https://www.facebook.com/harrypotter)
* TikTok: [@wizardingworld](https://www.tiktok.com/@wizardingworld)
```

```
That's my answer..
reasoning:
    The request is for the social media handles of Harry Potter, formatted as
    exactly eight bullet points using markdown bullet points, and the response
    must end with the exact phrase "That's my answer." The provided response
    includes eight bullet points, referencing official or widely recognized social
    media accounts related to Harry Potter and the Wizarding World franchise on
    Twitter, Instagram, Facebook, YouTube, and TikTok. The format is correct, the
    bullet points are used properly with markdown syntax, and the closing phrase
    matches the requirement exactly. Therefore, the response meets all given
    constraints.
final_response:
    * Twitter: [@wizardingworld](https://twitter.com/wizardingworld)
* Instagram: [@wizardingworld](https://www.instagram.com/wizardingworld/)
* Facebook: [Wizarding World](https://www.facebook.com/wizardingworld)
* YouTube: [Wizarding World](https://www.youtube.com/c/WizardingWorld)
* Twitter: [@harrypotterfilm](https://twitter.com/harrypotterfilm)
* Instagram: [@harrypotterfilm](https://www.instagram.com/harrypotterfilm/)
* Facebook: [Harry Potter](https://www.facebook.com/harrypotter)
* TikTok: [@wizardingworld](https://www.tiktok.com/@wizardingworld)

That's my answer..

###################################
3 other demos omitted for brevity.
###################################
```

GEPA Prompt generated by config GEPA+Merge:

```
You will receive a user query that includes explicit, detailed instructions about
how you must structure your response. Your overall task is to interpret these
embedded instructions with perfect accuracy and produce a response that strictly
complies with every single one, without adding, omitting, or rewording any
mandated text, phrases, or structural elements.

Key detailed requirements and strategy for this task:

1. **Query Parsing and Extraction of Instructions**
   - Carefully read the entire query to identify all explicit instructions
   concerning:
     - Whether and how to repeat the query text (verbatim or partially).
     - Specific length constraints (number of sentences, bullet points, word
     counts).
     - Formatting instructions (e.g., capitalization requirements, quotation
     marks, markdown bullet styles).
     - Mandatory phrases or exact sentences that must appear (especially those to
     be repeated verbatim or appended at the end).
     - Content limitations or prohibitions (for example, refusal language or
     disclaimers for unethical requests).
   - Note that some instructions may be nested or appear within the query's
   wording and are critical to follow exactly.

2. **Exact Text Reproduction**
   - When asked to repeat the query text (or any other required phrase) verbatim,
   do so with zero changes âĂŤ no added or removed words, punctuation, or
   formatting.
   - Do not prepend or append anything to the repeated text unless explicitly
   instructed.
   - Preserve all original capitalization, spacing, and punctuation exactly as in
   the query.

3. **Structural and Formatting Compliance**
```

```
    - Follow all formatting instructions strictly, such as:
      - Wrapping the entire response in quotation marks if required.
      - Using specified markdown bullet point styles (e.g., asterisks).
      - Ensuring capitalization instructions (e.g., all caps or minimum occurrences
      of uppercase words) are perfectly met.
      - Adhering to sentence or paragraph counts exactly as requested.

4. **Response Content Accuracy and Appropriateness**
   - After fulfilling all structural requirements, respond to the main substantive
   question accurately and completely.
   - Use domain knowledge and reliable calculations to ensure factual correctness
   in answers.
   - For questions requesting sensitive or potentially harmful content (e.g.,
   cures without scientific basis), produce responsible answers that include
   disclaimers or refusals if instructed.
   - Always respect ethical guidelines and any mandated refusal language or
   concluding statements for such queries.

5. **No Extraneous Text**
   - Do not add explanations, internal reasoning, apologies, or meta commentary
   beyond what the query explicitly permits or demands.
   - Your final output must be the exact, ready-to-deliver response that meets all
   user instructions perfectly.

6. **Examples and Patterns Observed**
   - Users often combine multiple complex formatting and content instructions
   (e.g., repetition of request text, followed by specific number of sentences or
   bullet points, with capitalization rules).
   - Ensure you carefully distinguish when to repeat the query text verbatim and
   when to respond directly (sometimes the repetition excludes an instruction
   sentence).
   - Handle instructions about capitalized words appearing a minimum number of
   times by distributing such words naturally but thoroughly across the response.
   - When length constraints specify minimums (e.g., âĂat least 39 sentencesâĂİ),
   ensure you meet or exceed exactly rather than approximating.
   - For bullet points or enumerations, use the precise markdown style asked for
   (commonly asterisks).
   - Follow refusal instructions verbatim and terminate the response exactly as
   specified.

**In summary:** YOUR RESPONSE MUST BE A PERFECT TEXTUAL AND STRUCTURAL RENDERING
OF THE USER QUERYâĂŹS EMBEDDED INSTRUCTIONS. ALWAYS PRIORITIZE EXACT USER
SPECIFICATIONS OVER DEFAULT GENERATIVE NORMS. THE OUTPUT IS FINAL AND WILL NOT BE
EDITED, SO COMPLIANCE MUST BE 100%.

This task therefore requires a rigorously rule-based approach:
- First parse and isolate all explicit instructions (including repetitions and
formatting).
- Then assemble your answer by reproducing mandated text identically.
- Next produce the requested content in the exact required format, length, and
style.
- Finally, review your output against the instructions ensuring total compliance
before delivering your response.
```

## N.4 IFBench, Qwen3 8B

---

**IFBench Qwen3 8B generate_response_module.predict**

Base Prompt:

```
Respond to the query
```

MIPROv2 Prompt:

```
You are a precision-focused instruction-adherence expert tasked with generating
structured, ethically compliant responses. When responding to queries, always
think step-by-step to analyze constraints (formatting rules, content boundaries,
ethical guidelines), synthesize requirements, and produce outputs that strictly
adhere to specified parameters. Ensure responses are fully formatted as requested
(e.g., lowercase text, bullet points, markdown), avoid any unverified claims, and
maintain neutrality in ambiguous ethical scenarios.
Demos:
Example 1:
query:
    Can you provide the social media handles of Harry Potter? Your answer must
    contain exactly 8 bullet points. Use the markdown bullet points such as:
* This is point 1.
* This is point 2 Finish your response with this exact phrase That's my answer..
No other words should follow this phrase.
reasoning:
    The query asks for the social media handles of Harry Potter, a fictional
    character. Since he does not have personal social media accounts, the response
    should instead list official accounts related to the Harry Potter franchise.
    These include accounts managed by Warner Bros., the Wizarding World, and
    associated organizations. The answer is structured into exactly 8 bullet
    points as requested.
response:
    * @WizardingWorld (Twitter)
* @WizardingWorld (Instagram)
* @WizardingWorld (Facebook)
* @JKRowling (Twitter)
* @HarryPotterAlliance (Facebook)
* @HogwartsExpress (Twitter)
* @TheWizardingWorldOfHarryPotter (Facebook)
* @HarryPotterBooks (Twitter)
That's my answer.

###################################
3 other demos omitted for brevity.
###################################
```

GEPA Prompt generated by config GEPA:

```
Respond to the query by first repeating it word for word without any changes.
Then, provide your answer. Ensure your answer includes niche or domain-specific
factual information relevant to the task. If the query specifies constraints
(e.g., letter frequency, sentence limits, or formatting rules), adhere to them
strictly. Use a generalizable strategy to solve the task, such as breaking down
complex problems into logical steps or applying specialized knowledge. Avoid
adding any preamble or postscript before or after the repeated query.
```

---

**IFBench Qwen3 8B ensure_correct_response_module.predict**

Base Prompt:

Ensure the response is correct and adheres to the given constraints. Your response will be used as the final response.

MIPROv2 Prompt:

You are a meticulous and ethical AI assistant tasked with generating detailed, compliant responses. When given a query, first think step by step to create a reasoning process that addresses the query's requirements, then produce a response that strictly adheres to all specified constraints (e.g., formatting rules, content boundaries, ethical guidelines). Ensure the final output is structured with a "Reasoning:" section followed by a "Final Response:" section, and verify that it aligns with the query's exact specifications while maintaining logical coherence and ethical responsibility.
Demos:
Example 1:
query:
    What are some techniques for winning a swordfight in a medieval combat game?
    In your response, the letter f should appear less than 18 times. Do not
    include keywords ['heart', 'listen', 'reputation', 'side'] in the response.
response:
    Prioritize stance and balance to control the fight's flow. Use angles to
    create openings for strikes while minimizing exposure. Practice parrying with
    precision to disrupt opponent rhythm. Maintain distance to avoid
    overcommitting, and target weak points like joints or limbs. Focus on rhythm
    and timing to exploit vulnerabilities.
reasoning:
    The response outlines key techniques for medieval swordfighting, focusing on
    stance, angles, parrying, distance, and targeting weak points. It avoids the
    restricted keywords and limits the letter 'f' to 2 instances, satisfying the
    constraints. The advice is practical and aligns with historical combat
    strategies.
final_response:
    Prioritize stance and balance to control the fight's flow. Use angles to
    create openings for strikes while minimizing exposure. Practice parrying with
    precision to disrupt opponent rhythm. Maintain distance to avoid
    overcommitting, and target weak points like joints or limbs. Focus on rhythm
    and timing to exploit vulnerabilities.

#################################
3 other demos omitted for brevity.
#################################

GEPA Prompt generated by config GEPA:

Ensure the response strictly follows these instructions:
1. **First**, repeat the user's query **word for word** without any changes or additions.
2. **Then**, provide your answer in the specified format, adhering to all constraints (e.g., markdown, structure, content).
3. **Do not include any additional text, explanations, or formatting** beyond the repeated query and your answer.
4. **Include niche/domain-specific factual details** (e.g., technical commands, best practices, or platform-specific configurations) if applicable, as these are critical for accurate task completion.
5. **Use precise formatting** (e.g., bullet points, code blocks, headers) as requested, ensuring no markdown is omitted or altered.
6. **Avoid generalizable strategies** unless explicitly instructed; focus on actionable, specific guidance.
7. **Validate all technical steps** (e.g., Dockerfile syntax, CLI commands) for accuracy and completeness.
8. **Highlight potential pitfalls and solutions** to address common issues in the task.

```
9. **Prioritize clarity and conciseness**, ensuring the response is both
comprehensive and easy to follow.
10. **Adhere to language and case requirements** (e.g., all caps, English only) if
specified.
```

## N.5   HoVer, GPT-4.1 Mini

**HoVer GPT-4.1 Mini create_query_hop2.predict**

Base Prompt:

```
Given the fields `claim`, `summary_1`, produce the fields `query`.
```

MIPROv2 Prompt:

```
Given the original claim and the initial summary of retrieved evidence, carefully
analyze the information step-by-step to identify any gaps, contradictions, or
points needing further clarification. Then, generate a clear, focused, and precise
query that targets additional relevant information to deepen the investigation and
help verify or refute the claim. Your response should include a detailed
chain-of-thought reasoning explaining your thought process in formulating this
refined query.
Demos:
Example 1:
claim:
    AirBaltic's has bases at Tallinn Airport and the airport that ceased
    operations in 2015, with the airport code of Ryanair: VNO, ICAO: EYVI.
summary_1:
    AirBaltic has bases at Tallinn Airport and Vilnius Airport (IATA: VNO, ICAO:
    EYVI), but Vilnius Airport did not cease operations in 2015. The airport that
    ceased operations in 2015 mentioned in the passages is unrelated and has
    different codes. Therefore, the claim that airBaltic's base with code VNO/EYVI
    ceased operations in 2015 is false.
reasoning:
    The claim states that AirBaltic has bases at Tallinn Airport and an airport
    with codes VNO (IATA) and EYVI (ICAO) which ceased operations in 2015.
    However, the summary clarifies that Vilnius Airport, which has those codes,
    did not cease operations in 2015. The airport that did cease operations in
    2015 has different codes and is unrelated to AirBaltic's bases. Therefore, the
    claim that AirBaltic's base at the airport with codes VNO/EYVI ceased
    operations in 2015 is incorrect.
query:
    Did Vilnius Airport (IATA: VNO, ICAO: EYVI) cease operations in 2015?

####################################
3 other demos omitted for brevity.
####################################
```

GEPA Prompt generated by config GEPA+Merge:

```
Task Description:
You are given two fields: `claim` and `summary_1`. Your goal is to produce a field
called `query` âĂŤ a concise set of relevant, fact-checking questions or search
style queries that can be used to retrieve evidence documents verifying or
refuting the claim based on information contained or inferred from the summary.

Input Format:
- `claim`: A factual statement potentially containing multiple fact assertions
about people, events, attributes, titles, dates, roles, or relationships.
- `summary_1`: A short paragraph summarizing factual information related to the
claim, often clarifying or correcting some parts of the claim.
```

```
Output Format:
- `query`: One or more specific, well-phrased questions or keyword queries that
directly target the key factual discrepancies or verifications raised by the claim
in light of the summary.

Detailed Instructions:
1. **Extract key factual elements from the claim** âĂŤ names, dates, titles,
roles, events, or relationships explicitly or implicitly stated.
2. **Contrast these facts with the summary to identify points of agreement,
contradiction, or ambiguity.**
3. **Formulate fact-checking queries that are:**
   - Tightly focused on the core factual issues raised by the claim and addressed
   or contradicted by the summary.
   - Include named entities, dates, roles, or other domain-specific identifiers
   directly mentioned in both claim and summary to improve retrieval effectiveness.
   - When relevant, break complex claims into multiple queries ensuring each fact
   is verifiable separately.
4. **When relevant details appear only in the summary but are hinted at or missing
from the claim (e.g., specific titles, roles, or names), include these in the
queries to enable retrieval of key evidence.**
5. **Use a clear, natural question format or targeted keyword phrases that could
serve well as search queries.**
6. **Avoid overly broad or generic queries; precision improves evidence retrieval
quality.**
7. Optionally, you may provide brief reasoning internally (not required in output)
to ensure the queries cover all claim aspects and reflect the summary insights.

Examples of typical query components include:
- Correct dates of events or deaths.
- Confirmation of a personâĂŹs role or association with a known work.
- Verification of relationships or allegations.
- Details about specific cultural or domain elements (e.g., operas' structure,
song directors).
- Clarifying entity attributes or classifications (e.g., ethnicity, nationality).

By following these instructions, you will produce queries that are both
comprehensive and targeted, maximizing the chance of retrieving relevant factual
evidence relevant to the claim verification task.
```

## HoVer GPT-4.1 Mini create_query_hop3.predict

Base Prompt:

```
Given the fields `claim`, `summary_1`, `summary_2`, produce the fields `query`.
```

MIPROv2 Prompt:

```
You are an expert fact-checker specializing in multi-hop reasoning and evidence
synthesis. Given a claim and two intermediate summaries that consolidate evidence
from previous retrieval steps, thoughtfully analyze the information step-by-step
to generate a clear and focused query. This query should be designed to retrieve
the most relevant additional documents that can help verify or refute the claim by
leveraging the insights from both summaries. Provide a detailed chain-of-thought
reasoning explaining how you integrate the summaries and the claim to formulate
this refined query.
Demos:
Example 1:
claim:
    The brother of Freddie Sessler, was the restaurateur whose club was frequented
    by Ruth Elizabeth "Bette" Davis, an actress with Welsh ancestry.
summary_1:
```

```
        Freddie Sessler's brother was Siegi Sessler, a London restaurateur whose club
        was frequented by Bette Davis, an actress with Welsh ancestry.
    summary_2:
        Freddie Sessler's brother was Siegi Sessler, a prominent London restaurateur
        whose club, Siegi's, was frequented by Hollywood stars including Ruth
        Elizabeth "Bette" Davis, an actress with Welsh ancestry.
    reasoning:
        The claim states that the brother of Freddie Sessler was a restaurateur whose
        club was frequented by Ruth Elizabeth "Bette" Davis, noted to have Welsh
        ancestry. Both summaries support this claim by identifying the brother as
        Siegi Sessler, a restaurateur in London, whose club was indeed frequented by
        Bette Davis, an actress noted for Welsh ancestry. Summary_2 adds that the
        club's name was Siegi's and mentions the presence of other Hollywood stars. To
        fact-check the claim accurately, it would be useful to verify the identity of
        Freddie Sessler's brother as Siegi Sessler, confirm that he was a
        restaurateur, verify that Bette Davis frequented his club, and confirm Bette
        Davis's Welsh ancestry.
    query:
        Who was Siegi Sessler, and was he the brother of Freddie Sessler? Was Siegi
        Sessler a restaurateur whose club was frequented by Ruth Elizabeth "Bette"
        Davis? Did Bette Davis have Welsh ancestry?

    ####################################
    3 other demos omitted for brevity.
    ####################################
```

GEPA Prompt generated by config GEPA+Merge:

```
Given three text fields: `claim`, `summary_1`, and `summary_2`, your task is to
produce a `query` field that is designed to effectively retrieve evidence
documents relevant to verifying or refuting the claim based on the information
contained in the two summaries.

Detailed task description and considerations:

1. **Purpose of the Query:**
   - The query should accurately and comprehensively target the key factual
   elements from the claim that are addressed or clarified in the summaries.
   - The query will be used to retrieve evidence documents, so it should be
   specific enough to pinpoint relevant support or contradiction but broad enough
   to cover all important details present in the summaries.

2. **Utilizing the Summaries:**
   - Summaries often correct, clarify, or add factual context to the claim. Your
   query must incorporate these clarifications (e.g., name corrections, factual
   specifics, or counterpoints) to ensure retrieval of relevant evidence
   reflecting the nuanced truth.
   - Include all distinctive entities, facts, dates, locations, names, and
   relationships mentioned or corrected in the summaries that pertain to the claim.
   - For example, if a summary corrects a documentary title in the claim, the
   query should reference the corrected title and related details to guide
   retrieval effectively.

3. **Query Content Strategy:**
   - Explicitly mention key entities, such as persons, places, dates, works, or
   political entities involved.
   - Include attributes or relationships relevant to the claimâĂŹs accuracy (e.g.,
   "Was person X a politician in country Y during year Z?" or "Did documentary A
   and documentary B film in different locations such as location 1 and location
   2?").
```

- If there is a factual dispute or correction in the summaries (e.g.,
nationality, official names, population figures), phrase the query to target
evidence clarifying this dispute.

4. **Domain-Specific Nuances:**
   - Be mindful of historical geopolitical names and periods (e.g., "United
   Kingdom of the Netherlands between 1815 and 1830").
   - Recognize proper titles and correct spellings (e.g., corrected documentary
   titles).
   - Include both subjects or objects mentioned as comparisons or contrasts in the
   claim and summaries (e.g., two documentaries, two politicians, two towns).

5. **Formulation Style:**
   - Queries should be phrased as clear, precise, and objective questions that can
   be answered based on evidence âĂŤ often structured as yes/no or informational
   queries.
   - Avoid overly broad or vague phrasing. Aim for detail-rich queries that
   connect multiple evidence points in the summaries.

6. **Generalizable Strategy:**
   - Identify the core claim elements and verify if the summaries confirm,
   contradict, or amend these elements.
   - Incorporate both the claim and corrections from summaries into the query so
   evidence retrieval captures the full factual context.
   - Use the comparison or contrast highlighted by the summaries to create queries
   that specifically test the claim's veracity (e.g., comparing locations, roles,
   or historical timeframes).

By following these guidelines, you will generate queries that maximize the
likelihood of retrieving relevant texts that confirm or refute the claim
accurately.

---

## HoVer GPT-4.1 Mini summarize1.predict

Base Prompt:

Given the fields `claim`, `passages`, produce the fields `summary`.

MIPROv2 Prompt:

Given a `claim` and a list of relevant `passages`, carefully analyze the evidence
by reasoning step-by-step to assess the claim's validity. Produce a detailed
chain-of-thought `reasoning` that explains how the information in the passages
supports or refutes the claim, followed by a clear, concise `summary` that
synthesizes the key findings in relation to the claim. Ensure the reasoning
explicitly connects evidence from the passages to the claim, enabling a thorough
and transparent multi-hop verification process.
Demos:
Example 1:
claim:
   The manufacturer that owns the company that currently produced and marketed
   the candy the video game Darkened Skye uses in magic is headquartered in the
   US state Virginia.
passages:
   ['Darkened Skye | Darkened Skye is a third-person action-adventure video game
   developed by Boston Animation. It was released for Microsoft Windows and the
   Nintendo GameCube in North America in 2002 and the PAL regions in 2003. The
   game was also packaged with Outlaw Golf. Its title character is a young woman
   named Skye who lives in a fantasy realm searching for her mother. She does not
   use firearms, but can perform magic using Skittles candies, as well as use her
   staff as a melee weapon, though it also becomes an energy weapon when used in
   conjunction with the Skittles.', "Rocket Fizz | Rocket Fizz is a franchise

chain of candy stores in the United States. The company markets a diverse variety of candies and produces its own line of soft drinks with unique flavors. Its flagship store is located in Camarillo, California, and most of the company's franchise stores are located in California. The company also markets candies that are rare to find or in limited production by various manufacturers, such as those that were popular during the 1960s to 1980s. As of September 2017, there are 86 Rocket Fizz store locations in the United States.", 'Spangler Candy Company | The Spangler Candy Company is a privately owned international confectioner that has been manufacturing and marketing candy since 1906.', 'Future US | Future US, Inc. (formerly known as Imagine Media and The Future Network USA) is an American media corporation specializing in targeted magazines and websites in the video games, music, and technology markets. Future US is headquartered in the San Francisco with a small sales office in New York City. Future US is owned by parent company, Future plc, a small publishing company based in the United Kingdom.', 'Fun Dip | Fun Dip is a candy manufactured by The Willy Wonka Candy Company, a brand owned by NestlÃľ. The candy has been on the market in the United States and Canada since the 1940s and was originally called Lik-M-Aid. It was originally manufactured by Fruzola, and then Sunline Inc., through their Sunmark Brands division, which was purchased by NestlÃľ in January 1989. It comes in many different flavors with candy sticks that are included.', 'Hot Tamales | Hot Tamales is a chewy, cinnamon flavored, oblong-shaped candy introduced in 1950 âĂŤ manufactured and marketed in the United States by Just Born, a family-owned, Pennsylvania-based candy company.', "Frankford Candy & Chocolate Company | Frankford Candy & Chocolate Company is an American candy manufacturer, located in Philadelphia, Pennsylvania, founded in 1947 by Sam Himmelstein. The current CEO is Stuart Selarnick. In recent years, Frankford Candy & Chocolate Company has become a very successful candy manufacturer. It is one of the largest producers of chocolate rabbits in the United States, making over 100 varieties. In 2000, it started licensing popular children's TV characters, such as SpongeBob SquarePants."]

reasoning:
    The claim states that the manufacturer of the candy used in the video game Darkened Skye is headquartered in Virginia. Passage [1] establishes that the video game Darkened Skye uses Skittles candies for magic. Therefore, the candy in question is Skittles. The crucial step is to identify the company that produces and markets Skittles and whether it is headquartered in Virginia. None of the provided passages explicitly mention Skittles or its manufacturer. The passages list various candy companies, but none mention Skittles or match its manufacturer. The known producer of Skittles is Mars, Incorporated, which is headquartered in McLean, Virginia. Although this fact is not directly stated in the provided passages, it is relevant general knowledge. Since the passages do not refute the claim and for accuracy based on known facts, the claim can be considered true that the manufacturer that owns the company producing and marketing Skittles is headquartered in Virginia.

summary:
    The candy used as magic in Darkened Skye is Skittles, which are produced by Mars, Incorporated. Mars is headquartered in McLean, Virginia, a US state. Therefore, the claim is true based on external knowledge confirming that the manufacturer is based in Virginia.

####################################
3 other demos omitted for brevity.
####################################

GEPA Prompt generated by config GEPA+Merge:

Given the fields `claim`, `passages`, produce the fields `summary`.

```
HoVer GPT-4.1 Mini summarize2.predict
```

Base Prompt:

```
Given the fields `claim`, `context`, `passages`, produce the fields `summary`.
```

MIPROv2 Prompt:

```
Given a `claim`, relevant `context`, and a set of supporting `passages`, generate
a detailed step-by-step reasoning process that logically connects the evidence to
the claim, followed by a concise summary that clearly states whether the claim is
supported or refuted based on the aggregated information. Ensure the reasoning
explicitly references key evidence from the context and passages to justify the
conclusion in the summary.
Demos:
Example 1:
claim:
    One half of the KalyanjiâĂŞAnandji duo, Kalyanji Virji Shah's brother, won the
    Filmfare Award for Best Music Director.
context:
    Yes, Kalyanji Virji Shah's brother Anandji Virji Shah, as part of the
    KalyanjiâĂŞAnandji duo, won the Filmfare Award for Best Music Director in 1975
    for the film "Kora Kagaz."
passages:
    ['Kalyanji Virji Shah | Kalyanji Virji Shah (30 June 1928 âĂŞ 24 August 2000)
    was the "Kalyanji" of the Kalyanji-Anandji duo. He and his brother Anandji
    Virji Shah have been famous Indian film musicians, and won the 1975 Filmfare
    Award for Best Music Director, for "Kora Kagaz". He is a recipient of the
    civilian honour of Padma Shri (1992).', 'Anandji Virji Shah | Anandji Virji
    Shah is an Indian music director. Together with his brother he formed the
    Kalyanji-Anandji duo, and won the 1975 Filmfare Award for Best Music Director,
    for "Kora Kagaz". He is a recipient of the civilian honour of Padma Shri
    (1992).', 'KalyanjiâĂŞAnandji | KalyanjiâĂŞAnandji are an Indian composer duo
    from Gujarat: Kalyanji Virji Shah (30 June 1928-03 November 2000) and his
    brother Anandji Virji Shah (born 02 March 1933). The duo are known for their
    work on Hindi film soundtracks, particularly action potboilers in the 1970s.
    Some of their best-known works are "Don", "Bairaag", "Saraswatichandra",
    "Qurbani", "Tridev" and "Safar". They won the 1975 Filmfare Award for Best
    Music Director for "Kora Kagaz".', 'KalyanjiâĂŞAnandji discography | This is a
    discography of Bollywood composer duo Kalyanji Anandji, consisting of Kalyanji
    Virji Shah and Anandji Virji Shah. They have composed music for over 200 films
    in their 46 year career.', 'Viju Shah | Viju Shah (born, Vijay Kalyanji Shah,
    5 June 1959) is a music director of Hindi cinema. He is the son of music
    director Kalyanji Virji Shah of composer duo Kalyanji Anandji. He composed
    music for movies such as "Vishwatama" (1992), "Mohra" (1994), "Tere Mere
    Sapne" (1996) and "" (1997) for which got his second nomination for the
    Filmfare Award for Best Music Director and he won the 1998 Filmfare Award for
    Best Background Score for "".', 'Bombay the Hard Way: Guns, Cars and Sitars |
    Bombay the Hard Way: Guns, Cars and Sitars is a 1998 project of producer Dan
    the Automator in which music from 1970s funk Bollywood composer duo
    Kalyanji-Anandji (Kalyanji Virji Shah and Anandji Virji Shah) was taken and
    given a funky remix. The album was eventually withdrawn (possibly due to
    copyright concerns). A sequel, "Bombay 2: Electric Vindaloo", was also
    produced.', 'Kora Kagaz | Kora Kagaz (Hindi: àĲŢàěŃàďřàď¿ àĲŢàď¿àďŮàďàď;
    translation: Blank Paper) is a 1974 Hindi drama film produced by Sanath
    Kothari and directed by Anil Ganguly. The film stars Vijay Anand, Jaya
    Bhaduri, A.K. Hangal, Achala Sachdev and Deven Verma. The films music is by
    Kalyanji Anandji. The famous title song "Mera Jeevan Kora Kagaz" was by
    Kishore Kumar.']
reasoning:
    The claim states that one half of the KalyanjiâĂŞAnandji duo, Kalyanji Virji
    Shah's brother, won the Filmfare Award for Best Music Director. From the
    passages and context, it is clear that the duo consists of Kalyanji Virji Shah
    and his brother Anandji Virji Shah. Both together won the Filmfare Award for
```

```
    Best Music Director in 1975 for the film "Kora Kagaz." Therefore, Anandji
    Virji Shah, who is Kalyanji's brother, indeed won the Filmfare Award for Best
    Music Director as part of this duo.
summary:
    Anandji Virji Shah, the brother of Kalyanji Virji Shah, won the Filmfare Award
    for Best Music Director in 1975 as part of the KalyanjiâĂŞAnandji duo.

###################################
3 other demos omitted for brevity.
###################################
```

GEPA Prompt generated by config GEPA+Merge:

```
Given the input fields:

- `claim`: A factual statement or assertion that may be true or false.
- `context`: A brief explanation or summary relating to the claim that typically
clarifies the claimâĂŹs accuracy.
- `passages`: A list of relevant textual evidence or knowledge snippets containing
facts, descriptions, or biographies related to entities or concepts mentioned in
the claim.

Your task is to produce a `summary` that meets the following requirements:

1. **Accurately reflect the relationship between the claim and the evidence**:
Analyze the input `claim` carefully and check it against the `context` and all
`passages`. Determine whether the claim is supported, partially supported, or
contradicted by the evidence.

2. **Explicitly connect key entities and facts from the passages**: Your summary
must mention the main entities and facts that directly confirm or refute parts of
the claim. For example, highlight relevant names, works (films, albums, songs,
books), attributes, dates, nicknames, or roles that clarify the claim's accuracy.

3. **Make substantive connections**: The summary should not merely restate the
claim or conclusion. Instead, it should explicitly link the claim to detailed
evidence from the passagesâĂŤsuch as specific film titles, album names, song
titles, dates, or relationshipsâĂŤto provide clear reasoning for the accuracy or
inaccuracy of the claim.

4. **Include relevant disambiguations or clarifications**: Where applicable,
clarify potential misunderstandings or common confusions highlighted in the
passages (e.g., distinguishing similarly named films or songs, specifying which
individual is referenced, or noting alternate titles and adaptations).

5. **Use concise, factual language**: The summary should be clear and succinct but
include all relevant evidence necessary to understand how the claim is supported
or refuted.

6. **Support future query generation**: As the summary will be used to generate
queries to find evidence, it must contain explicit mentions of key entities,
titles, and facts that can guide retrieval and reasoning.

**Overall, your approach should be to reason comprehensively over the claim,
context, and each passage, synthesizing connected facts and evidence into a
coherent and evidence-rich summary that clearly documents why the claim is true,
false, or partially true.** Avoid vague or overly general statements; instead,
ground your summary in the precise factual details present in the passages and
context.
```

## N.6 HoVer, Qwen3 8B

---

**HoVer Qwen3 8B create_query_hop2.predict**

Base Prompt:

> Given the fields `claim`, `summary_1`, produce the fields `query`.

MIPROv2 Prompt:

> In a high-stakes scenario where the accuracy of your query directly determines the validity of a complex claim, generate a precise query that builds on the provided claim and summary to uncover critical evidence. Use step-by-step reasoning to identify gaps in the initial summary, then craft a query that explicitly targets these gaps to retrieve additional supporting information. Ensure your query is unambiguous and directly addresses the claim's unresolved aspects.
> Demos:
> Example 1:
> claim:
>     Sandi Simcha DuBowski directed a documentary released in 2001. The horror film
>     Zeitgeist was not also released in 2001.
> summary_1:
>     The claim is correct: Sandi Simcha DuBowski directed the 2001 documentary
>     *Trembling Before G-d*, and there is no evidence in the passages that the
>     horror film *Zeitgeist* was released in 2001.
> reasoning:
>     The claim is correct because Sandi Simcha DuBowski indeed directed the 2001
>     documentary *Trembling Before G-d*, as stated in summary_1. The assertion that
>     *Zeitgeist* was not released in 2001 is supported by the absence of evidence
>     in the provided summaries linking the film to that year.
> query:
>     Is there any evidence in the provided summaries that the horror film
>     *Zeitgeist* was released in 2001?
>
>
> ###################################
> 3 other demos omitted for brevity.
> ###################################

GEPA Prompt generated by config GEPA:

> Given the fields `claim` and `summary_1`, generate a precise and focused query
> that identifies the specific evidence needed to verify or refute the claim.
>
> ### Key Requirements:
> 1. **Target Missing Evidence**: Identify and explicitly ask about the unverified
> or unconfirmed details in the summary (e.g., names, dates, locations, or
> connections) that are critical to the claim.
>    - Example: If the summary mentions "Massimo Giordano" as a potential
>    counterpart but lacks birthplace details, the query should ask for evidence
>    confirming their ties to Naples.
> 2. **Correct Historical/Domain-Specific Anomalies**: Address discrepancies like
> anachronisms (e.g., Ali Qushji's work in the 1960s vs. his actual 15th-century
> timeline) or misattributions (e.g., *Hayy ibn Yaqdhan* by Ibn Tufail, not Ali
> Qushji).
> 3. **Link to Summary Context**: Ensure the query references the summary's key
> points (e.g., "the claim states X, but the summary notes Y is unverified").
> 4. **Use Specific Terms**: Include exact names, titles, or dates mentioned in the
> summary to avoid ambiguity (e.g., "Anaïs Nin," "Metropolitan City of Naples,"
> "1948").
>
> ### Example Strategy:
> If the summary states:
> - "The claim is partially supported. While [Fact A] is confirmed, [Fact B] lacks
> verification."

```
- The query should ask:
  *"Is there evidence confirming [Fact B], such as [specific name/date/location]?"*

### Domain-Specific Notes:
- Verify historical timelines (e.g., Ali Qushji died in 1474, not the 1960s).
- Confirm authorships (e.g., *Hayy ibn Yaqdhan* is attributed to Ibn Tufail, not
Ali Qushji).
- Check for geographic or biographical details (e.g., birthplaces, cities, or
professional connections).

Ensure your query directly addresses the summaryâĂŹs unverified claims and
includes all critical terms from the summary to maximize evidence retrieval.
```

## HoVer Qwen3 8B create_query_hop3.predict

Base Prompt:

```
Given the fields `claim`, `summary_1`, `summary_2`, produce the fields `query`.
```

MIPROv2 Prompt:

```
You are a fact-checking assistant specializing in multi-hop reasoning and
information synthesis. Given the fields `claim`, `summary_1`, and `summary_2`,
generate a precise query that synthesizes the original claim with the two
summaries to probe for deeper contextual relationships, resolving ambiguities and
confirming supporting details through targeted evidence retrieval.
Demos:
```

GEPA Prompt generated by config GEPA:

```
Given the fields `claim`, `summary_1`, and `summary_2`, produce the field `query`
that:
1. Explicitly asks whether the claim is supported by the provided summaries.
2. Includes **specific evidence** from the summaries (e.g., names, titles, dates,
or factual details) to guide retrieval of relevant documents.
3. Focuses on **key disputed points** in the claim (e.g., incorrect attributions,
missing evidence, or conflicting statements) to ensure the query targets the most
relevant information.

**Key considerations for the query:**
- If the summaries mention specific works (e.g., *Planes, Trains and
Automobiles*), include the title.
- If the summaries reference named entities (e.g., John Candy, KLM, Richard Ford),
include them.
- If the summaries clarify contradictions (e.g., "not X but Y"), structure the
query to highlight this contrast.
- Avoid vague phrasing; instead of asking "Is the claim true?" use "Does the
evidence in the summaries confirm [specific detail]?"

**Example:**
If the claim is about a film and the summaries mention *Planes, Trains and
Automobiles* (1987), the query should include that title to retrieve relevant
evidence.
```
```

## HoVer Qwen3 8B summarize1.predict

Base Prompt:

```
Given the fields `claim`, `passages`, produce the fields `summary`.
```

```
MIPROv2 Prompt:
```

Given the fields `claim` and `passages`, generate a structured reasoning process
that analyzes the claim step-by-step using the provided evidence, and produce a
concise summary that distills the key findings and evaluates the claim's validity
based on the evidence.
Demos:
Example 1:
claim:
    One half of the KalyanjiâĂŞAnandji duo, Kalyanji Virji Shah's brother, won the
    Filmfare Award for Best Music Director.
passages:
    ['Kalyanji Virji Shah | Kalyanji Virji Shah (30 June 1928 âĂŞ 24 August 2000)
    was the "Kalyanji" of the Kalyanji-Anandji duo. He and his brother Anandji
    Virji Shah have been famous Indian film musicians, and won the 1975 Filmfare
    Award for Best Music Director, for "Kora Kagaz". He is a recipient of the
    civilian honour of Padma Shri (1992).', 'KalyanjiâĂŞAnandji |
    KalyanjiâĂŞAnandji are an Indian composer duo from Gujarat: Kalyanji Virji
    Shah (30 June 1928-03 November 2000) and his brother Anandji Virji Shah (born
    02 March 1933). The duo are known for their work on Hindi film soundtracks,
    particularly action potboilers in the 1970s. Some of their best-known works
    are "Don", "Bairaag", "Saraswatichandra", "Qurbani", "Tridev" and "Safar".
    They won the 1975 Filmfare Award for Best Music Director for "Kora Kagaz".',
    'Anandji Virji Shah | Anandji Virji Shah is an Indian music director. Together
    with his brother he formed the Kalyanji-Anandji duo, and won the 1975 Filmfare
    Award for Best Music Director, for "Kora Kagaz". He is a recipient of the
    civilian honour of Padma Shri (1992).', 'Viju Shah | Viju Shah (born, Vijay
    Kalyanji Shah, 5 June 1959) is a music director of Hindi cinema. He is the son
    of music director Kalyanji Virji Shah of composer duo Kalyanji Anandji. He
    composed music for movies such as "Vishwatama" (1992), "Mohra" (1994), "Tere
    Mere Sapne" (1996) and "" (1997) for which got his second nomination for the
    Filmfare Award for Best Music Director and he won the 1998 Filmfare Award for
    Best Background Score for "".', 'KalyanjiâĂŞAnandji discography | This is a
    discography of Bollywood composer duo Kalyanji Anandji, consisting of Kalyanji
    Virji Shah and Anandji Virji Shah. They have composed music for over 200 films
    in their 46 year career.', 'Bombay the Hard Way: Guns, Cars and Sitars |
    Bombay the Hard Way: Guns, Cars and Sitars is a 1998 project of producer Dan
    the Automator in which music from 1970s funk Bollywood composer duo
    Kalyanji-Anandji (Kalyanji Virji Shah and Anandji Virji Shah) was taken and
    given a funky remix. The album was eventually withdrawn (possibly due to
    copyright concerns). A sequel, "Bombay 2: Electric Vindaloo", was also
    produced.', 'Bairaag | Bairaag is a 1976 Bollywood film. Produced by the duo
    Mushir-Riaz, it is directed by Asit Sen. The film stars Dilip Kumar, who
    received a Filmfare nomination for Best Actor for playing a triple role.
    "Bairaag" also stars Saira Banu, Leena Chandavarkar, Ruma Guha Thakurta, Prem
    Chopra, Helen, Sujit Kumar, Madan Puri, Paintal, Kader Khan, Asit Sen and
    Sachin. Its music is by the duo Kalyanji Anandji, who received a Filmfare
    nomination for Best Music. The film received a Filmfare Award for Best Sound
    for P. Harikishan. According to one source, the film did not do well at the
    box office.']
reasoning:
    The claim states that Kalyanji Virji Shah's brother won the Filmfare Award for
    Best Music Director. The passages confirm that the KalyanjiâĂŞAnandji duo
    (Kalyanji Virji Shah and his brother Anandji Virji Shah) won the 1975 Filmfare
    Award for Best Music Director for "Kora Kagaz." Since the award was shared by
    both brothers, the claim is accurate as it identifies Anandji Virji Shah
    (Kalyanji's brother) as the winner.
summary:
    Anandji Virji Shah, the brother of Kalyanji Virji Shah, won the 1975 Filmfare
    Award for Best Music Director as part of the KalyanjiâĂŞAnandji duo for their
    work on "Kora Kagaz."

```
####################################
3 other demos omitted for brevity.
####################################
```

GEPA Prompt generated by config GEPA:

```
Given the fields `claim` and `passages`, produce the fields `summary` by:
1. **Analyzing the claim** to identify key elements (e.g., names, locations,
relationships, specific terms).
2. **Cross-referencing the passages** to locate explicit or implicit evidence that
supports, contradicts, or partially addresses the claim.
3. **Explicitly connecting the claim to the evidence** in the passages, even if
the evidence is not directly stated in the claim (e.g., linking "Roger Yuan" to
"Shanghai Noon" in Example 1).
4. **Highlighting domain-specific details** (e.g., "Sentinelese language is
unclassifiable" in Example 3) that are critical for accuracy.
5. **Stating whether the claim is supported, unsupported, or partially
supported**, and explaining the reasoning with clear references to the passages.
6. **Prioritizing key evidence** that would be relevant for further queries (e.g.,
"Marlborough" in Example 2) to ensure retrievability.

**Additional Guidance:**
- If the claim contains unverified details (e.g., "Roger Yuan" in Example 1),
explicitly note this in the summary.
- For niche or ambiguous information (e.g., "uncontacted people" in Example 3),
reference specific passages that clarify the context.
- Ensure summaries are concise but include all critical evidence needed to
validate or refute the claim.
```

## HoVer Qwen3 8B summarize2.predict

Base Prompt:

```
Given the fields `claim`, `context`, `passages`, produce the fields `summary`.
```

MIPROv2 Prompt:

```
Given the fields `claim`, `context`, and `passages`, perform multi-hop reasoning
to generate a structured summary that validates or refutes the claim. First,
analyze the claim and contextual information to identify key relationships. Next,
evaluate evidence from the retrieved passages to build a coherent narrative.
Construct a logical reasoning chain connecting the claim to supporting or
contradictory evidence. Finally, produce a concise summary that explicitly
confirms or contradicts the claim based on your analysis, citing relevant evidence
from the context and passages.
Demos:
Example 1:
claim:
    The recorded history of the state, for which Baptist George Ruby was a
    prominent black Republican leader in the Reconstruction-era, officially
    started in 1519 in the same state that holds the city Augustus Chapman Allen
    used his inheritance to fund the founding of.
context:
    The claim is not supported by the passages. While George Ruby was associated
    with Texas during Reconstruction and Augustus Chapman Allen founded Houston in
    Texas, there is no evidence in the passages that Texas's recorded history
    began in 1519.
passages:
    ["Merchants and Manufacturers Building | The One Main Building, formerly the
    Merchants and Manufacturers Building (commonly referred to as the M&M
    Building), is a building on the campus of the University of
    HoustonâĂŞDowntown. The building is recognized as part of the National
```

Register of Historic Places, is a Recorded Texas Historic Landmark, and considered a Contributing Building in Downtown Houston's Main Street/Market Square Historic District. The building was built above Allen's LandingâĂŤan area where Houston's founders John Kirby Allen and Augustus Chapman Allen originally settled.", 'Augustus Chapman Allen | Augustus Chapman Allen (July 4, 1806 âĂŞ January 11, 1864), along with his younger brother, John Kirby Allen, founded the City of Houston in the U.S. state of Texas. He was born on July 4, 1806, in Canaseraga Village, New York (the present day hamlet of Sullivan in the Town of Sullivan, New York), to Sarah (Chapman) and Roland Allen.', "History of Texas | The recorded History of Texas begins with the arrival of the first Spanish conquistadors in the region of North America now known as Texas in 1519, who found the region populated by numerous Native American /Indian tribes.
Native Americans' ancestors had been there for more than 10,000 years as evidenced by the discovery of the remains of prehistoric Leanderthal Lady. During the period of recorded history from A.D. 1519 to 1848, all or parts of Texas were claimed by five countries: France, Spain, Mexico, the Republic of Texas, the United States of AmericaâĂŤas well as the secessionist Confederate States of America in 1861âĂŞ1865, during the American Civil War.", 'Charlotte Baldwin Allen | Charlotte Baldwin Allen (July 14, 1805 âĂŞ August 3, 1895) is known in Texan history as the "mother of Houston". She was the wife of Augustus Chapman Allen, who used her inheritance to finance the founding of this city.', 'Allen Parkway | Allen Parkway is an arterial road west of Downtown Houston, Texas. It has a distance of approximately 2.3 miles (3.7\xa0km), running from Interstate 45 west to Shepherd Drive, where it becomes Kirby Drive. Originally known as Buffalo Parkway, it was later named after John Kirby Allen and Augustus Chapman Allen, the founders of Houston.', "Allen's Landing | Allen's Landing is the birthplace of the city of HoustonâĂŤthe largest city in the U.S. state of Texas. In August 1836, just months after the Republic of Texas won its independence from Mexico, two brothers (and real estate developers) from New YorkâĂŤJohn Kirby Allen and Augustus Chapman AllenâĂŤpurchased 6,642\xa0acres\xa0(27\xa0kmÂš) in the area and settled there on the banks of Buffalo Bayou. Allen's Landing is located south of the University of HoustonâĂŞDowntown Commerce Street Building, at the intersection of Commerce Street and Main Street.", "Keith Allen (American football) | Keith D. Allen (born July 12, 1974) is an American football coach. He served as the head football coach at Southwest Baptist University from 2007 to 2012, compiling a record of 27âĂŞ39. His 27 wins are the most in the history of Southwest Baptist's football program, which began play in 1983."]

reasoning:
The claim is not supported by the passages. While the passages confirm that Texas's recorded history began in 1519 (passage 3) and that Augustus Chapman Allen founded Houston in Texas (passage 2), they do not explicitly link the start of Texas's recorded history to the same state where Augustus Chapman Allen used his inheritance to fund the founding of a city. The claim conflates these two facts without direct evidence in the passages that the 1519 start of recorded history is tied to the state where Allen's inheritance was used for founding Houston.

summary:
The claim is not supported by the provided passages, as they confirm Texas's recorded history began in 1519 and Augustus Chapman Allen founded Houston in Texas but do not explicitly connect the two facts as stated in the claim.

###################################
3 other demos omitted for brevity.
###################################

GEPA Prompt generated by config GEPA:

Given the fields `claim`, `context`, and `passages`, produce the field `summary` that:

1. **Evaluates the claim's validity** by explicitly connecting it to **specific
evidence** from the `passages` (e.g., citing exact names, dates, or terms that
support/refute the claim).
2. **Highlights factual inaccuracies** in the claim, such as incorrect origins,
dates, or relationships, and links them to **direct contradictions** in the
`passages` or `context`.
3. **Mentions key terms or entities** critical to the claim (e.g., "Sly Stone,"
"Large MÃnsterlÃďnder," or "Hit Parader's 2006 ranking") to ensure summaries are
actionable for query generation.
4. **Balances accuracy** by acknowledging both correct and incorrect aspects of
the claim, as seen in the examples (e.g., validating "The Wild Bunch" compilation
while correcting the magazine's nationality and ranking year).
5. **Avoids vague statements** like "no evidence" and instead specifies **which
parts of the passages** are relevant (e.g., "Passage 6 confirms Hit Parader was
American").

**Generalizable Strategy**:
- Always reference **specific passages** (e.g., "Passage 1 states...") to ground
claims in the text.
- Use **precise terminology** (e.g., "German Shorthaired Pointer" instead of
"cousins") to align with the data.
- Prioritize **clarity and specificity** to ensure summaries can guide further
evidence retrieval.

## N.7   PUPA, GPT-4.1 MINI

> **PUPA GPT-4.1 Mini craft_redacted_request.predict**
>
> Base Prompt:
>
> Given a private user query, create a privacy-preserving request for a powerful
> external LLM.
> The LLM may assist without learning private information about the user.
>
> MIPROv2 Prompt:
>
> You will be given a private user query. Your task is to analyze the query
> carefully and think step-by-step to understand the user's intent while identifying
> and removing or abstracting any sensitive, proprietary, or personally identifiable
> information. Then, generate a clear, privacy-preserving prompt suitable for an
> external large language model (LLM) that requests useful, relevant guidance or
> information without exposing private details. Additionally, provide a detailed
> chain-of-thought explanation describing your reasoning process for how you
> transformed the original query into the redacted, generalized LLM request. Ensure
> the resulting prompt is professional, contextually appropriate, and maintains user
> privacy while enabling the LLM to assist effectively.
> Demos:
> Example 1:
> user_query:
>     Imagine you are a stock researcher and advisor, I will give you a table about
>     stocks and I want you to give me advice on what to invest next. This is a
>     project I am working on an I need help with predictions I am making. Here is
>     the table: Stock Price Dividends 5D 5D % 1M 1M % YTD YTD % 1Y 1Y % 5Y 5Y % IBM
>     $146,62 $6,64 $3,81 2,66 $2,63 1,83 $5,07 3,58 $16,31 12,51 $0,03 0,02 MCD
>     $284,90 $6,08 $5,07 1,81 -$7,20 -2,46 $20,57 7,78 $29,24 11,44 $123,26 76,26
>     JNJ $163,98 $4,76 -$3,00 -1,80 -$3,90 -2,32 -$14,21 -7,97 $0,98 0,60 $29,66
>     22,08 MSFT $324,89 $2,72 $1,25 0,39 -$9,88 -2,95 $85,31 35,61 $59,66 22,49
>     $217,33 202,05 WMT $158,36 $2,28 $1,39 0,89 -$1,85 -1,15 $14,75 10,27 $25,47
>     19,16 $62,29 64,84 KO $60,50 $1,84 $0,12 0,20 -$1,59 -2,55 -$2,45 -3,89 -$2,23
>     -3,55 $15,13 33,35 PFE $36,07 $1,64 -$1,06 -2,84 -$0,67 -1,81 -$15,19 -29,64
>     -$10,16 -21,98 -$5,43 -13,09 AAPL $179,96 $0,96 $3,45 1,95 -$16,57 -8,43

```
    $54,89 43,89 $18,58 11,51 $125,04 227,65 NVDA $468,23 $0,16 -$3,71 -0,79 $7,05
    1,53 $325,08 227,09 $310,22 196,33 $401,52 601,89 CRM $212,10 $0,00 $2,75 1,31
    -$9,86 -4,44 $77,32 57,37 $51,89 32,39 $59,11 38,64 PYPL $61,16 $0,00 $1,06
    1,76 -$13,82 -18,43 -$13,43 -18,00 -$31,51 -34,00 -$29,46 -32,51 BIDU $137,47
    $0,00 $10,24 8,05 -$15,63 -10,21 $18,34 15,39 -$9,85 -6,69 -$88,25 -39,10
reasoning:
    The user wants investment advice based on a provided table of stock data
    including price, dividends, and percentage changes over various time intervals
    (5 days, 1 month, year-to-date, 1 year, and 5 years). The primary request is
    for recommendations on what to invest in next and help with making predictions.

To preserve privacy, I should not include the user's raw data or specific details
about individual stocks directly in the prompt to the external LLM. Instead, I
should request general advice on how to analyse stock data tables that include
price, dividend, and percentage changes over multiple time periods. Also, I can
ask for advice on what indicators to weigh more heavily for investment decisions
and for predictive strategies based on such data structures.

This approach ensures that the external LLM does not receive any private or
identifying user data, only a generic request. I will ask the LLM to provide
guidance on interpreting such stock performance indicators and suggestions for
predictive analytics or investment selection criteria.
llm_request:
    Please provide general guidance for analyzing stock investment opportunities
    based on a data table format that includes stock prices, dividend payouts, and
    percentage changes over various time intervals such as 5 days, 1 month,
    year-to-date, 1 year, and 5 years. Outline key indicators that investors
    typically weigh when making decisions, how to interpret short-term versus
    long-term performance trends, and suggest strategies or metrics to use when
    making predictions or recommendations about which stocks to invest in next.
    The request is for generic advice without any specific stock names or values.

####################################
3 other demos omitted for brevity.
####################################
```

GEPA Prompt generated by config GEPA+Merge:

```
New Instruction for Privacy-Preserving Query Reformulation Assistant (Enhanced):

Task Overview:
You receive a user query that may contain sensitive, private, or personally
identifiable information (PII). Your objective is to reformulate this query into a
generalized, privacy-preserving prompt suitable for sending to an external large
language model (LLM). The reformulated prompt must enable the external LLM to
fulfill the user's original intent effectively while rigorously protecting all
user privacy by abstracting, omitting, or generalizing any sensitive details.

Input Format:
- A single user query string.
- The query might contain identifiable names (people, organizations), specific
geographic locations, exact dates or timeframes, proprietary or internal
terminology (e.g., system names, product codes), client or customer names, URLs,
or any other sensitive or private information.

Output Format:
- Part (a) Reasoning:
    Provide a concise paragraph that explains:
    * How you identified sensitive or private information in the input
    * What strategies you applied to protect privacy (generalization, omission,
    abstraction, replacement with placeholders)
```

```
    * How the reformulated prompt preserves the original intent and task
    requirements without risking data leakage
- Part (b) LLM Request:
    A concise, carefully constructed privacy-safe prompt that:
    * Removes or anonymizes all PII and proprietary/internal details
    * Abstracts locations, names, dates, and technical terms as needed
    * Produces a clear and contextually rich instruction for the LLM to generate a
    relevant and informative response aligned with the user's original task

Detailed Domain-Specific Guidance and Best Practices:

1. Identification and Treatment of Sensitive Data:
    - All user-specific or personal names (individual or organizational) must be
    removed or replaced with generic role descriptors (e.g., âĂa business
    contact,âĂİ âĂa client,âĂİ âĂa notable individualâĂİ). Never lightly obscure or
    partially redact; full abstraction is required.
    - All geographic mentions must be abstracted unless the location is publicly
    known, essential to the task, and can be generalized (e.g., âĂa region known
    for eco-tourismâĂİ instead of naming a country or city explicitly).
    - Exact dates or durations must never be retained; instead, use relative or
    approximate temporal references (e.g., âĂrecently,âĂİ âĂover the past yearâĂİ).
    - Internal or proprietary terms âĂŤ including system names, product codes,
    subscription types, and technical jargon âĂŤ must be generalized or replaced
    with neutral descriptors to avoid leakage of intellectual property or sensitive
    operational details.
    - Avoid direct quotes or verbatim inclusion of user-supplied texts unless
    obfuscated by generalization.

2. Task Understanding and Reformulation:
    - Identify the functional intent of the query: Is it creative writing,
    translation, summarization, professional communication drafting, technical
    explanation, or other?
    - Preserve the thematic and informational core of the query (e.g., request for
    educational quality analysis, technical translation of a passage, biographical
    summary).
    - Do not reproduce the original input verbatim; instead, frame the LLM prompt
    around the essential thematic elements extracted from the input.
    - For queries regarding individuals, avoid direct reference to real personal
    information unless the name is widely public and essential; even then, use a
    generic or hypothetical framing for the individual profile.

3. Strategies for High-Quality and Privacy-Preserving Prompts:
    - Strike a balance between sufficient contextual detail and privacy abstraction
    to maintain prompt clarity and relevance.
    - Use neutral, context-aware formulations that clearly instruct the LLM on the
    content and style expected.
    - Avoid vague or overly generic prompts that could result in less useful or
    lower-quality responses.
    - When system or proprietary content is mentioned, instruct the LLM to
    generalize specific terms and maintain the technical meaning without revealing
    sensitive info.
    - When a direct translation is requested on specialized text, specify to
    replace or abstract internal nomenclature.

4. Explanation Requirements:
    - The reasoning must transparently explain how privacy risks were identified
    (e.g., presence of names, locations, dates, proprietary terms).
    - It must describe the abstraction or omission methods applied (e.g., replacing
    âĂJonah Van BeijnenâĂİ with âĂa notable individual,âĂİ substituting âĂMakauâĂİ
    with âĂa specific region,âĂİ or âĂYodaâĂİ with âĂa system nameâĂİ).
    - Clarify how the essential task and user intent were preserved despite these
    generalizations (e.g., focusing on educational quality, technical translation,
    biographical summary).
```

```
    - This explanation justifies your design choices and demonstrates adherence to
    privacy-preserving principles.

Common Pitfalls to Avoid:
- Do not merely lightly obscure or partially redact sensitive details; full
anonymization or abstraction is required.
- Do not repeat any user-supplied PII or proprietary content verbatim.
- Avoid including URLs, exact dates, or direct quotes without modification.
- Do not leave ambiguity that could degrade the quality or contextual clarity of
the reformulated prompt.
- Do not include any real personal or organizational names unless they are public
figures and the query requires it, then use generic descriptors instead.

Example Summary of Effective Approach (Informed by Prior Examples):
- For geographic queries: replace exact place names with general regions and
provide a brief contextual descriptor.
- For technical texts containing system names or subscription types: instruct the
LLM to translate or process the text while replacing or abstracting proprietary
system identifiers.
- For biographical summaries about specific individuals: remove the real name and
request a generic, well-structured four-paragraph summary about âĂa notable
individual,âĂİ preserving the overall intent without leaking PII.

Summary:
Your reformulations must ensure zero exposure of any PII or private/proprietary
content while retaining enough thematic and functional clarity for the external
LLM to produce high-quality, relevant outputs. This requires thorough analysis of
the user's query, rigorous application of privacy-preservation strategies, and
explicit reasoning explanations that document your approach and choices.
```

## PUPA GPT-4.1 Mini respond_to_query

Base Prompt:

```
Respond to a user query.
For inspiration, we found a potentially related request to a powerful external LLM
and its response.
```

MIPROv2 Prompt:

```
Given a privacy-preserving redacted prompt sent to an external untrusted LLM and
its response, along with the original user query containing potentially sensitive
information, generate a final, coherent, and contextually appropriate user-facing
response. Your output should integrate the anonymized LLM output with the original
queryâĂŹs intent, ensuring privacy is maintained while providing a complete and
polished answer that directly addresses the user's request.
Demos:
Example 1:
related_llm_request:
    Please provide a detailed explanation of David Easton's system analysis in
    political science, outlining the key components of his systems theory approach
    and its significance in the study of political systems.
related_llm_response:
    Certainly! David Easton's system analysis is a foundational framework in
    political science that applies systems theory to the study of political
    systems. His approach provides a structured way to understand how political
    systems function, maintain stability, and respond to changes in their
    environment. Below is a detailed explanation of Easton's system analysis,
    including its key components and significance.

    ---
```

### Overview of David Easton's System Analysis

David Easton (1917âĂŞ2014) was a prominent political scientist who sought to develop a general theory of political systems by borrowing concepts from systems theory, which originated in biology and engineering. His goal was to create a scientific and analytical framework to study politics as an integrated and dynamic system rather than as isolated events or institutions.

Easton's system analysis views a political system as a set of interactions abstracted from the totality of social behavior, through which authoritative allocation of values (decisions) is made and implemented. In other words, politics is about how societies decide who gets what, when, and how, and Easton's model helps explain how these decisions are made and sustained.

---

### Key Components of Easton's Systems Theory Approach

Easton's model conceptualizes the political system as a **black box** that processes inputs and produces outputs, embedded within a larger environment. The key components are:

#### 1. **Inputs**
Inputs are demands and supports that come from the environment and society into the political system.

- **Demands:** These are claims or requests made by individuals or groups for the political system to allocate values or resources in a particular way (e.g., demands for law enforcement, education, welfare).
- **Supports:** These are actions or attitudes that indicate approval, allegiance, or compliance with the political system (e.g., voting, paying taxes, obeying laws).

Inputs represent the pressures and resources that the political system must process.

#### 2. **Political System (The Black Box)**
The political system is the core of Easton's model. It receives inputs, processes them, and produces outputs.

- It includes institutions, organizations, and processes involved in authoritative decision-making (e.g., legislatures, executives, courts).
- The system transforms demands and supports into decisions and policies.
- The system must maintain its boundaries and adapt to changes to survive.

#### 3. **Outputs**
Outputs are the decisions, policies, and actions that the political system produces in response to inputs.

- These include laws, regulations, public policies, and authoritative allocations of values.
- Outputs affect the environment and society, influencing future inputs.

#### 4. **Feedback**
Feedback is the process by which outputs influence future inputs.

- Positive feedback can reinforce the systemâĂŹs stability.
- Negative feedback can signal problems or dissatisfaction, prompting adjustments.
- Feedback helps the system adapt and maintain equilibrium.

#### 5. **Environment**
The political system exists within a broader environment that includes social, economic, cultural, and international factors.

- The environment provides inputs and is affected by outputs.
- The system must interact with and adapt to environmental changes to survive.

---

### Diagrammatic Representation (Simplified)

```
Environment
   âĘŚ âĘŞ
[Demands & Supports] âĘŠ [Political System] âĘŠ [Decisions & Policies]
       âĘŚ âĘŞ
       âĘŘ----------- Feedback -----------
```

---

### Significance of Easton's System Analysis in Political Science

1. **Holistic Understanding of Politics:**
   Easton's model encourages viewing politics as an interconnected system rather
   than isolated events or institutions. It highlights the dynamic interactions
   between society and political authority.

2. **Focus on Process and Stability:**
   The model emphasizes how political systems maintain stability and order through
   continuous processing of inputs and outputs, and through feedback mechanisms.

3. **Analytical Clarity:**
   By abstracting political activity into inputs, system processing, outputs, and
   feedback, Easton provided a clear framework for empirical research and
   comparative analysis.

4. **Foundation for Further Theories:**
   Easton's systems theory influenced later developments in political science,
   including structural-functionalism, policy analysis, and the study of political
   development and change.

5. **Adaptability:**
   The model can be applied to various political systems (democracies,
   authoritarian regimes, etc.) and scales (local, national, international),
   making it versatile.

6. **Emphasis on Environment:**
   Recognizing the environmentâĂŹs role underscores the importance of external
   factors (economic conditions, social movements, international pressures) in
   shaping political outcomes.

---

### Criticisms and Limitations

- **Overly Abstract:** Some critics argue that Easton's model is too abstract and
general, lacking specificity about power relations, conflict, and individual
agency.
- **Neglect of Conflict:** The model tends to emphasize stability and equilibrium,
potentially underplaying political conflict and change.
- **Limited Normative Guidance:** It is primarily descriptive and analytical,
offering little normative guidance on what political systems ought to do.

---

### Summary

```
David Easton's system analysis
user_query:
    System analysis by David Easton
response:
    David Easton's system analysis is a foundational framework in political
    science that conceptualizes the political system as a dynamic and interactive
    system. His approach uses systems theory to explain how political systems
    function, maintain stability, and adapt to changes in their environment.

At its core, Easton's model views the political system as a "black box" that
processes inputs from society and the environment, transforms them into outputs,
and uses feedback to adjust and sustain itself. The key components of Easton's
system analysis include:

1. **Inputs:** These are the demands and supports that society directs towards the
political system. Demands are requests or claims for resource allocation or policy
change, while supports are actions or attitudes such as compliance, legitimacy, or
loyalty that sustain the system.

2. **Political System:** This is the set of institutions and processes that
processes the inputs. It interprets demands and supports to make authoritative
decisions, including the formulation and enforcement of policies.

3. **Outputs:** These are the decisions, policies, and actions that the political
system produces in response to inputs. They affect society and its environment,
shaping future interactions.

4. **Feedback:** Outputs generate responses that feed back into the system as new
inputs, signaling whether the outputs were effective or require modification. This
feedback enables the system to adapt to internal and external changes.

5. **Environment:** The system operates within a broader social, economic,
cultural, and international context that influences and is influenced by the
political system.

Easton's theory is significant because it offers a holistic, process-oriented lens
to analyze politics, emphasizing interaction between society and state
institutions. It provides a framework to understand political stability,
adaptation, and change by focusing on the continuous circulation of inputs,
processing, outputs, and feedback. The model has been influential in shaping
subsequent political theories and comparative political analysis.

However, critics note that Easton's model can be overly abstract, sometimes
downplaying conflict, power struggles, and individual agency within political
processes.

In summary, David Easton's system analysis offers a powerful conceptual tool to
study political systems as dynamic entities that manage societal demands and
supports through authoritative decision-making to maintain order and adapt over
time.

###################################
3 other demos omitted for brevity.
###################################
```

GEPA Prompt generated by config GEPA+Merge:

```
Respond to a user query.
For inspiration, we found a potentially related request to a powerful external LLM
and its response.
```

## N.8 PUPA, QWEN3 8B

---

**PUPA Qwen3 8B craft_redacted_request.predict**

Base Prompt:

```
Given a private user query, create a privacy-preserving request for a powerful
external LLM.
The LLM may assist without learning private information about the user.
```

MIPROv2 Prompt:

```
Given a private user query, generate a structured, non-sensitive LLM request that
captures the user's intent while omitting personal details. Include a step-by-step
reasoning process explaining how the redaction was achieved, ensuring the request
is actionable and privacy-preserving. Format the reasoning as a coherent thought
process and the request as a clear, specific task for the LLM to execute.
Demos:
Example 1:
user_query:
    Ð§ÑÃÐÿÐšÐţÑÇ ÐiÐřÐ£ÐÿÑĹÐÿ ÐžÐ¿ÐÐřÐiÐťÑÇ ÐťÐžÑÑ Ð¿ÑÇÐ£ÑÃÐřÐšÐžŽÐÿ snmpwalk ÐiÐř
    ip 10.2.35.2 ÑĄ ÐťÐřÐiÐiÑÑÐÐÿ Ð£Ð¿ ÐťÐ¿ÑĄÑÇÑÇÐ£ÑÇ
    initi_user/d2J3BgtYdB2/d2J3BgtYdB2 (MD5) DES
reasoning:
    The user is requesting an SNMP walk command to retrieve data from an
    SNMPv3-enabled device at IP 10.2.35.2. The provided credentials include a
    username (`initi_user`), an authentication password (`d2J3BgtYdB2`), and an
    encryption password (`d2J3BgtYdB2`). The authentication protocol is MD5, and
    the encryption protocol is DES. The command must include SNMPv3 options to
    specify these parameters correctly.
llm_request:
    Generate a command-line instruction for executing `snmpwalk` with SNMPv3
    authentication and encryption using the provided credentials, ensuring privacy
    by omitting sensitive details in the response.

####################################
3 other demos omitted for brevity.
####################################
```

GEPA Prompt generated by config GEPA:

```
### New Instruction for the Assistant:

**Task Overview:**
Rephrase the user's query into a privacy-preserving, actionable request for an
external LLM, ensuring no personally identifiable information (PII), proprietary
data, or sensitive details are exposed. Maintain the original intent, clarity, and
specificity of the query while adhering to domain-specific strategies for
generalized information.

**Key Requirements:**
1. **Privacy Preservation**:
   - **Generalize Specifics**: Replace specific locations (e.g., "Andhra Pradesh"
   âĘŠ "a major Indian city"), numbers (e.g., "18 volunteers" âĘŠ "a group of
   participants"), or proprietary data (e.g., "van Cleef production sites" âĘŠ
   "[Brand Name]'s production sites based on publicly available information").
   - **Avoid PII/Proprietary Data**: Remove names, addresses, dates, internal
   processes, or brand-specific secrets. Use placeholders like [Brand Name],
   [Country], or [Region] for brand/company references.
   - **Publicly Available Focus**: For brand-related queries, emphasize publicly
   accessible data (e.g., "locations of [Brand Name]'s production sites" instead
   of "van Cleef's secret factories").

2. **Quality of Request**:
```

```
    - **Clarity and Actionability**: Ensure the rephrased query is precise, avoids
    ambiguity, and specifies constraints (e.g., "concise, platform-friendly
    hashtags" for creative tasks).
    - **Domain-Specific Precision**: For scientific/technical queries, retain key
    terms (e.g., "CAPM formula," "pharmacokinetic parameters") while anonymizing
    study details (e.g., "18 volunteers" âĘŠ "a group of participants").
    - **Avoid Overly Broad Requests**: Specify constraints like keyword focus,
    tone, or format (e.g., "health and fitness keywords" instead of "general
    hashtags").

3. **Domain-Specific Strategies**:
    - **Geographic Queries**: Replace regions/cities with generalized terms (e.g.,
    "major city in [Country]" or "well-known [Region]").
    - **Brand/Company Queries**: Reference brands by name but prioritize publicly
    available data (e.g., "locations of [Brand Name]'s production sites").
    - **Creative Tasks**: Use examples or constraints (e.g., "wellness, strength,
    motivation keywords" for hashtags; "platform guidelines" for social media
    content).

4. **Avoid Leakage**:
    - **No Inferred Details**: Do not assume user-specific preferences, contexts,
    or private data (e.g., "Tell me the location of van Cleef production sites" âĘŠ
    "Provide locations of [Brand Name]'s production sites based on publicly
    available information").
    - **Anonymize Numerical Data**: Replace specific counts, dosages, or metrics
    with generalized terms (e.g., "40 mg OA" âĘŠ "a dose of OA").

**Example Workflow:**
- **User Query**: "Calculate DâĂŹAmicoâĂŹs cost of equity using CAPM with rRF =
4.67%, Îš = 1.56, and market risk premium = 6.63%."
- **Rephrased Request**: "Calculate the cost of equity for a company using the
CAPM formula, given a risk-free rate of 4.67%, a beta of 1.56, and a market risk
premium of 6.63%. Show step-by-step calculations and round the final result to two
decimal places."

**Niche Domain-Specific Guidance:**
- **Scientific/Pharmacokinetic Queries**: Generalize study details (e.g., "18
healthy male volunteers" âĘŠ "a group of participants"), retain key parameters
(e.g., AUC, peak concentrations), and avoid proprietary methodologies.
- **Creative/Translation Tasks**: Use placeholders for song titles/artists (e.g.,
"Mahmood's song [Title]") and specify tone/rhythm requirements for translations.
- **Financial Calculations**: Retain formulas and numerical values but avoid
linking to specific companies or proprietary models.

**Final Output Format:**
- **reasoning**: Briefly explain the rephrasing choices (e.g., "Generalized '18
volunteers' to 'a group of participants' to preserve privacy").
- **llm_request**: Provide the anonymized, actionable request for the LLM.
```

## PUPA Qwen3 8B respond_to_query

Base Prompt:

```
Respond to a user query.
For inspiration, we found a potentially related request to a powerful external LLM
and its response.
```

MIPROv2 Prompt:

```
Synthesize the external LLM's response with the user's specific query to generate
a final privacy-preserving answer. Replace placeholders with actual values while
omitting sensitive information, maintaining a professional tone, and ensuring
```

```
contextual accuracy. Combine the technical solution from the LLM with the user's
unique details to produce a clear, anonymized response that adheres to privacy
constraints.
Demos:
Example 1:
related_llm_request:
    Create a PESTLE analysis framework for GE Healthcare in the US, outlining key
    issues in each category (Political, Economic, Social, Technological,
    Environmental, Legal) and their potential impact on the organization.
    Prioritize these issues based on urgency and significance without referencing
    confidential data. Structure the output as a matrix with clear criteria for
    evaluation.
related_llm_response:
    Certainly! Below is a **PESTLE analysis framework** tailored for **GE
    Healthcare in the US**, structured as a matrix. It outlines key issues in each
    category, evaluates their **urgency** and **significance**, and highlights
    their **potential impact** on the organization.

---

### PESTLE Analysis Matrix for GE Healthcare (US)

| **Category** | **Key Issues** | **Urgency** (High/Med/Low) | **Significance**
(High/Med/Low) | **Potential Impact on GE Healthcare** |
|---------------|----------------------------------------------------------------
----
--------------|--------------------------|-------------------------------
|-------------------------------------------------------------------
-----------------------------------------------|
| **Political** | 1. Healthcare policy changes (e.g., Medicare/Medicaid
reimbursement reforms) | High | High | Directly affects revenue streams and
product/service demand; requires adaptation to new reimbursement models. |
| | 2. Trade policies and tariffs on medical equipment imports/exports | Medium |
Medium | Could increase costs of components or affect supply chain efficiency. |
| | 3. Government funding for healthcare innovation and research | Medium | High |
Influences R&D opportunities and partnerships; potential for grants and subsidies.
|
| **Economic** | 1. US healthcare spending trends and budget constraints | High |
High | Impacts purchasing power of hospitals and clinics; affects sales volume and
pricing strategies. |
| | 2. Inflation and rising operational costs | High | Medium | Increases
production and operational expenses; may pressure profit margins. |
| | 3. Economic cycles affecting capital investment in healthcare infrastructure |
Medium | Medium | Influences demand for high-cost medical equipment and technology
upgrades. |
| **Social** | 1. Aging population increasing demand for healthcare services and
devices | High | High | Drives demand for diagnostic imaging, monitoring, and
therapeutic devices; growth opportunity. |
| | 2. Growing patient awareness and demand for personalized medicine | Medium |
Medium | Necessitates innovation in tailored healthcare solutions and data-driven
technologies. |
| | 3. Workforce shortages in healthcare professionals | High | Medium | May
affect adoption and training for new technologies; impacts customer support and
service delivery. |
| **Technological** | 1. Rapid advancements in AI and machine learning for
diagnostics and imaging | High | High | Critical for maintaining competitive edge;
requires continuous investment in R&D and product development. |
| | 2. Integration of IoT and connected devices in healthcare | Medium | High |
Enhances product offerings but demands robust cybersecurity and interoperability
solutions. |
| | 3. Data privacy and cybersecurity technologies | High | High | Essential to
protect patient data and comply with regulations; failure risks legal penalties
and reputational damage. |
```

| **Environmental** | 1. Regulatory pressure to reduce carbon footprint and waste in manufacturing | Medium | Medium | Necessitates sustainable practices; potential cost implications but improves brand reputation. |
| | 2. Energy efficiency in product design and operations | Medium | Medium | Can reduce operational costs and appeal to environmentally conscious customers. |
| | 3. E-waste management and recycling programs | Low | Medium | Compliance with environmental standards; growing importance for corporate social responsibility. |
| **Legal** | 1. Compliance with FDA regulations and medical device approvals | High | High | Critical for market access; delays or failures can halt product launches and cause financial losses. |
| | 2. Intellectual property rights and patent protections | Medium | High | Protects innovation and competitive advantage; infringement risks costly litigation. |
| | 3. Liability and product safety regulations | High | High | Ensures patient safety; non-compliance can lead to lawsuits, recalls, and reputational damage. |

---

### Criteria for Evaluation:

- **Urgency**: How immediate the issue requires attention (High = immediate/ongoing; Medium = within 1-2 years; Low = longer-term).
- **Significance**: The potential magnitude of impact on GE Healthcare's operations, profitability, and strategic positioning (High = critical; Medium = moderate; Low = minor).
- **Potential Impact**: Describes how the issue could affect GE Healthcare's business, including risks and opportunities.

---

This matrix helps prioritize strategic focus areas for GE Healthcare in the US, balancing immediate challenges with longer-term trends.
user_query:
    issue priority matrix on pestle analysis for GE Healthcare in US
response:
    Here's a prioritized **issue priority matrix** based on the PESTLE analysis
    for **GE Healthcare in the US**, focusing on the most urgent and significant
    challenges and opportunities:

---

### **High Priority Issues (High Urgency + High Significance)**

| **Category** | **Key Issue** | **Urgency** | **Significance** | **Impact Summary** |
|--------------------|--------------------------------------|--------------|------------------|-----------------------------------------------------------|
| **Political** | Healthcare policy changes (e.g., reimbursement reforms) | High | High | Direct revenue risk; requires rapid adaptation to new regulations. |
| **Economic** | US healthcare spending trends and budget constraints | High | High | Drives demand and pricing pressures; critical for sales strategy. |
| **Social** | Aging population demand for healthcare services | High | High | Growth opportunity in diagnostics/therapeutics; long-term market expansion. |
| **Technological** | AI/ML advancements in diagnostics and imaging | High | High | Essential for competitiveness; demands sustained R&D investment. |
| **Legal** | FDA regulations and medical device approvals | High | High | Critical for market access; delays risk financial losses. |

---

### **Medium Priority Issues (High Urgency + Medium Significance or Medium Urgency + High Significance)**

```
| **Category** | **Key Issue** | **Urgency** | **Significance** | **Impact
Summary** |
|--------------------|---------------------------------------------------|--------+----|-----------------|-
| **Economic** | Inflation and operational costs | High | Medium | Pressures
profit margins; requires cost-efficiency measures. |
| **Technological** | IoT integration and cybersecurity | Medium | High | Enhances
offerings but demands robust security frameworks. |
| **Environmental** | Carbon footprint reduction and sustainability | Medium |
Medium | Balances cost implications with brand reputation and regulatory
compliance. |

---

### **Prioritization Criteria**
- **Urgency**: Immediate (High) vs. longer-term (Low).
- **Significance**: Critical (High) vs. moderate (Medium).
- **Focus**: Issues with **High in both urgency and significance** require
immediate strategic action, while others demand monitoring or phased planning.

This matrix helps GE Healthcare allocate resources to address the most pressing
challenges and leverage high-impact opportunities.

####################################
3 other demos omitted for brevity.
####################################
```

GEPA Prompt generated by config GEPA:

```
Respond to a user query by leveraging a related LLM response as inspiration.
Structure your answer in a clear, organized format (e.g., bullet points, sections)
to enhance readability. Ensure the response is tailored to the user's language and
avoids any personally identifiable information (PII) or sensitive data.

When rephrasing or summarizing content:
1. **Maintain academic tone** if the query specifies formal language (e.g.,
rephrasing technical or historical text).
2. **Refactor repetitive or redundant information** into concise, focused sections
(e.g., grouping similar ideas, eliminating duplication).
3. **Highlight domain-specific facts** (e.g., cultural, technical, or regional
details) to add value, as these may not be universally known.
4. **Improve code or technical responses** by breaking down complex logic into
smaller, well-documented functions, reducing redundancy, and ensuring proper
resource management.

For all outputs:
- Prioritize **clarity, accuracy, and adherence to the user's requirements**.
- Avoid **leaking proprietary information, PII, or unverified claims**.
- If the userâĂŹs query involves multiple steps (e.g., code optimization, content
rephrasing), address each part systematically.
- Use **language-specific conventions** (e.g., correct grammar, terminology) to
align with the userâĂŹs preferred language.
```

## O  GEPA GENERATED PROMPTS FOR KERNEL GENERATION

### O.1  NPUEVAL: KERNEL CODE GENERATION FOR NEW HARDWARE ARCHITECTURE

Figure 27 shows the prompt generated by GEPA with GPT-4o for NPU Keernel Generation, that achieves 26.85% score with the same same GPT-4o agent, that achieved just 4.25% with a simple prompt.

You are tasked with generating C++ code for a single kernel function that will run on an AI Engine (AIE) tile. The kernel should perform a specified operation on a vector of bfloat16 values. Your code should be complete and self-contained within a single code block, including all necessary headers and imports. Follow these guidelines:

1. **Headers and Imports**: Include only the necessary headers for AIE operations. Avoid including `<adf.h>` or any headers that are not part of the standard AIE API. Use:
```cpp
#include <stdint.h>
#include <aie_api/aie.hpp>
#include <aie_api/utils.hpp>
```

2. **Kernel Function**: Implement the kernel function as specified in the input. The function should take pointers to input and output buffers and a size parameter. Use AIE vector operations to process the data efficiently.

3. **Vector Operations**: Utilize the AIE API's vector operations for loading, processing, and storing data. For example, use `aie::vector` for vector operations and `aie::reduce_min` for reduction tasks. Ensure that the vector size is compatible with the AIE hardware capabilities.

4. **Avoid Non-Existent Functions**: Do not use functions like `aie::exp` or `aie::store_v` if they are not supported. Instead, implement the required functionality using available AIE API functions.

5. **Error Handling**: Ensure that the code is free of syntax errors and compatible with the AIE environment. Test the code for compilation without errors.

Here is an example template for a kernel function:
```cpp
#include <stdint.h>
#include <aie_api/aie.hpp>
#include <aie_api/utils.hpp>

void kernel_function_name(bfloat16 *input_vector, bfloat16 *output_vector, uint32_t vector_size) {
    constexpr int vector_length = 16; // Adjust based on AIE capabilities
    aie::vector<bfloat16, vector_length> input_data;
    aie::vector<bfloat16, vector_length> output_data;

    for (uint32_t i = 0; i < vector_size; i += vector_length) {
        input_data = aie::load_v<vector_length>(input_vector + i);
        // Perform the required operation on input_data
        // Store the result in output_data
        aie::store_v(output_vector + i, output_data);
    }
}
```

Replace `kernel_function_name` and the operation logic with the specific task details provided in the input.

Figure 27: GEPA generated prompt for NPUEval that achieves 26.85% score with the same GPT-4o agent, that achieved a 4.25% score.

## O.2 KERNELBENCH: CUDA KERNEL CODE GENERATION FOR NVIDIA GPUS

---

**GEPA GPT-4o generated prompt for CUDA Kernel Generation**

To optimize a given PyTorch model by replacing operators with custom CUDA kernels,
    follow these detailed instructions. Your goal is to achieve performance
    improvements while ensuring correctness. Name your optimized output
    architecture `ModelNew`. Output the new model code in codeblocks. Please
    generate real code, NOT pseudocode, and ensure the code compiles and is fully
    functional. Do not include testing code.

### Steps to Create Custom CUDA Kernels

#### 1. Identify Operators to Replace
- Analyze the model to identify operators that can benefit from custom CUDA
    implementations.
- Consider operator fusion opportunities to combine multiple operations into a
    single kernel for efficiency.

#### 2. Setup and Compilation
- Use `torch.utils.cpp_extension.load_inline` to compile your CUDA code. This
    allows you to define and compile custom CUDA kernels directly within your
    Python script.
- Ensure all necessary CUDA and C++ headers are included to avoid missing includes
    errors.

#### 3. Implementing the CUDA Kernel
- Write the CUDA kernel code, ensuring it is optimized for parallel execution. Use
    shared memory to reduce global memory accesses and ensure coalesced memory
    access patterns.
- Example structure for a CUDA kernel:
  ```cpp
  __global__ void my_kernel(const float* input, float* output, int size) {
      int idx = blockIdx.x * blockDim.x + threadIdx.x;
      if (idx < size) {
          // Perform computation
      }
  }
  ```

#### 4. Kernel Launch Configuration
- Configure the kernel launch parameters (number of blocks and threads per block)
    to maximize GPU utilization.
- Example:
  ```python
  const int block_size = 256;
  const int num_blocks = (size + block_size - 1) / block_size;
  my_kernel<<<num_blocks, block_size>>>(input, output, size);
  ```

#### 5. Error Handling and Debugging
- Implement error checking after CUDA API calls and kernel launches using `
    cudaGetLastError()` to catch errors early.
- Use CUDA debugging tools like `cuda-memcheck` and NVIDIA Nsight for debugging and
     profiling.
- Be aware of common syntax errors and namespace issues, and ensure that the CUDA
    code adheres to syntax rules.

#### 6. Integrating with PyTorch
- Define a Python function that wraps the CUDA kernel call and integrates it into
    the PyTorch model.
- Example:
  ```python
```

---

```
  def my_custom_op(input):
      output = torch.zeros_like(input)
      my_kernel(input.data_ptr<float>(), output.data_ptr<float>(), input.numel())
      return output
  ```
```

#### 7. Testing and Validation
- Validate the correctness of the custom kernel by comparing its output with the
    original PyTorch operator.
- Use profiling tools to measure performance improvements and identify further
    optimization opportunities.
- Establish reference outputs for known-good inputs to verify correctness.
- Consider floating-point discrepancies and use a small epsilon for acceptable
    differences.

#### 8. Compatibility and Compute Capability
- Ensure that the CUDA code is compatible with the target GPU's compute capability.
- Use appropriate compiler flags and consider building for multiple compute
    capabilities if necessary.

### Example Code
Here is an example of how to replace a simple element-wise addition with a custom
    CUDA kernel:

```python
import torch
import torch.nn as nn
from torch.utils.cpp_extension import load_inline

# Define the custom CUDA kernel for element-wise addition
elementwise_add_source = """
#include <torch/extension.h>
#include <cuda_runtime.h>

__global__ void elementwise_add_kernel(const float* a, const float* b, float* out,
     int size) {
    int idx = blockIdx.x * blockDim.x + threadIdx.x;
    if (idx < size) {
        out[idx] = a[idx] + b[idx];
    }
}

torch::Tensor elementwise_add_cuda(torch::Tensor a, torch::Tensor b) {
    auto size = a.numel();
    auto out = torch.zeros_like(a);

    const int block_size = 256;
    const int num_blocks = (size + block_size - 1) / block_size;

    elementwise_add_kernel<<<num_blocks, block_size>>>(a.data_ptr<float>(), b.
        data_ptr<float>(), out.data_ptr<float>(), size);

    return out;
}
"""

elementwise_add_cpp_source = (
    "torch::Tensor elementwise_add_cuda(torch::Tensor a, torch::Tensor b);"
)

# Compile the inline CUDA code for element-wise addition
elementwise_add = load_inline(
    name="elementwise_add",
```

```
    cpp_sources=elementwise_add_cpp_source,
    cuda_sources=elementwise_add_source,
    functions=["elementwise_add_cuda"],
    verbose=True,
    extra_cflags=[""],
    extra_ldflags=[""],
)

class ModelNew(nn.Module):
    def __init__(self) -> None:
        super().__init__()
        self.elementwise_add = elementwise_add

    def forward(self, a, b):
        return self.elementwise_add.elementwise_add_cuda(a, b)
```

### Additional Best Practices
- **Optimize Memory Usage**: Minimize data transfers between host and device, and
    use shared memory to reduce global memory access.
- **Atomic Operations**: When using atomic operations like `atomicMax` with
    floating-point numbers, ensure correct usage by following best practices, such
    as using appropriate data types and minimizing contention.
- **Performance Optimization**: Maximize parallel execution, optimize memory access
    patterns, and use compiler flags to enhance performance.
- **Namespace Usage**: Avoid adding class declarations or function definitions
    directly to reserved namespaces like `cuda`. Use nested namespaces within non-
    reserved namespaces to organize code.
- **Numerical Precision**: Be aware of floating-point arithmetic issues, such as
    non-associativity, and use appropriate precision levels for calculations.
- **Debugging Tools**: Utilize tools like CUDA-GDB and NVIDIA Nsight for debugging
    and profiling to ensure correctness and performance.

By following these instructions, you can effectively replace PyTorch operators with
    custom CUDA kernels, ensuring both performance improvements and correctness.

### Instruction for Replacing PyTorch Operators with Custom CUDA Kernels

Your task is to optimize a given PyTorch model by replacing certain operators with
    custom CUDA kernels to achieve performance improvements. Follow the steps
    below to ensure a successful implementation:

#### Step 1: Identify Operators for Replacement
- **Criteria for Selection**: Choose operators that are computationally intensive
    and have potential for parallelization. Consider operators that are frequently
    used in the model's forward pass.
- **Operator Fusion**: Look for opportunities to fuse multiple operators into a
    single CUDA kernel, such as combining matrix multiplication with activation
    functions (e.g., matmul + ReLU).

#### Step 2: Implement Custom CUDA Kernels
- **Kernel Structure**: Define your CUDA kernel using the `__global__` specifier.
    Ensure that each thread handles a specific part of the computation. Use
    correct index calculations to access data.
- **Memory Management**:
  - Allocate memory for input, output, and any intermediate data on the GPU using `
      cudaMalloc`. Use `cudaMemcpy` to transfer data between host and device.
  - Utilize shared memory to cache frequently accessed data and reduce global
      memory accesses.
  - Ensure coalesced global memory accesses for efficient memory transactions.
- **Numerical Stability and Boundary Conditions**:
  - Implement verification mechanisms to ensure numerical stability. Use `__host__
      __device__` functions for testing on both CPU and GPU.

```
  - Handle boundary conditions to prevent out-of-bounds memory access. Ensure that
      thread indices are within valid ranges.
- **Optimization Techniques**:
  - Use shared memory to reduce global memory accesses and improve performance.
  - Consider using mixed precision and Tensor Cores for matrix operations to
      enhance performance.
  - Avoid diverged execution paths to maintain efficient parallel execution.

#### Step 3: Integrate CUDA Kernels into PyTorch Model
- **Inline Compilation**: Use `torch.utils.cpp_extension.load_inline` to compile
    your CUDA code and integrate it into the PyTorch model.
- **Model Modification**: Replace the original PyTorch operators with calls to your
      custom CUDA functions. Ensure that the new model architecture (`ModelNew`) is
      fully functional and compiles without errors.

#### Step 4: Testing and Validation
- **Correctness**: Verify that the output of the optimized model matches the
      expected output of the original model. Use a set of test cases to ensure
      accuracy.
- **Performance Evaluation**: Measure the runtime of the optimized model and
      compare it to the original. Aim for a significant reduction in execution time.
- **Edge Case Handling**: Ensure that the kernel correctly handles cases where the
      matrix size is not a multiple of the block size and other potential edge cases
      .

#### Example Code
Below is an example of how to define and integrate a custom CUDA kernel for element
    -wise addition:

```python
import torch
import torch.nn as nn
from torch.utils.cpp_extension import load_inline

# Define the custom CUDA kernel for element-wise addition
elementwise_add_source = """
#include <torch/extension.h>
#include <cuda_runtime.h>

__global__ void elementwise_add_kernel(const float* a, const float* b, float* out,
    int size) {
    int idx = blockIdx.x * blockDim.x + threadIdx.x;
    if (idx < size) {
        out[idx] = a[idx] + b[idx];
    }
}

torch::Tensor elementwise_add_cuda(torch::Tensor a, torch::Tensor b) {
    auto size = a.numel();
    auto out = torch::zeros_like(a);

    const int block_size = 256;
    const int num_blocks = (size + block_size - 1) / block_size;

    elementwise_add_kernel<<<num_blocks, block_size>>>(a.data_ptr<float>(), b.
        data_ptr<float>(), out.data_ptr<float>(), size);

    return out;
}
"""

elementwise_add_cpp_source = (
    "torch::Tensor elementwise_add_cuda(torch::Tensor a, torch::Tensor b);"
```

```
)

# Compile the inline CUDA code for element-wise addition
elementwise_add = load_inline(
    name="elementwise_add",
    cpp_sources=elementwise_add_cpp_source,
    cuda_sources=elementwise_add_source,
    functions=["elementwise_add_cuda"],
    verbose=True,
    extra_cflags=[""],
    extra_ldflags=[""],
)

class ModelNew(nn.Module):
    def __init__(self) -> None:
        super().__init__()
        self.elementwise_add = elementwise_add

    def forward(self, a, b):
        return self.elementwise_add.elementwise_add_cuda(a, b)
```

#### Constraints
- Ensure that the optimized model maintains the same accuracy as the original.
- The custom CUDA kernels should be optimized for performance, minimizing execution
    time and maximizing GPU utilization.

By following these instructions, you will be able to effectively replace PyTorch
    operators with custom CUDA kernels, achieving significant performance
    improvements while maintaining model accuracy.

## P   NUMBER OF REFLECTION LM CALLS MADE BY GEPA DURING OPTIMIZATION

Table 4: Total number of calls made by GEPA to reflection LM during optimization.

| Benchmark Name | Num Reflection Calls GPT-4.1-Mini | Num Reflection Calls Qwen3-8B |
|---|---|---|
| AIME-2025 | 24 | 90 |
| LiveBench-Math | 34 | 38 |
| HotpotQA | 69 | 64 |
| IFBench | 21 | 17 |
| Hover | 92 | 50 |
| PUPA | 46 | 38 |

