# OpenReview forum: "GEPA: Reflective Prompt Evolution Can Outperform Reinforcement Learning"
_ICLR.cc/2026/Conference — ICLR 2026 Oral_

### Official Review · Reviewer_VhzU · 2025-10-17

**Soundness:** 3
**Presentation:** 4
**Contribution:** 3
**Rating:** 6
**Confidence:** 3

**Summary:**

This paper proposes GEPA, a novel prompt optimization algorithm based on genetic algorithms. The key components of GEPA are (1) reflective prompt mutation based on execution traces generated by LLMs and evaluation traces from environments, and (2) pareto-based candidate selection that effectively balances exploration and exploitation. The paper shows that GEPA outperforms MIPROv2, a recent prompt optimization algorithm based on Bayesian optimization, and GRPO on various tasks encompassing mathematical reasoning, multi-hop reasoning, and instruction-following.

**Strengths:**

- Overall, the paper is well-written and easy to understand. Although the paper is 94 pages long, including the appendix, I had no significant difficulty reading all of it.
- The experiments are very well-designed. The benchmarks were appropriately selected across a diverse range of tasks, and the choice of models, one reasoning model (Qwen3) and one non-reasoning model (GPT-4.1), is also considered appropriate.
- The inference-time search experiment in the appendix looks very promising.
- The proposed system is production-grade, which makes it seem highly practical.

**Weaknesses:**

- The paper's comparison is insufficient, benchmarking GEPA against only one baseline method, MIPROv2. The authors should expand the comparison to include other relevant prompt optimization techniques, such as ORPO, TextGrad, and Trace [1, 2, 3]
- Reflective prompt optimization seems not novel, as it has been widely used in TextGrad and Trace.
- Pareto-based candidate selection seems to play a crucial role in performance improvement, yet the analysis on this component appears insufficient. The paper only presents a simple comparison against greedy selection.

[1] Yang et al., Large Language Models as Optimizers, ICLR 2024 \
[2] Yuksekgonul et al., TextGrad: Automatic "Differentiation" via Text, arXiv 2024 \
[3] Cheng et al., Trace is the Next AutoDiff: Generative Optimization with Rich Feedback, Execution Traces, and LLMs, NeurIPS 2024

**Questions:**

I will raise my rating to 8 or 10 if my major questions are well addressed.

- (Major) Could you explain the rationale for not comparing GEPA with TextGrad or Trace? If feasible, I would strongly encourage the authors to run these additional experiments.
- (Major) Why are the math benchmark results for Qwen3 missing? Could you explain the reason for excluding these results? Is it because the performance improvement was not significant compared to MIPROv2? Or, were the math benchmarks added closed to the deadline, which only allowed time to conduct experiments for GPT-4.1?
- (Major) Could you please compare GEPA with MIPROv2 without few-shot examples? If few-shot examples are not effective for Qwen3 or GPT-4.1, it seems better to remove them and allocate more budget to the response.
- (Major) Could you please elaborate on lines 219-221? You mentioned that feedback can be module-specific and augmented with human-written text. Could you please explain this more concretely using HotpotQA as an example?
- (Minor) What was the monetary cost of running the experiments on GPT-4.1? Also, what would be the estimated cost if you were to use GPT-5 instead?

---

> ### Author Response · Authors · 2025-11-25
>
> We thank the reviewer for their constructive feedback and suggested experiments, which we believe significantly improved our paper. We are encouraged by the reviewer’s appreciation for the depth in design of experiments, paper’s presentation, the practicality of our proposed system, and the promise of GEPA’s demonstrated use as an inference-time search strategy.
>
> We executed all experiments suggested by the reviewer: (1) Comparison with Trace, (2) TextGrad, (3) Qwen3-8B on math benchmarks, and (4) MIPROv2 without few-shot demos. We have updated the paper to include these new results.
>
> Summary: GEPA and GEPA+Merge outperform all new baselines (Trace, TextGrad, MIPROv2-No-Demos) across all benchmarks. And the math benchmark results with Qwen3-8B follow the paper's observation.
>
> > (Major) TextGrad or Trace … strongly encourage … additional experiments.
>
> We added comparisons to TextGrad and Trace in Table 2\. Both GEPA and GEPA+Merge outperform Trace and TextGrad across all 6 benchmarks. TextGrad and Trace achieve aggregate improvements of \+6.11% and \+3.27% respectively. (again, GEPA+Merge & GEPA achieve aggregate improvements of \+13.33% and \+12.19% respectively.)
>
> While reflective optimization has been used in TextGrad/Trace, GEPA’s novelty lies in integrating reflection with a population-based evolutionary search utilizing Pareto selection. This specific combination yields significantly higher performance gains compared to strong baselines, including TextGrad, MIPROv2 (+No-demos), and Trace.
>
> > (Major) math benchmark for Qwen3 …
>
> As rightly pointed out by the reviewer, the math experiments were indeed performed close to the submission deadline. We have included new results for Qwen3-8B with math benchmarks in Table 1\. Across AIME-2025 and LiveBench-Math, while MIPROv2 resulted in a performance degradation by \-7.33% and \-2.1% respectively, GEPA improved the performance over baseline by \+4.67% and \+3.25% respectively.
>
> > (Major) ... MIPROv2 w/o few-shot examples
>
> We implemented “MIPROv2-No-Demos” (optimizing instructions only via Bayesian selection) in Table 2\. GEPA+Merge and GEPA continue to outperform MIPROv2 without few-shot examples across all 6 benchmarks individually, as well as achieving \+13.33% and \+12.19% aggregate improvement, in comparison to MIPROv2-No-Demos’ improvement of \+4.11%.
>
> > (Major) Elaborate … module-specific feedback. Explain … using HotpotQA as an example?
>
> In practise, human-written solutions or human-written rubrics are often available for training-set queries, that can be used by GEPA, when returned as text feedback in the form “Your answer {pred\_answer} was incorrect, the correct answer was {gold\_answer}. A detailed step-by-step process to arrive at the answer is {human\_written\_solution}.”, which GEPA can use to find a prompt incorporating the demonstrated process. However, this is not a necessary requirement for GEPA, and for all tasks (except AIME, which includes human-written solutions), we rely only on automatically generated text feedback.
>
> When available, GEPA can also utilize module-specific feedback. We use a multi-hop retrieval-based QA system for HotpotQA, which requires reasoning over multiple supporting documents to answer the question. For text-feedback, we identify the remaining set of gold documents to be retrieved after each hop (set difference), as the feedback for the specific hop (discussed at line 785 in the paper). For example, if the gold documents are {A, B, C} and after hop1, the program had retrieved {A, D}, then the feedback text for hop1 module can include “Your query successfully retrieved {A}, but failed to retrieve {B, C}, which are relevant to the question”. While module-specific feedback can enhance performance, it is not a requirement for GEPA: All experiments over IFBench, LiveBench-Math, AIME, PUPA use just a global text-feedback module as described in the paper.
>
> > Pareto-based candidate selection
>
> We highlight that the ablation experiments (line 324\) compare our novel pareto-based candidate selection against greedy selection, showing a \+6.4% aggregate performance improvement. Greedy selection is a commonly applied strategy (also used by TextGrad), but as demonstrated in Figure 4, it converges prematurely to local optima. Pareto selection maintains population diversity by balancing multiple objectives, avoiding such premature convergence. We also compare against and outperform MIPROv2, which uses Bayesian selection, both in instruction-only and joint instruction+few-shot optimization settings.
>
> > (Minor) Monetary cost of running the experiments on GPT-4.1? Also, what would be the estimated cost if you were to use GPT-5 instead?
>
> We have updated the relevant information in Appendix E. In summary, all experiments cost under \\$500. The total cost of running all 6 benchmarks: (1) GEPA+Merge: \\$67, (2) GEPA: \\$86, (3) MIPROv2: \\$76. A rough estimate for the cost of running all experiments with GPT-5 is around \\$2500.

---

> ### Comment · Reviewer_VhzU · 2025-11-26
> **Good rebuttal**
>
> Thank you for the detailed explanations and the extensive additional experiments. Incorporating these results has made the paper much stronger. Since most of my concerns have been fully addressed, I am increasing my score from 6 to 8.
>
> I still have one minor concern regarding your Pareto-based candidate selection. While comparisons with greedy-based methods are indeed useful, evaluating it against other approaches that maintain multiple candidates (e.g., [1]) would further strengthen your claim. That said, I believe your paper is already strong overall.
>
> [1] Reid Pryzant et al., Automatic Prompt Optimization with "Gradient Descent" and Beam Search, EMNLP 2023

---

> > ### Author Response · Authors · 2025-11-28
> >
> > We sincerely thank the reviewer for the positive evaluation and for raising the score to 8. We are encouraged that our previous responses and experiments have strengthened the paper. We also appreciate the excellent suggestion to compare GEPA against APO’s candidate selection strategy [1].
> >
> > We implemented Beam Search (beam width 4, as used in [1]) as an additional candidate selection strategy in the same evolution harness as GEPA. The results confirm that GEPA’s Pareto based candidate selection significantly outperforms Beam Search:
> > * **Beam Search (APO): +5.11%** aggregate improvement over Baseline.
> > * SelectBestCandidate (TextGrad): +6.05% aggregate improvement over Baseline.
> > * **GEPA (Pareto): +12.44%** aggregate improvement over Baseline.
> >
> > Beam Search tends to saturate the candidate pool with similar high-scoring variations (local optima), facing a problem similar to the SelectBestCandidate strategy. In contrast, GEPA’s Pareto frontier preserves candidates with varying strength/weakness profiles, enabling broader exploration while spending further exploration budget only on candidates that include a unique winning strategy for some task in the problem space. Further, one of the primary baselines in the paper (MIPROv2) uses a Bayesian selection strategy.
> >
> > We have updated Table 3 (line 329, mirrored below) and Observation 3 (line 351) in the paper to include these findings.
> >
> > **Updated Table 3: Comparison of candidate selection strategies across different tasks.**
> >
> > | Candidate Selection Strategy | HotpotQA | IFBench | Hover | PUPA | Aggregate | Improvement |
> > | :---- | :---- | :---- | :---- | :---- | :---- | :---- |
> > | Baseline | 42.33 | 36.90 | 35.33 | 80.82 | 48.84 | \--- |
> > | SelectBestCandidate | 58.33 | 30.44 | 45.33 | 85.45 | 54.89 | \+6.05 |
> > | BeamSearch | 57.33 | 36.39 | 41.00 | 81.08 | 53.95 | \+5.11 |
> > | GEPA | **62.33** | **38.61** | **52.33** | **91.85** | **61.28** | **\+12.44** |
> >
> > We hope that this new set of experiments resolve the reviewer’s final concern regarding candidate selection, and would be grateful for the reviewer’s consideration.

---

> > > ### Comment · Reviewer_VhzU · 2025-11-28
> > > **Great!**
> > >
> > > Thank you for your prompt response and the additional experiments. Now, it seems difficult to find any weaknesses in the paper. Since you have answered all my questions thoroughly and convincingly, I would like to raise my score to 10 as I promised. However, after modifying my score from 6 to 8 two days ago, the edit button disappeared and I can no longer modify my review. I will contact the AC regarding this.

---

> > > > ### Comment · Area_Chair_HT2t · 2025-11-28
> > > >
> > > > Noted. Thanks!

---

### Official Review · Reviewer_J1yS · 2025-10-24

**Soundness:** 2
**Presentation:** 3
**Contribution:** 3
**Rating:** 6
**Confidence:** 2

**Summary:**

This paper argues that the interpretable nature of natural language can serve as a richer learning signal for large language models (LLMs) compared to conventional reinforcement learning (RL) fine-tuning. To this end, the authors propose GEPA (Genetic-Pareto), a natural-language-based prompt optimization framework that integrates reflective reasoning and evolutionary search. GEPA iteratively samples trajectories, reflects on them in natural language to diagnose errors, proposes and tests improved prompts, and aggregates complementary insights from the Pareto frontier of its own optimization history. This approach reportedly enables more efficient improvement of LLM performance using significantly fewer rollouts.

**Strengths:**

1.	The motivation to reduce rollout costs during fine-tuning while maintaining or improving performance is both practically valuable and timely, given the rising computational demands of LLM alignment and adaptation.
2.	The proposed method is intuitively appealing and conceptually straightforward, making it easy to follow.

**Weaknesses:**

1.	Although Equation (2) defines two sets of parameters for potential optimization, the proposed framework appears to update only the prompt parameters, leaving the model weights fixed. In contrast, baseline methods such as GRPO involve updating the network parameters directly in their original paper. This discrepancy raises concerns about fairness in the comparison, as the two approaches optimize fundamentally different objectives and operate under different levels of model adaptability.
2.	While the problem is framed as an optimization task in Equation 2, GEPA does not employ gradient-based backpropagation ensuring convergence to an optimum. The paper should clarify how the method guarantees optimization stability or convergence, either through theoretical justification or empirical analysis. Without such assurance, it remains unclear whether GEPA reliably approaches a Pareto-optimal solution or simply performs heuristic search without convergence guarantees.

**Questions:**

See the weakness.

---

> ### Author Response · Authors · 2025-11-25
>
> We thank the reviewer for assessing our work as intuitively appealing and recognizing that our motivation to reduce rollout costs is practically valuable and timely. We believe the concerns may come from a misunderstanding of the problem setting (black-box optimization vs. gradient updates), which we clarify below.
>
> > This discrepancy raises concerns about fairness in the comparison, as the two approaches optimize fundamentally different objectives and operate under different levels of model adaptability.
>
> The reviewer raises a concern that comparing GEPA (prompt updates) to GRPO (weight updates) involve  different levels of adaptability. We believe that this difference highlights GEPA's strength:
>
> * GEPA achieves better results with stricter constraints: We demonstrate that optimizing in the prompt space ($\\Pi$) alone can outperform optimizing the model weights ($\\Theta$) via GRPO, which is a more expensive and invasive procedure. Specifically, on Qwen3-8B (Table 1), GEPA outperforms GRPO (24k rollouts) by \+6% on average (up to 19%) while using up to 35x fewer rollouts.
> * In practice, many real-world AI systems are built around models with black-box API only access (e.g., GPT-4/5, Claude, etc.), making it impossible to run GRPO optimization. In these settings, GEPA provides a practical alternative where GRPO is impossible or may not be feasible (for example, if the rollouts are expensive).
> * Finally, to ensure fair comparison within the prompt optimization landscape, we extensively compared GEPA against three strong prompt optimizer baselines: MIPROv2, Trace, and TextGrad. As shown in Table 2, our method systematically outperforms all three, achieving a \+13.33% aggregate improvement compared to MIPROv2’s \+5.64%.
>
> > GEPA does not employ gradient-based backpropagation ensuring convergence to an optimum. The paper should clarify how the method guarantees optimization stability or convergence … empirical analysis. Without such assurance, it remains unclear whether GEPA reliably approaches a Pareto-optimal solution or simply performs heuristic search without convergence guarantees.
>
> We agree that prompt optimization over discrete language space lacks the theoretical convergence proofs of gradient descent for convex optimization. However, we highlight that GEPA can utilize a validation set to track the performance of different prompt candidates, ensuring monotonically non-decreasing performance on the validation set. Further, our novel Pareto-based candidate selection strategy avoids local optima by evolving diverse candidates, showing empirical improvement of \+6.4% on aggregate compared to greedy candidate selection (used in baseline optimizers, for example, TextGrad). Finally, figures 1 and 10-13 (in the appendix) show the learning curves for all benchmarks, demonstrating consistent improvement across 6 tasks.

---

> > ### Comment · Reviewer_J1yS · 2025-11-28
> >
> > Thank you for the author's detailed reply. All my questions have been answered.

---

### Official Review · Reviewer_QSDo · 2025-10-30

**Soundness:** 2
**Presentation:** 2
**Contribution:** 2
**Rating:** 2
**Confidence:** 4

**Summary:**

This paper introduces GEPA, a reflective prompt optimizer for compound AI systems that combines textual reflection with multi-objective evolutionary search. The method iteratively mutate prompts by leveraging natural language feedback from rollouts. Evaluations across six benchmarks demonstrate its effectiveness and efficiency.

**Strengths:**

The method leverages explicit reflection and Pareto-based selection for prompt optimization, and shows its superior sample efficiency and performance against baselines.

**Weaknesses:**

The novelty of the paper is limited. Reflection has been used in prior works (e.g. Reflexion and Self-Refine) for iterative improvement, and genetic algorithms have also already been applied to prompt optimization (e.g., EvoPrompt, Promptbreeder, and EvoAgent).

The paper is hard to follow, and needs more clarification. For example, what is the detail of the Reflective Prompt Mutation, especially for the feedback. In addition, System aware merge should be described clearly in the paper. How to select these two strategies to create new candidates in each iteration, and how many prompts are selected to mutate? According to the problem statement, the system comprises multiple modules, yet the paper does not elaborate on these modules in sufficient detail.
Moreover, while GEPA is presented as optimizing only the prompts (i.e., only optimizing $\Pi$), the optimization problem defined in Equations (2) and (3) also includes the module weights $\Theta$. This apparent inconsistency should be addressed and clarified.

In GRPO, it is common practice to generate 16 rollouts per query. However, the paper mentions 24,000 rollouts in GRPO, which requires clarification. Additionally, the computational overhead of GEPA itself (e.g., the number of LLM calls required for reflection) is not discussed.

The experiments are insufficient in several aspects. First, the paper should compare with other prompt optimization methods such as EvoPrompt, PromptBreeder, and EvoAgent. Second, how do the performance of these methods vary under different computational budgets? Third, why are not all benchmarks included in Table 1. Specifically, why are the results for AIME-2025 and LiveBench-Math missing in Table 1?

**Questions:**

See the Weaknesses

---

> ### Author Response · Authors · 2025-11-25
> **Individual response (Part 1)**
>
> We thank the reviewer for their feedback, which we believe helped us improve the paper. We are encouraged that they acknowledge GEPA's superior performance compared to strong baselines like MIPRO and GRPO, which is one of our primary contributions. We have updated our paper with 5 new experiments to address the reviewer's suggestions.
>
> The reviewer raises a concern about limited novelty, noting that reflection and evolutionary search are established concepts. While these are known components, GEPA advances how we can optimize arbitrary Compound AI Systems (i.e., not just a single LLM call with supervised labels) using a novel Pareto-based Selection strategy. For instance, work like Reflexion or Self-Refine are inference-time techniques, whereas GEPA is a learning algorithm that updates the system itself using reflection.
>
> Concretely, GEPA advances over prior work in two fundamental ways:
>
> * Algorithmic Innovation (Pareto-based candidate selection): GEPA maintains a Pareto frontier of diverse, non-dominated prompt candidates over a subset of training dataset. This helps the optimization avoid premature convergence to a local optima (figure 4). Through our ablation study (Observation 3, Line 346), we empirically demonstrate that this Pareto-based selection provides a \+6.4% aggregate performance improvement over a greedy candidate selection strategy (which is used in strong baselines including TextGrad).
> * Problem Setting
>   * (Compound vs. Single-Step): The cited works (EvoPrompt, PromptBreeder) optimize a single monolithic prompt for one-step LLM calls. GEPA optimizes Compound AI Systems—policies composed of multiple interacting prompts ($\\pi\_1, \\pi\_2, \\dots, \\pi\_N$) governing distinct modules with arbitrary control flow. This structural complexity introduces credit assignment challenges that single-step optimizers cannot natively address. We instead compare against TextGrad, Trace and MIPROv2, which are natively designed for this setting and represent a stronger state-of-the-art.
>   * (Training/Compile-time vs. Inference-time): Reflexion and Self-Refine are focused on single-instance trajectory self-correction during inference. GEPA, conversely, is a compile-time framework designed to find a robust, generalized *policy (consisting of multiple prompts in an AI system)* over a distribution of tasks, which is then fixed for deployment.
>
> To address the reviewer's request for broader comparisons and similar requests from other reviewers, we have significantly expanded our evaluation (updated Table 2\) to compare GEPA against the strongest available recent Compound AI System optimizers: TextGrad, Trace (OptoPrime), and MIPROv2 (in 2 settings: with instruction only, and joint instruction and few-shot optimization).
>
> * Superior Performance: GEPA (+12.19%) and GEPA+Merge (+13.33%) consistently outperform TextGrad (+6.11%), Trace (+3.27%) and MIPROv2 (+5.64%) on aggregate across 6 diverse benchmarks.
> * Generalization (New Finding): Following a suggestion from Reviewer dW6h, we tested cross-model generalization. Prompts optimized by GEPA for (and using) Qwen3-8B but evaluated on GPT-4.1-mini achieved a \+9% aggregate improvement, surprisingly surpassing MIPROv2, TextGrad and Trace even when those methods optimized directly for (and using) GPT-4.1-mini.

---

> ### Author Response · Authors · 2025-11-25
> **Individual response continued (Part 2)**
>
> > what is the detail of the Reflective Prompt Mutation, especially for the feedback.
>
> Reflective Prompt Mutation is described in Section 3, with the full algorithm provided in Appendix D (Figure 5). Following the reviewer’s suggestion, we have updated section 3 to include more details. As outlined in the algorithm, GEPA selects a program candidate from the Pareto frontier (as per our Pareto-based candidate selection strategy), executes the selected candidate on a stochastically sampled minibatch of input queries from the trainset, tracing the program’s execution (like the inputs and outputs and reasoning by various LLM calls), and calls the user-defined evaluation function, which returns a numeric score and text feedback including details about the evaluation (like compiler error messages, failed rubrics, etc. GEPA selects the subset of prompts (among the set of prompts composed in the language program) to update based on a policy (round-robin), and a reflection LLM is then shown the (current prompt, language program trajectory, score, feedback) with the task to generate an updated prompt. The updated prompt, with the rest of the language program, is tested again on the minibatch, and if the score improves, then the new program is accepted to the candidate pool (tracking changes to the pareto-frontier).
>
> > System aware merge should be described clearly in the paper. How to select these two strategies to create new candidates in each iteration, and how many prompts are selected to mutate?
>
> We thank the reviewer for the suggestion. System-Aware Merge is described in Algorithm 4 (appendix), due to space constraints. Intuitively, merge will be helpful when there are candidates in the pool that learn complementary strategies. Algorithm 3 defines the selection criteria: candidates are merged only if they share a common ancestor but have optimized disjoint sets of prompts (complementary strategies), are pareto-optimal, and both descendents improve upon the aggregate performance of the ancestor. GEPA routinely checks if the pool has 2 such candidates, invoking merge when identified. As noted in the experimental setup (and visualized in Appendix I), these strict lineage conditions mean merge occurs sparsely; it is explicitly capped at a maximum of 5 invocations per optimization. We have now consolidated these details about System-Aware Merge in revised appendix D.1.
>
> > According to the problem statement, the system comprises multiple modules, yet the paper does not elaborate on these modules in sufficient detail.
>
> We provide details about the setup of language programs (the multi-module AI system being optimized) for each benchmark in appendix E (line 829 onwards), including details about modules used. We adopt our benchmarks and module specifications primarily from LangProBe, which is a benchmark for language programs.
>
> > While GEPA is presented as optimizing only the prompts, the optimization problem defined in Equations (2) and (3) also includes the module weights . This apparent inconsistency should be addressed and clarified.
>
> It is true that the general problem setup includes optimizing over both weights and prompts, while GEPA optimizes only over prompts. The reason the general problem is written is to compare against GRPO (which optimizes over weights). We have updated section 2 of the paper to clarify this.
>
> > In GRPO, it is common practice to generate 16 rollouts per query. However, the paper mentions 24,000 rollouts in GRPO, which requires clarification
>
> GRPO typically uses 8-16 rollouts per group, but many groups within and across mini-batches. We use standard group sizes (12-16) for GRPO training in all our experiments (line 950), while ensuring maximum hardware memory utilization. In our problem setting of sample-efficient compound AI system optimization, we measure computational budget in terms of the total number of rollouts used by the optimization algorithm (as highlighted in equation 2, line 176). In this context, we fix the total number of rollouts (budget) used in GRPO training to be 24,000, while being consistent with GRPO best practices. We achieve this by running GRPO for `24,000/batch_size` number of training steps.

---

> ### Author Response · Authors · 2025-11-25
> **Individual response (Part 3)**
>
> > Computational overhead of GEPA itself (e.g., the number of LLM calls required for reflection) is not discussed.
>
> Thanks for the suggestion. We have added the following (along with the table below) in Appendix P. As can be seen from the table, the number of LLM calls required for reflection is a small fraction of the total rollout budget. The following table provides the total number of calls made to the reflection LLM during GEPA optimization for all benchmarks with both the models:
>
> | Benchmark Name | Num Reflection Calls GPT-4.1-Mini | Num Reflection Calls Qwen3-8B | Total Rollouts Budget |
> | :---- | :---- | :---- | :---- |
> | AIME-2025 | 24 | 90 | 1839 |
> | LiveBench-Math | 34 | 38 | 1839 |
> | HotpotQA | 69 | 64 | 6871 |
> | IFBench | 21 | 17 | 3593 |
> | Hover | 92 | 50 | 7051 |
> | PUPA | 46 | 38 | 2426 |
>
> > Why are not all benchmarks included in Table 1\. Specifically, why are the results for AIME-2025 and LiveBench-Math missing in Table 1?
>
> The math experiments were performed close to the submission deadline, allowing us time to only include results on these benchmarks for GPT-4.1-Mini. We have now extended Table 1 in the paper to include both math benchmark results as well. Specifically, while MIPROv2 leads to a performance degradation of \-7.33% and \-2.1% on AIME-2025 and LiveBench-Math respectively, GEPA improves the performance by \+4.67% and \+3.25% respectively. In comparison, GRPO achieves \+10.67% and \+2.56% improvement respectively. On LiveBench-Math, GEPA, after just 38 calls to reflection LM outperforms GRPO (after training with 24,000 rollouts).
>
> > How do the performance of these methods vary under different computational budgets?
>
> We measure computational budget by the number of rollouts used in the optimization of each method. To compare different methods under different computational budgets, we show the full optimization curve for each of the optimizers studied in the paper (GRPO, MIPROv2, GEPA) providing the performance achieved at different computational budget levels (up to 24,000 rollouts) in Appendix I.

---

> > ### Comment · Reviewer_QSDo · 2025-11-26
> >
> > Thank you for your response, which addresses most of my concerns. I have updated my score accordingly.

---

### Official Review · Reviewer_pax9 · 2025-10-31

**Soundness:** 3
**Presentation:** 3
**Contribution:** 2
**Rating:** 6
**Confidence:** 4

**Summary:**

This paper introduces GEPA (Genetic-Pareto), a novel and highly sample-efficient prompt optimizer for LLMs. The authors argue that conventional adaptation methods, particularly RL techniques like GRPO, are inefficient as they require tens of thousands of trials and rely on sparse scalar rewards, failing to utilize the rich information in an LLM's own operational traces. In contrast, GEPA leverages the LLM's inherent linguistic capabilities through a process of "reflective evolution." It analyzes the detailed, natural language traces of its own attempts—including reasoning steps and tool usage—to diagnose failures, propose targeted improvements to its prompts, and learn high-level strategies. This reflective process is combined with a Pareto-based evolutionary algorithm that maintains a diverse set of high-performing prompt candidates to avoid local optima. Across multiple benchmarks, GEPA is shown to significantly outperform GRPO, achieving superior results with up to 35 times fewer rollouts, and also surpasses the state-of-the-art prompt optimizer MIPROv2. The work demonstrates that leveraging language-based reflection is a more powerful and efficient method for optimizing complex, real-world AI systems compared to traditional RL approaches.

**Strengths:**

1、It proposes a prompt optimization algorithm based on the Pareto frontier, which avoids the problem of local optima found in previous prompt optimization methods.

2、Combining prompt optimization method with the multiple-rollout approach of GRPO, it demonstrates a significant improvement in the effectiveness and sample efficiency of prompt optimization. I think the main improvement comes from contrasting multiple rollouts and produce a better prompt.

3、The performance of the proposed framework is demonstrated across multiple tasks.

**Weaknesses:**

1、Like many prompt optimization methods, this approach is highly dependent on the model's own reasoning and summarization capabilities. It requires the model to analyze successful and failed rollouts and distill effective textual experience into the prompt. Consequently, this method may not be suitable for less capable LLMs, an aspect the paper fails to analyze.

2、The paper does not analyze the types of tasks for which this prompt optimization method is effective and those where it might not be. For instance, on tasks like mathematical reasoning, the effectiveness of this approach is not guaranteed.

3、Does the prompt exclusively contain summarized experiences, or does it also incorporate the trajectories of failed rollouts?

4、I am concerned that the context within the prompt could become increasingly long. Theoretically, as the number of learning samples grows, the volume of summarized experience in the prompt should also increase.

5、The paper lacks some necessary citations. [1] is very similar to the process in this paper; both automatically optimize prompts through LLM.

Missing citations：

[1] Zhang, Wenqi, et al. "Agent-pro: Learning to evolve via policy-level reflection and optimization." ACL 2024.

**Questions:**

Stated in Weaknesses

---

> ### Author Response · Authors · 2025-11-25
>
> We thank the reviewer for the detailed review. We are encouraged that the reviewer agrees with the high sample efficiency, effectiveness, and novelty of our proposed approach, and highlights the significant improvements demonstrated across multiple tasks. We have updated our paper to discuss the “Agent-pro” work in the related-work section, and respond to the reviewer’s questions inline below.
>
> > approach … dependent on the model's own reasoning and summarization capabilities. May not be suitable for less capable LLMs, an aspect the paper fails to analyze.
>
> The reviewer rightly points out that GEPA leverages a language model’s reasoning and summarization abilities to perform reflective prompt mutation. In our experiments, we test 3 models of varying capabilities including a small open-source model (Qwen3-8B) alongside 2 closed-source models (GPT-4.1-mini and GPT-4o (Line 389)). We found that all models, including Qwen3-8B, possess sufficient reflective capability to improve their own prompts, boosting performance by as much as +12.44% on aggregate across 6 benchmarks, showing that frontier-scale reasoning is not required for GEPA to succeed. Further, in practice, the reflection model is invoked just about once every 60 rollouts. Hence, for even smaller / less capable models than Qwen3-8B, it is feasible to use a relatively stronger model (like Qwen3-8B) as the reflection model.
>
> > The paper does not analyze the types of tasks for which this prompt optimization method is effective and those where it might not be. For instance, on tasks like mathematical reasoning, the effectiveness of this approach is not guaranteed.
>
> We acknowledge that there could be tasks for which prompt optimization may not be effective (especially tasks vastly out-of-distribution for the models). That said, our  results demonstrate GEPA’s performance across 6 diverse tasks, including those that tackle math and complex reasoning:
> * Mathematical Reasoning: On AIME-2025, GEPA improves accuracy by +10.0% (from 49.33% to 59.33%) and on LiveBench-Math by +5.9% (from 58.20% to 64.13%) using GPT-4.1 Mini.
> * Complex Code Generation: We apply GEPA to a challenging code optimization task (NPU Kernel optimization). While the baseline GPT-4o agent achieved only 4.25% on NPUEval, GEPA improved the same agent’s performance to 26.85%.
> * Generalization: GEPA consistently improves upon baselines on Multi-hop QA (HotpotQA) (up to +31% gain on GPT-4.1 Mini), Privacy Delegation (PUPA) (+17% gain), Instruction Following (IFBench) (+8%) and Claim Verification (Hover) (+10%).
>
> > Does the prompt exclusively contain summarized experiences, or does it also incorporate the trajectories of failed rollouts? I am concerned that the context within the prompt could become increasingly long. Theoretically, as the number of learning samples grows, the volume of summarized experience in the prompt should also increase.
>
> We clarify that GEPA does not maintain a growing list of summarized experiences, or store successful/failed trajectories. Instead, GEPA uses an LLM to fully rewrite existing prompts by reflecting on new rollouts, distilling observed trajectories into high-level declarative instructions and strategies, effectively merging related instructions. Figure 25 and Appendix M.1 shows the intermediate prompts generated by GEPA starting from the base prompt to the best performing prompt. As can be seen, while prompts get increasingly nuanced, the prompt size grows much slower than the number of observed trajectories. In spite of the increasing nuance in GEPA-generated prompts, Figures 16 and 17 in the Appendix show that GEPA-generated prompts are up to 9.2x shorter than those generated by MIPROv2 while outperforming it significantly. That said, future work should certainly study “regularizing” the prompts more explicitly, including against excessive length or complexity.
>
>
> > The paper lacks some necessary citations. [1] is very similar to the process in this paper; both automatically optimize prompts through LLM.
>
> We thank the reviewer for pointing us to this reference, we have updated the related work section of our paper to discuss it.

---

> > ### Comment · Reviewer_pax9 · 2025-11-27
> > **Thank for response**
> >
> > Thank you for the author's detailed reply. All my concerns have been answered.

---

### Official Review · Reviewer_dW6h · 2025-11-01

**Soundness:** 3
**Presentation:** 3
**Contribution:** 4
**Rating:** 10
**Confidence:** 4

**Summary:**

This paper addresses the challenge of post-training large language models (LLMs). Existing RL–based methods, such as GRPO, demand thousands of rollouts, leading to substantial computational costs. To overcome this, the authors introduce GEPA, an evolutionary, LLM-driven approach to prompt optimization. Unlike RL, which relies solely on scalar rewards, GEPA leverages execution and evaluation traces to provide richer, denser learning signals. It also maintains a Pareto set during optimization, promoting solution diversity. Experiments across multiple benchmarks demonstrate that GEPA is up to 35× more sample-efficient than GRPO and achieves superior performance compared to both GRPO and other prompt optimization frameworks.

**Strengths:**

1. The paper addresses an important and timely challenge—improving the sample efficiency of LLM post-training—and presents a well-motivated approach to tackle it.

2. The proposed method overcomes key challenges of existing RL-based approaches, such as credit assignment from single scalar rewards and low sample efficiency.

3. The results are compelling and challenge the existing paradigm of RL-based post-training in the field of LLM post-training.

**Weaknesses:**

1. (Line 220) “Human-written explanations” – does the human have to provide textual feedback based on the scalar reward?

2. What is r in Algorithm 4?

3. Other LLM-based evolutionary methods have been explored in the context of adversarial prompt generation and red-teaming, and these works should be appropriately cited, for example [1].

[1] Samvelyan, M. et al. (2024). Rainbow Teaming: Open-Ended Generation of Diverse Adversarial Prompts. In NeurIPS 2024.

**Questions:**

1. In Observation 4, could the weaker performance from adding few-shot demonstrations be attributed to the evolutionary framework of MIPROv2? In other words, how might GEPA perform if it were similarly constrained to include few-shot demonstrations alongside the instructions?

2. In the experiments without a validation set (Observation 1), does this hurt the generalization performance?

3. An additional advantage of GEPA is that the optimized prompts can potentially generalize across models. Have the authors tried this?

---

> ### Author Response · Authors · 2025-11-25
>
> We sincerely thank the reviewer for their feedback. We are thrilled that the reviewer finds our method timely, addressing key challenges of existing approaches, and the experiments compelling. The reviewer’s suggested experiment on evaluating the generalization of GEPA-generated prompts across models was very insightful and led to a new finding, which we have included in the paper. We have also updated the related work section to discuss Rainbow Teaming \[1\]. Please find our responses to the reviewer’s questions inline below.
>
> > An additional advantage of GEPA is that the optimized prompts can potentially generalize across models. Have the authors tried this?
>
> We implemented the suggested approach and find that GEPA prompts optimized for (and using) Qwen3-8B, generalize well to GPT-4,1-Mini, achieving a 9% aggregate improvement over GPT-4.1-mini’s baseline results. This exceeds the improvements by strong optimizers including TextGrad (+6.11%), MIPROv2 (+5.64%) and Trace (+3.27%), even when they are optimized for GPT-4.1-Mini directly. We have included this new finding as observation 6 (line 1105\) and the new results have been added as “GEPA-Qwen-Opt” in Table 2\.
>
> > (Line 220\) “Human-written explanations” – does the human have to provide textual feedback based on the scalar reward?
>
> GEPA does not rely on human-written explanations as feedback. However, in practice, there are domains where human-graders rate the AI system’s responses, along with providing detailed feedback justifying their scalar ratings. In line 229 (updated), we only highlight that GEPA can additionally utilize such rich information, whenever available (especially in real-world deployments). We have clarified this in the paper.
>
> > What is r in Algorithm 4?
>
> *r* represents a seeded stochastic sampler, and it was an oversight not to clarify this in the paper. We have updated the caption to reflect this.
>
> > In Observation 4, could the weaker performance from adding few-shot demonstrations be attributed to the evolutionary framework of MIPROv2?
>
> To test if optimizing few-shot demonstrations was hindering the baseline, we ran a new ablation, MIPROv2-No-Demos, which optimizes instructions-only using MIPROv2. Removing demonstrations decreased performance (Aggregate: 58.67 to 57.14), indicating that the few-shot examples were beneficial to MIPROv2, not detrimental. We have added the new results for MIPROv2-No-Demos to Table 2 in the paper.
>
> > Other LLM-based evolutionary methods have been explored in the context of adversarial prompt generation and red-teaming, and these works should be appropriately cited, for example \[1\].
>
> We appreciate that the reviewer pointed out this important relevant work. We have updated the paper to discuss the reference.

---

> > ### Comment · Reviewer_dW6h · 2025-11-26
> >
> > Thank you for the comprehensive responses. All of my concerns have now been resolved.

---

### Author Response · Authors · 2025-11-25
**General Response**

We thank the reviewers for their suggestions, which have helped us improve the paper. We are encouraged that they find our work to address a timely and valuable problem (dW6h, J1yS) with a production-grade method (VhzU) that is novel (pax9, VhzU) and intuitively appealing (J1yS), effectively overcoming key limitations of existing RL approaches (dW6h) by leveraging language-based reflection and Pareto optimization (pax9, QSDo). Furthermore, we are glad the reviewers appreciate the experiments and presentation (VhzU), which demonstrate our approach’s superior sample efficiency across diverse tasks (dW6h, pax9, QSDo).

Following the reviewers’ suggestions, we provide results from 5 new experiments: Comparing GEPA against strong baselines including (1) TextGrad, (2) Trace (OptoPrime), (3) MIPRO without few-shot optimization (suggested by VhzU), (4) Testing the generalizability of prompt optimized by GEPA for one model (Qwen3-8B) to another (GPT-4.1-Mini) (suggested by dW6h), and (5) evaluating GEPA with Qwen3-8B on 2 math benchmarks (AIME-2025, LiveBench-Math).

The updated results are summarized in Tables 1 and 2 below (mirroring the now updated Tables 1 and 2 from the paper).

GEPA+Merge (+13.33%) and GEPA (+12.19%) continue to have superior performance in comparison to strong baselines of MIPROv2 (+5.64%), TextGrad (+6.11%), and Trace (OptoPrime) (+3.27%) optimizers across all 6 benchmarks evaluated.

Moreover, the experiment suggested by reviewer dW6h led to a new finding that GEPA-generated prompts generalize across models: Table 2, row “GEPA-Qwen-Opt”, which demonstrates the performance of prompts optimized by GEPA for (and using) Qwen3-8B, but evaluated (without modification) on GPT-4.1-mini. “GEPA-Qwen-Opt” achieves an improvement of \+9% aggregate across 6 benchmarks, surpassing MIPROv2 (+5.64%), TextGrad (+6.11%) and Trace (+3.27%), even though all optimized for GPT-4.1-mini.

We address specific questions and detailed feedback in our individual responses to each reviewer below.

### **Table 1: Benchmark results for different optimizers with Qwen3-8B**

*GEPA and GEPA+Merge achieve better performance than GRPO with far fewer rollouts on all benchmarks except AIME.*

| Qwen3 8B | HotpotQA | IFBench | Hover | PUPA | AIME-2025 | LiveBench-Math | Aggregate | Improvement |
| :---- | :---- | :---- | :---- | :---- | :---- | :---- | :---- | :---- |
| Baseline | 42.33 | 36.90 | 35.33 | 80.82 | 27.33 | 48.70 | 45.23 | \--- |
| GRPO | 43.33 | 35.88 | 38.67 | 86.66 | **38.00** | 51.26 | 48.91 | \+3.68 |
| MIPROv2 | 55.33 | 36.22 | 47.33 | 81.55 | 20.00 | 46.60 | 47.84 | \+2.61 |
| GEPA | 62.33 | **38.61** | **52.33** | **91.85** | 32.00 | **51.95** | **54.85** | **\+9.62** |
| GEPA+Merge | **64.33** | 28.23 | 51.67 | 86.26 | 32.00 | **51.95** | 52.40 | \+7.17 |
| **Total optimization budget (\# rollouts)** |  |  |  |  |  |  |  |  |
| GEPA (+Merge) | 6871 | 3593 | 7051 | 2426 | 1839 | 1839 | 3936 | \--- |
| GRPO | 24000 | 24000 | 24000 | 24000 | 24000 | 24000 | 24000 | \--- |

### **Table 2: Benchmark results for different optimizers evaluated on GPT-4.1 Mini**

*GEPA works off-the-shelf on closed-source models, outperforming state-of-the-art prompt optimizers. Additionally, GEPA-optimized prompts demonstrate cross-model generalization.*

| GPT-4.1 Mini | HotpotQA | IFBench | Hover | PUPA | AIME-2025 | LiveBench-Math | Aggregate | Improvement |
| :---- | :---- | :---- | :---- | :---- | :---- | :---- | :---- | :---- |
| Baseline | 38.00 | 47.79 | 46.33 | 78.57 | 49.33 | 58.20 | 53.03 | \--- |
| Trace (OptoPrime) | 60.33 | 51.19 | 46.00 | 74.18 | 45.33 | 60.74 | 56.30 | \+3.27 |
| MIPROv2-No-Demos | 38.00 | 52.04 | 51.33 | 91.85 | 48.67 | 60.97 | 57.14 | \+4.11 |
| MIPROv2 | 58.00 | 49.15 | 48.33 | 83.37 | 51.33 | 61.84 | 58.67 | \+5.64 |
| TextGrad | 62.33 | 48.64 | 47.67 | 85.68 | 46.67 | 63.84 | 59.14 | \+6.11 |
| GEPA | **69.00** | 52.72 | 51.67 | 94.47 | **59.33** | **64.13** | 65.22 | \+12.19 |
| GEPA+Merge | 65.67 | **55.95** | **56.67** | **96.46** | **59.33** | **64.13** | **66.36** | **\+13.33** |
| **Optimized with Qwen3-8B** |  |  |  |  |  |  |  |  |
| GEPA-Qwen-Opt | 65.67 | 49.83 | 54.67 | 90.05 | 52.67 | 59.31 | 62.03 | \+9.00 |

---

### Author Response · Authors · 2025-12-03
**Summary of Discussion: New Experiments, Baselines, and Clarifications**

We thank the reviewers for the productive discussion. We are encouraged that reviewers found our work to address a timely and valuable problem with a production-grade method that is novel and intuitively appealing. We were also encouraged by the reviewers' engagement, confirming that our responses addressed their concerns and also for significantly increasing their scores after the discussion.

In response to reviewer feedback and over the discussion period, we conducted **6 new experiments** and updated our paper with several additional strong baselines, ablations, and clarifications. Below is a summary of our responses and updates:

### **1\. New Baselines and Comparative Analysis**

We significantly expanded our evaluation to compare GEPA against strong, state-of-the-art prompt optimizers and candidate selection strategies:

* **TextGrad & Trace (OptoPrime):** We compared GEPA against strong compound AI system optimizers TextGrad and Trace. GEPA (+12.19%) and GEPA+Merge (+13.33%) consistently outperform both TextGrad (+6.11%) and Trace (+3.27%) across all benchmarks (updated Table 2).
* **Candidate Selection Strategy (Beam Search used in APO):** To validate our Pareto-based candidate selection, we implemented Beam Search (width 4, as used in APO) in our evolution harness as a baseline. GEPA (+12.44%) significantly outperforms Beam Search (used in APO) (+5.11%) and SelectBestCandidate (used in TextGrad) (+6.05%) (updated Table 3), confirming that our Pareto frontier based selection strategy better preserves candidate diversity compared to approaches that are prone to local optima.
* **MIPROv2 Ablation (No-Demos):** Following reviewer suggestion, we investigated if MIPROv2’s performance was hindered by optimizing few-shot demonstrations and if it would improve by optimizing instructions only with MIPROv2. Results showed that removing demonstrations decreased MIPROv2's performance (Aggregate 58.67% \-\> 57.14%), confirming its baseline was competitive.

### **2\. Cross-Model Generalization**

Following Reviewer dW6h’s suggestion, we tested if prompts optimized by GEPA on a smaller model could generalize to a larger one.

* **Result:** Prompts optimized for (and using) **Qwen3-8B** but evaluated on **GPT-4.1-Mini** ("GEPA-Qwen-Opt") achieved a **\+9.00% aggregate improvement**.
* **Significance:** This result beats strong baselines like MIPROv2 (+5.64%) and TextGrad (+6.11%) *even when those baselines were optimized directly for (and using) GPT-4.1-Mini*. This demonstrates GEPA's ability to discover robust, transferable prompt strategies.

### **3\. Expanded Benchmarks**

We expanded Qwen3-8B evaluation to math benchmarks, **AIME-2025** and **LiveBench-Math**, in our evaluation suite (updated Table 1), confirming that GEPA leads to +4.67% and +3.25% performance improvements on these benchmarks for (and using) Qwen3-8B model.

### **4\. Clarifications and Paper Updates**

* We updated the paper to discuss several clarifications and related work suggested by the reviewers.
* We have also highlighted all updates to the paper during the discussion period marked in blue.

We believe these new results and clarifications strongly reinforce GEPA’s position as a highly sample-efficient and effective optimizer for compound AI systems.

---

### Meta-Review · Area_Chair_868w · 2026-01-06

**Summary:**

This seems like a very strong paper with a very strong rebuttal that seems to have convinced the reviewer regarding their respective concerns. I do could not identify any blatant concerns that would still be open. The original scores were already quite high with only one review (2,6,6,6,10). The reviewer who gave a two was quite satisfied with the rebuttal saying "Thank you for your response, which addresses most of my concerns. I have updated my score accordingly."

**Reviewer Scores:**

Reviewer dW6h: 10 -> 10

Reviewer pax9: 6 -> 6

Reviewer QSDo: 2-> 6

Reviewer J1yS: 6-> 6

Reviewer VhzU: 6-> 10

---

### Decision · Program_Chairs · 2026-01-26

Accept (Oral)